# SH3KBP1 promotes skeletal myofiber formation and functionality through ER/SR architecture integrity

Alexandre Guiraud[1,7], Nathalie Couturier[1,7], Emilie Christin [1], Léa Castellano[1], Marine Daura [1], Carole Kretz-Remy[1], Alexandre Janin [1], Alireza Ghasemizadeh[1], Peggy del Carmine[1], Laloe Monteiro [1], Ludivine Rotard [1], Colline Sanchez[1], Vincent Jacquemond[1], Claire Burny[2], Stéphane Janczarski[2], Anne-Cécile Durieux[3], David Arnould [3], Norma Beatriz Romero[4], Mai Thao Bui [4], Vladimir L Buchman [5], Laura Julien[6], Marc Bitoun [6] & Vincent Gache [1]✉

## Abstract

**Dynamic changes in the arrangement of myonuclei and the organization of the sarcoplasmic reticulum are important determinants of myofiber formation and muscle function. To find factors associated with muscle integrity, we perform an siRNA screen and identify SH3KBP1 as a new factor controlling myoblast fusion, myonuclear positioning, and myotube elongation. We find that the N-terminus of SH3KBP1 binds to dynamin-2 while the C-terminus associates with the endoplasmic reticulum through calnexin, which in turn control myonuclei dynamics and ER integrity, respectively. Additionally, in mature muscle fibers, SH3KBP1 contributes to the formation of triads and modulates the Excitation-Contraction Coupling process efficiency. In Dnm2^{R465W/+} mice, a model for centronuclear myopathy (CNM), depletion of Sh3kbp1 expression aggravates CNM-related atrophic phenotypes and impaired autophagic flux in mutant skeletal muscle fiber. Altogether, our results identify SH3KBP1 as a new regulator of myofiber integrity and function.**

**Keywords** Endoplasmic Reticulum; Myonuclear Positioning; Centronuclear Myopathies; Triads
**Subject Categories** Autophagy & Cell Death; Molecular Biology of Disease; Musculoskeletal System

## Introduction

Skeletal muscles are essential for movement and metabolism. The building block of skeletal muscles is the muscle fiber made by a timely-controlled fusion process, allowing hundreds of myocytes to fuse together and build the myofiber (Demonbreun et al, 2015).

Myonuclei positioning in these giant post-mitotic syncytia is a challenging process. During muscle fiber development, myonuclei undergo several types of successive movements that sets their final positioning along the muscle fibers length and contribute to muscle fibers size (Azevedo and Baylies, 2020; Cramer et al, 2020). In mature myofibers, myonuclei positioning is finely regulated and is mainly dependent on muscle fibers ability to maintain a defined distance between adjacent myonuclei in a cytoplasmic adapted context related to either myofibers type, size and age (Bruusgaard et al, 2006; Qaisar and Larsson, 2014; Liu et al, 2009). The main actor involved in the dynamic control of myonuclei spreading, in developing muscle fiber, is the microtubule (MTs) network with its associated proteome such as Maps (for Microtubule-associated proteins) and molecular motors (kinesins and dyneins family members) that adjust myonuclei motion in the course of muscle fibers development and maturation (Gache et al, 2017; Metzger et al, 2012). In this view, myonuclei mis-localization and/or mis-shaping impacts cell fate, highlighting the need for muscle fibers to maintain a correct myonuclei setting to keep optimal muscle functionalities (Metzger et al, 2012; Ghasemizadeh et al, 2021; Roman et al, 2017). The respective contribution of other cytoskeletal or membrane-related networks and their implication in the control of myonuclei spreading remain to be characterized.

Abnormal myonuclei localization in muscle fibers that are not directly linked to excessive regenerative process is the hallmark of a group of human myopathies called centronuclear myopathies (CNMs) (Romero and Bitoun, 2011). The majority of defective proteins implicated in CNMs are involved in various aspects of membrane trafficking and remodeling and are relevant to essential cellular processes, including endocytosis, intracellular vesicle trafficking, and autophagy (Romero and Laporte, 2013). One common process affected in CNMs-related skeletal muscle fibers is the "excitation-contraction coupling" (ECC), based on the interplay between the sarcoplasmic reticulum (SR) and Transverse (T)-tubules at the triads. SR is a complex network of tubular endoplasmic reticulum (ER), essential for the transduction of the

¹CNRS/UCBL1 UMR 5261 - INSERM U1315, U1217, INMG-PGNM, INSERM, CNRS, Claude Bernard University Lyon 1, Lyon, France. ²Laboratoire de Biologie et Modélisation de la Cellule, ENS de Lyon, Lyon CEDEX 07, France. ³Laboratoire Interuniversitaire de Biologie de la Motricité, Université de Lyon, Université Jean Monnet, Saint Etienne, France. ⁴Unité de Morphologie Neuromusculaire, Institut de Myologie, Groupe Hospitalier Universitaire La Pitié-Salpêtrière, Paris, France. ⁵School of Biosciences, Cardiff University, Museum Avenue, Cardiff CF10 3AX, UK. ⁶Sorbonne Université, INSERM, Institute of Myology, Centre of Research in Myology, F-75013 Paris, France. ⁷These authors contributed equally: Alexandre Guiraud, Nathalie Couturier. ✉E-mail: Vincent.gache@inserm.fr

electrical impulse into contractile force through the release of stored calcium ions. The T-tubule network is continuous with the muscle cell plasma membrane (PM) and begins from the radial invagination of the PM in a repeated pattern at each sarcomere. With ECC, neuromuscular action potentials are transmitted along muscle T-tubule membrane to the T-tubule-SR junctions, triggering coordinated SR $Ca^{2+}$ release that allows synchronous sarcomere contraction (Allard, 2018). Although the importance of triads formation, organization and maintenance for muscle function is well established, the regulation and dynamics of this membranous system is largely unexplored and the potential link with myonuclei localization is still pending.

To identify mechanisms that contribute to the interplay between myonuclei organization/positioning and muscle fiber settings and functions, we conducted a limited siRNA screen on candidates previously identified as cytoskeleton interactors/modulators and identified SH3-domain-containing kinase-binding protein 1 (SH3KBP1) as a new factor controlling myotubes formation/elongation, myonuclear positioning and triads formation during the muscle myofiber development. SH3KBP1, also known as Ruk/CIN85 (Cbl-interacting protein of 85 kDa) is a ubiquitously expressed adapter protein, involved in multiple cellular processes including signal transduction and cytoskeleton remodeling (Havrylov et al, 2010; Buchman et al, 2002). SH3KBP1 is also associated with several compartments involved in membrane trafficking such as the Golgi complex and is mainly concentrated in COPI-positive subdomains (Havrylov et al, 2008). In the present study, we show that SH3KBP1 links the Dynamin-2 (DNM2) to the endoplasmic reticulum (ER) in developing skeletal muscle fibers especially at the I-band and contributes to T-tubule formation and functionality in skeletal muscle. Moreover, its downregulation contributes to an atrophic phenotype in wild-type mouse or in a knock-in mouse model expressing the most frequent mutation causing autosomal dominant centronuclear myopathy (AD-CNM-$Dnm2^{R465W/+}$). Altogether our results identify SH3KBP1 as a new regulator of myofiber integrity and function, through its involvement in myoblast fusion, myotubes elongation, myonuclear positioning/dynamic and ER scaffolding in developing muscle fibers and, in mature myofibers, through the scaffolding/maintenance of triads structure to ensure efficient SR $Ca^{2+}$ release and control of the autophagic pathway.

## Results

### SH3KBP1 is required for myonuclear positioning steps and elongation of developing myofibers

Myofibers are formed by the fusion of hundreds of specialized mononucleated cells (myoblasts/myocytes), which build syncytial cells named "myotubes". The early step of myotubes formation is defined by a phase of myotube elongation, depending on an interplay between myonuclei content/positioning and microtubules (MTs) organization, mainly controlled by MT-associated-proteins and molecular motor proteins (Gache et al, 2017; Hansson et al, 2020). Nevertheless, organelles such as Golgi apparatus and/or Endoplasmic Reticulum, through the modulation of their morphology can interfere with the MTs network organization and thus modulate cell polarity and nuclear positioning (Yang et al, 2017; Janota et al, 2022). To identify new factors that contribute to myotubes elongation in relation with

myonuclear spreading, we performed a limited siRNA screen on 30 candidates, previously identified as cytoskeleton interactors/modulators and associated with either the Golgi apparatus (GA), the endoplasmic reticulum (ER), the plasma membrane (PM) or the cytosolic compartment (Fig. 1A). Briefly, in vitro primary isolated murine myoblasts were first transfected with a pool of siRNA targeting no sequence (scramble siRNA) or candidate mRNA sequences and were then induced to fuse and form myotubes during 3 days. Myotubes length (Dataset EV1) and the distance between myotube centroid and each myonucleus (DiMycMyo) (Dataset EV2) were measured to follow nuclear spreading and classified according to myonuclei content by myotubes. Data were then associated and plotted to compare the evolution of the DiMycMyo parameter with myotubes length in each siRNA condition (Fig. 1B; Dataset EV3). Interestingly, we found that nearly all selected candidates appear as negative regulator of myotube length expansion, as knocking them down lead to an increase in myotube length relative to the scramble control condition, with the exception of KIFAP3 protein. As expected, an increase of myotube length was correlated with the increase in the DiMycMyo parameter, showing that elongation is mainly associated with myonuclear spreading capacity along the length of myotubes (Fig. 1B). Surprisingly, one candidate, SH3KBP1/Ruk/Cin85, appeared as an outlier regarding DiMycMyo/myotube length correlation when its expression was inhibited (Figs. 1B and EV1A–D). The detailed analysis of the evolution of myotube length according to myonuclear accretion showed that in control condition, a nearly linear relation between accretion of myonuclei into myotubes and expansion of myotubes length is observed, with an average of 10% increase of length after each myonuclear accretion (Fig. EV1C). In sh3kbp1-depleted myotubes, length repartition is homogenously extended by nearly 50% in myotubes containing up to 11 myonuclei (Fig. EV1C). The DiMycMyo quantification revealed a failure of homogeneous myonuclei spreading along myotubes length with a tendency for myonuclei accumulation at myotubes extremities/tips (Figs. 1C,D, arrows and EV1D). The Myonuclei Spreading Graph, representing the statistical probability to map myonucleus along the length of myotubes (Ghasemizadeh et al, 2021) confirmed this observation, as "statistical clustering zones" were present as four zones in scramble conditions and as eight zones in sh3kbp1-depleted myotubes, with less frequent myonuclei in the center of the myotube compared to the tips of myotubes (Fig. 1E,F). To further investigate the long-term consequences of sh3kbp1 depletion on myonuclei positioning, primary myotubes were induced to mature in vitro for 10 days as previously described (Fig. EV1E–I) (Falcone et al, 2014). In these conditions, myonuclei were compressed between myofiber's plasma membrane and contractile apparatus. We observed that sh3kbp1 depletion caused a significant reduction in the mean distance between adjacent myonuclei (Fig. EV1I), showing that precocious aggregation has a long-term impact on the myonuclei organization in mature myofiber. Overall, these data show that among screened candidates, SH3KBP1 acts as an "anti-elongation" factor by controlling myonuclear spreading in myotubes that, in turn negatively controls myotube elongation.

### SH3KBP1 controls myoblasts/myotubes fusion

Myoblasts/myotubes elongation and alignment is a key process that controls myoblast/myotubes fusion (Louis et al, 2008). As sh3kbp1 depletion led to increased myotubes length, we wondered if fusion

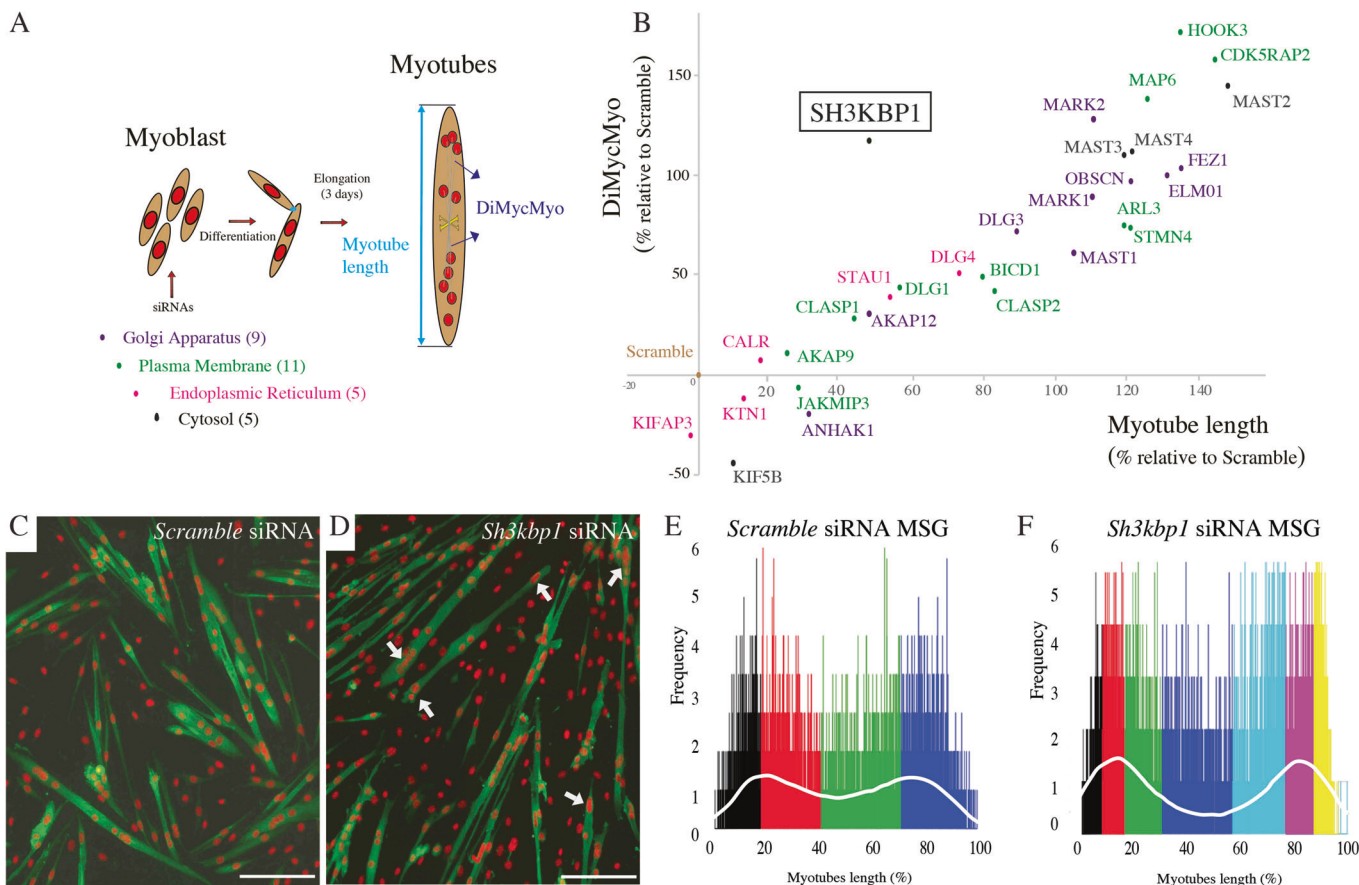

**Figure 1. Screening for candidates involved in the interplay between myotube length and myonuclear positioning: identification of the role of *sh3kbp1*.**

(A) Scheme of the sequential steps realized to obtain primary myotubes knocked-down for proteins associated with either the Golgi apparatus (purple), the endoplasmic reticulum (pink), the plasma membrane (green) or the cytosolic compartment (black). Pools of three different siRNAs were transfected 24 h before myoblasts fusion and cells were induced to fuse for 3 days. Quantification of myotubes length (see also Dataset EV1) and of the distance between myotubes centroid and each myonuclei (DiMycMyo) (see also Dataset EV2) were extracted from each myotube, according to the number of myonuclei. (B) Evolution of the DiMycMyo parameter as a function of myotubes length in each siRNA condition, normalized to scramble siRNA condition (see also Dataset EV3). (C, D) Representative immunofluorescence staining of myosin heavy chain (green) and myonuclei (red) in primary myotubes treated with scramble (C) or a pool of two different siRNAs targeting *Sh3kbp1* mRNA (D) after 3 days of differentiation; white arrows indicate "tips-aggregated myonuclei". Scale bar: 150 μm. (E, F) Myonuclei Spreading Graph (MSG) representing the statistical spatial distribution of myonuclei along myotubes in scramble (E) and *Sh3kbp1* siRNA-treated myotubes (F); white-line represents the mean value of the statistical frequency repartition.

could also be modified during differentiation. We thus evaluated the fusion capacity of primary myoblasts treated with siRNA targeting *sh3kbp1* mRNA (Fig. 2). We first analyzed myoblasts commitment, by evaluating the percentage of cells with a positive staining for myosin heavy chain (MHC+ cells) and found a normal commitment into the muscle differentiation process, suggesting that SH3KBP1 is not acting on the entry in the muscle differentiation program (Fig. 2A–C). We next assessed myoblast fusion capacity using fusion index quantification, reflecting the number of nuclei specifically present in myotubes cells (MHC+ cells). We found a significant increase of the total number of myonuclei in myotubes (Fig. 2D) and per se in the average number of myonuclei per myotube (Fig. 2E). Accordingly, in SH3KBP1-depleted conditions, the distribution of myotubes with respect to myonuclei content revealed an enhancement of myotubes with more than 10 myonuclei compared to control conditions (Fig. 2F).

We next investigated on the expression of SH3KBP1 during the course of in vitro myotubes formation using C2C12 cells (Fig. 3A,B). We show a slight enhancement of SH3KBP1 protein levels in the early stages of myotubes formation (Fig. 3A). These results were confirmed using RT-qPCR techniques where a twofold increase of *sh3kbp1* mRNA expression was observed at the onset of the differentiation step (Fig. 3B). To confirm SH3KBP1 implication during the early steps of muscle cell formation, we stably knocked-down *Sh3kbp1* mRNA expression in C2C12 cells using a small hairpin interfering RNA (shRNA) (Fig. 3C). After 3 days of differentiation, C2C12 myoblasts entered the "muscle differentiation program", illustrated by the detection of myosin heavy chain-positive (MHC +) cells (Fig. 3D,E, day 3). After 6 days of differentiation, a breaking event in fusion capacity is observed in *Sh3kbp1* knockdown conditions (Fig. 3D,E, day 6). In control conditions, myonuclei spread along thin myotubes length, while *Sh3kbp1* knockdown gives rise to huge myotubes in

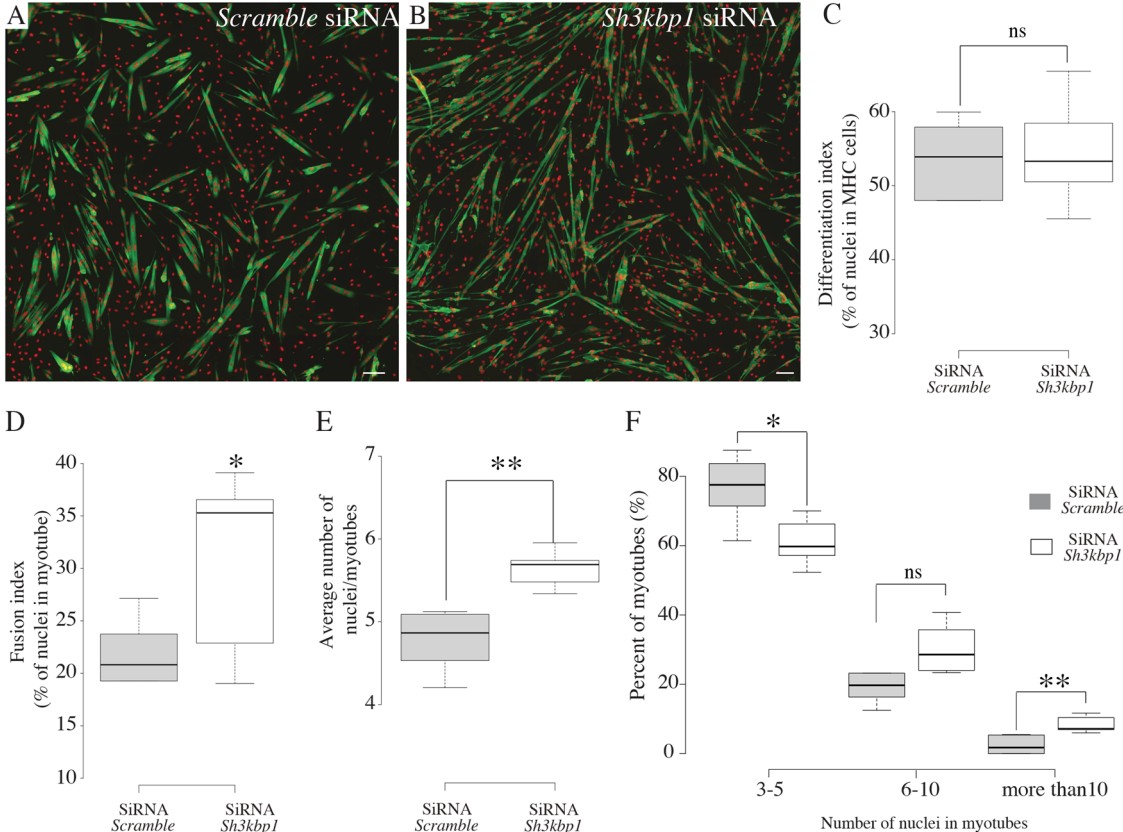

**Figure 2. SH3KBP1 controls myoblasts fusion parameters.**

(A, B) Representative immunofluorescent images of 3 days differentiated primary myotubes stained for myosin heavy chain (green) and myonuclei (red) and treated with either scramble (A) or with a pool of 2 individual *Sh3kbp1* siRNAs (B); scale bar: 50 μm. (C–F) ($n = 5$; biological replicates) Statistical analysis performed using unpaired *t* tests where *$P < 0.05$; **$P < 0.01$. Boxplot whiskers represent the maximum and minimum data values. Center lines show the medians; box limits indicate the 25th and 75th percentiles as determined by R software and represents the middle 50% of observed values. (C) Quantification of the differentiation index (percentage of nuclei in Myosin Heavy Chain-positive cells). (D) Quantification of the fusion index (percentage of nuclei inside myotubes with more than 2 nuclei) $P = 0.046$. (E) Quantification of the average number of myonuclei by myotubes $P = 0.001$. (F) Distribution of myotubes classified according to their nuclei content. For "3–5-nuclei" category, $P = 0.013$ and for "more than 10-nuclei" category, $P = 0.004$.

which we observed clustered myonuclei areas (Fig. 3D,E, zoom). In control condition, a limited accumulation of clustered myonuclei along myotubes length is observed, with a majority of clusters (80%) containing 4–6 myonuclei (Fig. 3H, Scramble shRNA condition). On the opposite, in *Sh3kbp1* knockdown, we observe a significant decrease in this cluster category that fall to 45% concomitantly with an increase in the proportion of clusters with more than 15 myonuclei (Fig. 3H). To address the specificity of myonuclei clustering phenotype to *Sh3kbp1*-depleted conditions, we expressed human full-length SH3KBP1 proteins in *Sh3kbp1*-depleted murine myoblasts and induced cells to differentiate and fuse into myotubes (Fig. 3F–H). Our results show that the formed myotubes appeared smaller in term of elongation and myonuclei number, and that the number of myonuclei clusters with more than ten myonuclei was decreased in myotubes re-expressing SH3KBP1 (Fig. 3F–H), suggesting that SH3KBP1 is also a key factor in the myonuclei repartition process. Together, these results show that *Sh3kbp1* is important to control myoblasts fusion and to monitor myonuclear spreading in the early steps of skeletal myotube formation.

## SH3KBP1 governs myonuclei motion that contributes to myonuclei positioning in mature myofibers

Myonuclei are dynamic organelles, particularly in developing myofibers, whose motion contributes to their correct spreading (Ghasemizadeh et al, 2021; Gache et al, 2017). To investigate the role of SH3KBP1 on myonuclei motion and its relative implication in myonuclei clustering, we tracked myonuclei displacement in mature in vitro primary muscle fibers after 5 days of differentiation (Fig. 4). Primary murine myoblasts were co-transfected with RFP-lamin-chromobody® to visualize myonuclei concomitantly with GFP-tagged-scramble shRNA or -*Sh3kbp1* shRNA (Fig. 4A–F). Myofibers containing both constructs (GFP and RFP-lamin-chromobody®) were then selected for the analyses. First, validating our previous results on myonuclei spreading, we observed 2-times more myonuclei clustering in myotubes treated with *sh3kbp1* shRNA compared to control condition (Fig. 4B–F, white asterisk). Myonuclei were then tracked every 20 min for a time period of 16 h (Fig. 4G; Movies EV1 and EV2). In control condition,

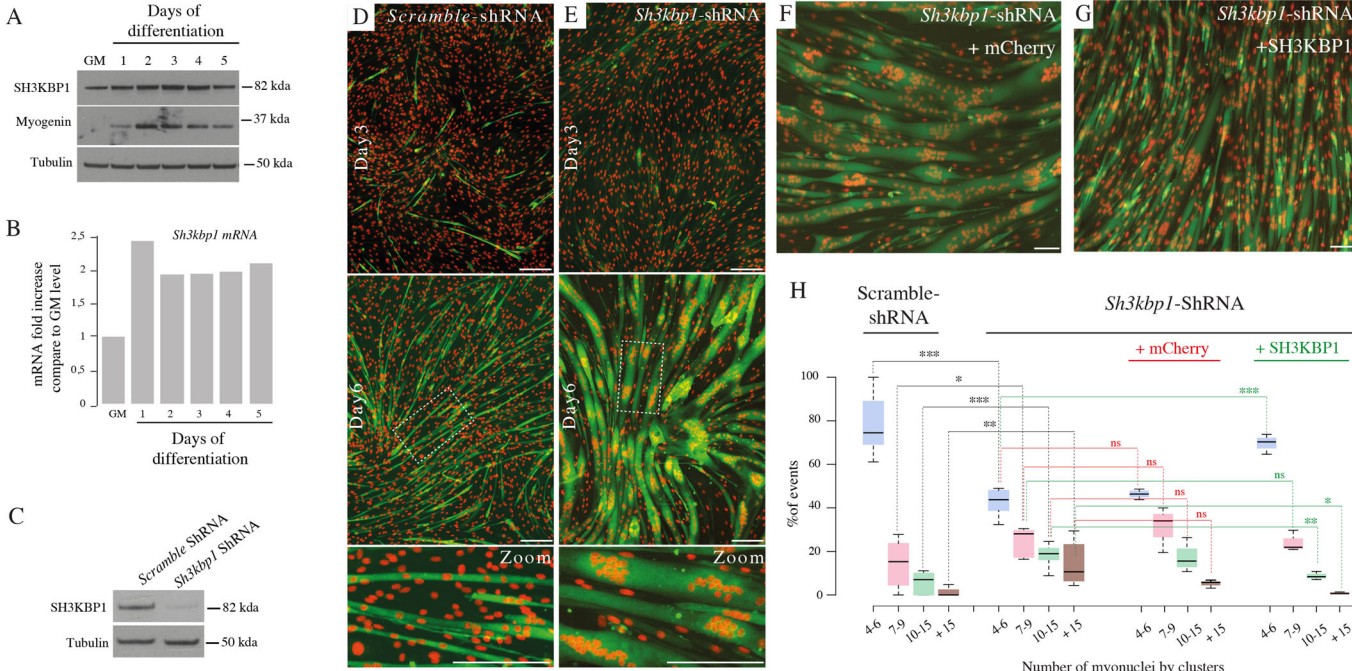

**Figure 3. SH3KBP1 controls myonuclei clustering in C2C12 myotubes.**

(A) Western blot analysis of SH3KBP1 protein levels in total protein extracts obtained from growing (GM) or differentiating C2C12 cells (from 1 to 5 days). C2C12 differentiation is assessed by Myogenin expression and Tubulin is used as loading control. (B) RT-qPCR analysis of *Sh3kbp1* gene expression level relative to housekeeping genes (*CycloB, Gapdh, GusB, Rpl41,* and *Tbp*) in proliferating C2C12 cells (GM) or in differentiating C2C12 (1 to 5 days of differentiation). (C) Representative western blots analysis of SH3KBP1 protein downregulation in stable cell line constitutively expressing shRNA construct targeting *Sh3kbp1*. Tubulin is used as loading control. (D, E) Representative immunofluorescence staining of myosin heavy chain (green) and myonuclei (red) in C2C12 stable cell lines expressing scramble shRNA (D) or shRNA targeting *Sh3kbp1* (E) after 3 or 6 days of differentiation. Zooms are magnifications of images in white dots rectangles in 6 days old myotubes. Scale bars: 150 μm, 15 μm in zoom. (F, G) Representatives immunofluorescent staining of myosin heavy chain (green) and myonuclei (red) in stable cell line expressing *Sh3kbp1* shRNA, co-transfected with plasmid coding either for mCherry (F) or the full-length human SH3KBP1 (G) after 5 days of differentiation. Scale bar: 150 μm. (H) Quantification of the percentage of myonuclei per clusters observed in (D–G) experiments. Statistical analysis performed using unpaired *t* tests where *$P < 0.05$; **$P < 0.01$; ***$P < 0.001$. Comparison between shRNA Scramble and ShRNA-*sh3kbp1* ($n = 8$; biological replicates) for the "4–6" nuclei category $P = 0.000014$, for the "7–9" nuclei category $P = 0.039$, for the "10–15" nuclei category $P = 0.00016$ and for the "+ 15" nuclei category $P = 0.004$. Comparison between ShRNA-*sh3kbp1* and ShRNA-*sh3kbp1* + SH3KBP1 ($n = 3$; biological replicates) for the "4–6" nuclei category $P = 0.00008$, for the "10–15" nuclei category $P = 0.0076$ and for the "+ 15" nuclei category $P = 0.04$. Boxplot whiskers represent the maximum and minimum data values. Center lines show the medians; box limits indicate the 25th and 75th percentiles as determined by R software and represents the middle 50% of observed values.

myonuclei are in motion during nearly 35% of the time (a movement being defined as more than 30 μm displacement between two time points) at a median speed of 0.232 ± 0.014 μm/min. In *sh3kbp1*-depleted conditions, it was technically impossible to correctly follow individual myonucleus inside myonuclei cluster (Movie EV2). Yet, myonuclei outside clusters were easily trackable and displayed an increase of more than 20% of the percentage of time in motion and of more than 30% of the median speed, which reaches 0.313 ± 0.014 μm/min (Fig. 4H,I). As myonuclei movement and spreading in developing myotubes have been largely related to the microtubule network (Cadot et al, 2012; Metzger et al, 2012; Ghasemizadeh et al, 2021; Wilson and Holzbaur, 2012), we investigated the impact of *sh3kbp1* depletion on the microtubule network organization. Microtubule network was revealed by Sir-tubulin® stainings in 5 days of differentiated myotubes obtained from stable C2C12 cell lines expressing either a scramble or an shRNA against *sh3kbp1*. In these conditions, we did not observe a dramatic impairment in the global microtubule network architecture/orientation, with the exception of the myotubes zones containing aggregated myonuclei (Fig. EV2A). The quantification

of the microtubule bundles orientation in-between individual myonuclei showed that the "microtubules bundle orientation" was not impaired in the absence of SH3KBP1, with the exception of a small pool of microtubules bundles perpendicular to the length of myotubes (Fig. EV2B). As microtubule network emanates from the membrane of the nucleus in myotubes (Bugnard et al, 2005; Winje et al, 2018), we asked whether the recruitment of specific proteins able to drive/initiate the polymerization of microtubules at the nuclear membrane such as Pericentrin or PCM1 was affected (Fig. EV2C–F). This approach shows that the re-localization of these proteins at the membrane of myonuclei was not affected by the absence of SH3KBP1. Finally, as we observed a small increase of the number of perpendicular microtubules (with respect to the longitudinal axis of myotubes) and that *Havrylov* et al previously identified the MTs-binding-proteins MAP7 as a potential interactor of SH3KBP1 protein, we performed immunoprecipitation assay using various constructs of MAP7 to evaluate the MAP7-SH3KBP1 interaction (Fig. EV2G–I). Various GFP-tagged-MAP7 constructs were expressed in C2C12 cells and immunoprecipitated using GFP as a trap. However, we failed to identify an interaction

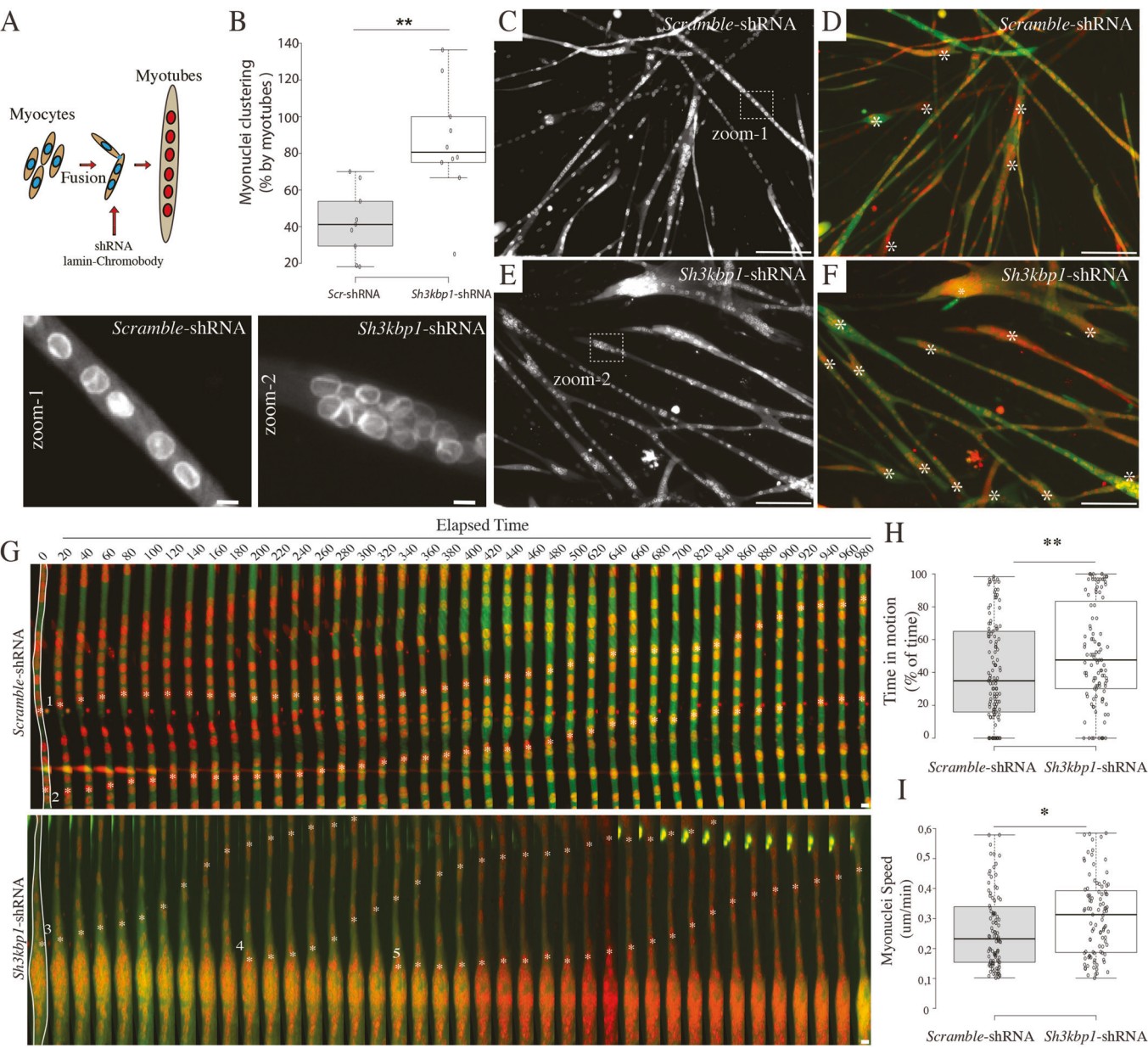

**Figure 4. Myonuclei motion is SH3KBP1-dependent in myotubes.**

(A) Scheme of the sequential steps performed to obtain murine primary myofibers labeled with red-myonuclei in *Sh3kbp1* knockdown conditions. Plasmids coding for scramble or *sh3kbp1* shRNA and lamin-chromobody were co-transfected during myoblasts fusion step. (B) Quantification of myonuclei clustering in primary murine myotubes transfected with the aforementioned plasmids ($n = 3$; biological replicates). Three independent replicates were performed for each experiment. Statistical analysis performed using unpaired $t$ tests where **$P < 0.01$. $P = 0.0019$. (C–F) Representative immunofluorescence staining of primary myoblasts co-transfected with either scramble or *Sh3kbp1* shRNA tagged with GFP (green) and with RFP-lamin-chromobody® (red) and induced to differentiate for 5 days into myofibers. (C, D) RFP, (D–F) merge. Scale bar: 150 μm. Zooms are magnifications of images in white dots rectangles in (C, E), respectively. (G) Kymograph showing frames extracted from a 16 h time-lapse movie in two channels (shRNA in green and lamin-chromobody® in red) of primary myotubes. In the first frame (time 0), myofibers are outlined in white, which corresponds to the region used to create the adjacent kymograph. Scale bar: 10 μm. (Asterisks 1–5 are examples of individual myonuclei tracking. (H) Quantification of myonuclei time in motion. $P = 0.006$. (I) Quantification of myonuclei speed $P = 0.039$. (H, I) ($n = 3$; biological replicates and $n > 30$ myonuclei per group, per repeat) Statistical analysis performed using unpaired $t$ tests where *$P < 0.05$; **$P < 0.01$. (B, H, I) Boxplot whiskers represent the maximum and minimum data values. Center lines show the medians; box limits indicate the 25th and 75th percentiles as determined by R software and represents the middle 50% of observed values.

with MAP7 in our conditions (Fig. EV2I). All together, these results suggested that the expression of SH3KBP1 during muscle fibers formation and maturation contribute to slow down myonuclei movements (time in motion and speed), which could contribute to a relative myonuclei motion stability during myotube formation, that in turn will facilitate myonuclei spreading, independently from any microtubule network organization impairment.

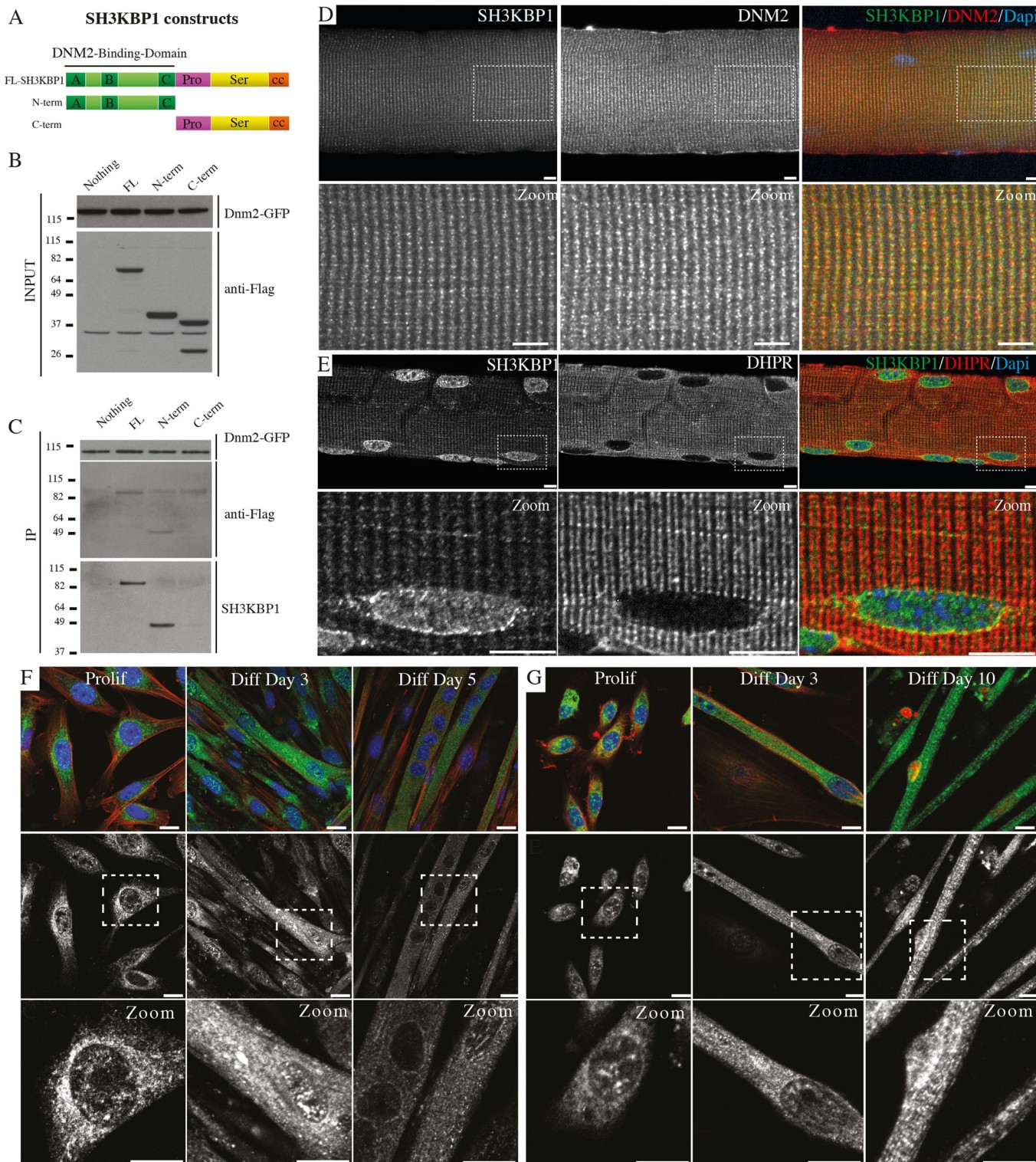

## SH3KBP1 localizes at the Z-line in the I-band where it interacts with DNM2

Several studies showed that SH3KBP1 is involved in multiple cellular processes including vesicle-mediated transport, membrane and cytoskeleton remodeling (Havrylov et al, 2010). The three SH3 domains localized in the N-terminal part of SH3KBP1 (Fig. 5A) are responsible for its high capacity to interact with diverse regulatory partners (Havrylov et al, 2009) while the C-terminus part, containing a coiled-coil domain, allows its targeting to endosomal membranes (Zhang et al, 2009). Among interacting proteins, Dynamin-2 (DNM2), a large GTPase implicated in cytoskeleton regulation and endocytosis,

**Figure 5. SH3KBP1 interacts with DNM2 and accumulates at the I-band in mature skeletal muscle fibers.**

(A) SH3KBP1 constructs used in the experiment. (B) Representative western blot of crude extracts of C2C12 cells co-expressing GFP-DNM2 and various Flag-SH3KBP1 constructs (FL full length, N-term N-terminal part of SH3KBP1, C-term C-terminal part of SH3KBP1) and stained with anti-GFP (top) or anti-Flag (bottom) antibodies. (C) Representative western blot after DNM2-GFP immunoprecipitation using GFP-Trap and aforementioned C2C12 cell extracts (B). The membrane was revealed with anti-GFP (top), anti-Flag (middle) and anti-SH3KBP1 (bottom) antibodies. (B, C) Blots were repeated more than three times. (D) Representative images of extracted *Tibialis Anterior* muscle fiber stained for SH3KBP1 (green), DNM2 (red) and myonuclei (blue) (single Z plan). Scale bars, 10 µm. (E) Representative images of extracted *Tibialis Anterior* muscle fiber stained for SH3KBP1 (green), DHPR1α (for DyHydroPyridine Receptor alpha, red) and myonuclei (blue) (Max intensity of Z stacks plans). Scale bars, 10 µm. (F) Representative immunofluorescent staining of SH3KBP1 (green), Actin (red) and myonuclei (blue) in the time course of C2C12 cells differentiation: proliferation (Prolif) and 3 or 5 days of differentiation (diff day 3, diff day 5) are presented. (G) Representative immunofluorescent staining of SH3KBP1 (green), Actin (red) and myonuclei (blue or red) along the time course of primary myoblasts cells differentiation: proliferation (Prolif) and 3 or 10 days of differentiation (diff day 3, diff day 10) are presented. Scale bars, 10 µm.

was previously described in HeLa cells as a SH3KBP1 interacting protein, through its proline-rich domain (Schroeder et al, 2010). To check if SH3KBP1 and DNM2 interact in muscle cells, we conducted a co-immunoprecipitation assay (Fig. 5B,C). Full-length GFP-tagged-DNM2 was co-expressed with FLAG-tagged SH3KBP1 fragments in C2C12 cells, and DNM2-GFP was immunoprecipitated using GFP as a trap. This experiment confirmed that DNM2 interacts with SH3KBP1 through its N-terminal sequence, containing the SH3 domains of SH3KBP1 (Fig. 5C). We conducted the same co-expression assay in 10 days in vitro primary myofibers and showed by immunofluorescence that DNM2 over-expression leads to the formation of small punctums along the length of myofibers, highly positive for SH3KBP1, confirming the association between these two proteins in myofibers (Fig. EV1J). DNM2 is described as a component of the I-band in mature muscle fibers (Durieux et al, 2010) and more specifically localized at the Z-line (Neves et al, 2023). As expected, SH3KBP1 staining is present as transversal punctums, co-localizing in part with DNM2 staining (Fig. 5D) and localized in-between the transversal DHPR staining (Fig. 5E), validating SH3KBP1 as a component of the Z-line at the I-band zone. Interestingly, SH3KBP1 staining also strongly accumulated at the vicinity of myonuclei, forming a "cage" around myonuclei (Fig. 5E). Moreover, SH3KBP1 localization related to myonuclei was confirmed using cross section of *Tibialis Anterior* mice muscle, highlighting that this localization seems specific to muscle cells (Fig. EV3A).

We next addressed SH3KBP1 subcellular localization during the time course of myotube formation using C2C12 myoblasts cell line and of myofiber formation using primary murine myoblasts (Fig. 5F,G). In proliferative conditions, SH3KBP1 is dispersed throughout the cytoplasmic compartment with an apparent higher concentration at the vicinity of myonuclei (Fig. 5F,G, Prolif). In myotubes, SH3KBP1 seems to spread along myotubes length and exhibits stronger labeling at the perinuclear zone (Fig. 5F,G, C2C12 day 3 and 5; primary diff day 3). Finally, in "mature-like" myofibers obtained using primary murine myoblasts, SH3KBP1 is still strongly accumulated at myonuclei vicinity and also exhibits longitudinal/transversal staining (Fig. 5G, diff day 10). Myonuclei-related localization of SH3KBP1 was also confirmed using staining of cross-sections of human muscle biopsies (Fig. EV3B,C). These results show that SH3KBP1 binds to DNM2 in skeletal muscle fibers and localizes at the Z-line in the I-band zone and also at the vicinity of myonuclei.

## SH3KBP1 is required for T-tubules formation and muscle excitability

T-tubules control the correct functioning of the Excitation-Contraction Coupling (ECC) in skeletal muscle (Allard, 2018). Of

interest, DNM2-dependent membrane remodeling is required for T-tubules structure, as DNM2 mutants lead to their fragmentation (Chin et al, 2015). In this view, we next assessed the impact of *Sh3kbp1* depletion on T-tubule formation. In vitro maturation of primary myofibers harboring mature sarcomeric and T-tubule structures were used to address this question (Fig. 6A–D). As reflected by the actin staining, myofibrillogenesis in *Sh3kbp1* depletion condition was normal after 10 days of differentiation (Fig. 6B,C,E). In control condition, nearly 30% of myofibers exhibited transversal DHPRα doublet bands staining alternatively with actin structures, reflecting maturation of T-tubules structures (Fig. 6B,C). In *Sh3kbp1*-depleted myofibers, DHPRα staining appeared as random dots, reflecting a delay or an absence of maturation of T-tubules, that also failed to correctly align with sarcomere structures (Figs. 6C,D and EV4A,B).

In skeletal muscle fibers, functional triads are structures that include Transverse tubules (T-tubules) and sarcoplasmic reticulum (SR) cisternae to allow excitation-contraction coupling (ECC). As T-tubules appeared severely affected in *Sh3kbp1*-depleted in vitro myofibers, we asked on the direct modification of the ECC efficiency after *Sh3kbp1* depletion in ex-vivo skeletal muscle fibers. To test this possibility, we compared voltage-activated SR $Ca^{2+}$ release in mouse *Flexor Digitalis Brevis* (FDB) muscle fibers depleted for *Sh3kbp1* (Fig. 6F–H). Downregulation of *Sh3kbp1* was achieved using intramuscular FDB muscle injections of an AAV cognate vector expressing shRNA targeting *Sh3kbp1* mRNA (AAV-shSh3kbp1) or with PBS for the control. DHPRα staining showed a doublet in control conditions whereas in *Sh3kbp1*-depleted conditions, DHPRα staining appeared as a single line (Fig. 6E). In this view, the average distance between DHPRα striations was significantly decreased in *Sh3kbp1*-depleted conditions compared to control conditions, suggesting a role of SH3KBP1 in the T-tubule architecture maintenance (Fig. EV4C). As we show that Sh3kbp1 is involved in myonuclear spreading (Figs. 1–4), we addressed in FDB muscle if, in *Sh3kbp1*-depleted myofibers, myonuclear spreading was affected. No significant changes were observed regarding the mean distance between nearest myonuclei in FDB muscle fibers, nevertheless, nearly 15% of myonuclei were present in myonuclei clusters in *Sh3kbp1*-depleted conditions compared to control ones, suggesting a myonuclei heterogeneity regarding the SH3KBP1-dependent control of myonuclear motion in mature myofibers (Fig. EV4D).

We next tested if SR $Ca^{2+}$ release amplitude and kinetics were affected in *Sh3kbp1*-depleted muscles fibers. Figure 6F shows rhod-2 $Ca^{2+}$ transients elicited by membrane depolarizing steps of increasing amplitude in *Sh3kbp1*-depleted and in control myofibers. Transients in control fibers exhibited a fast early rising phase upon

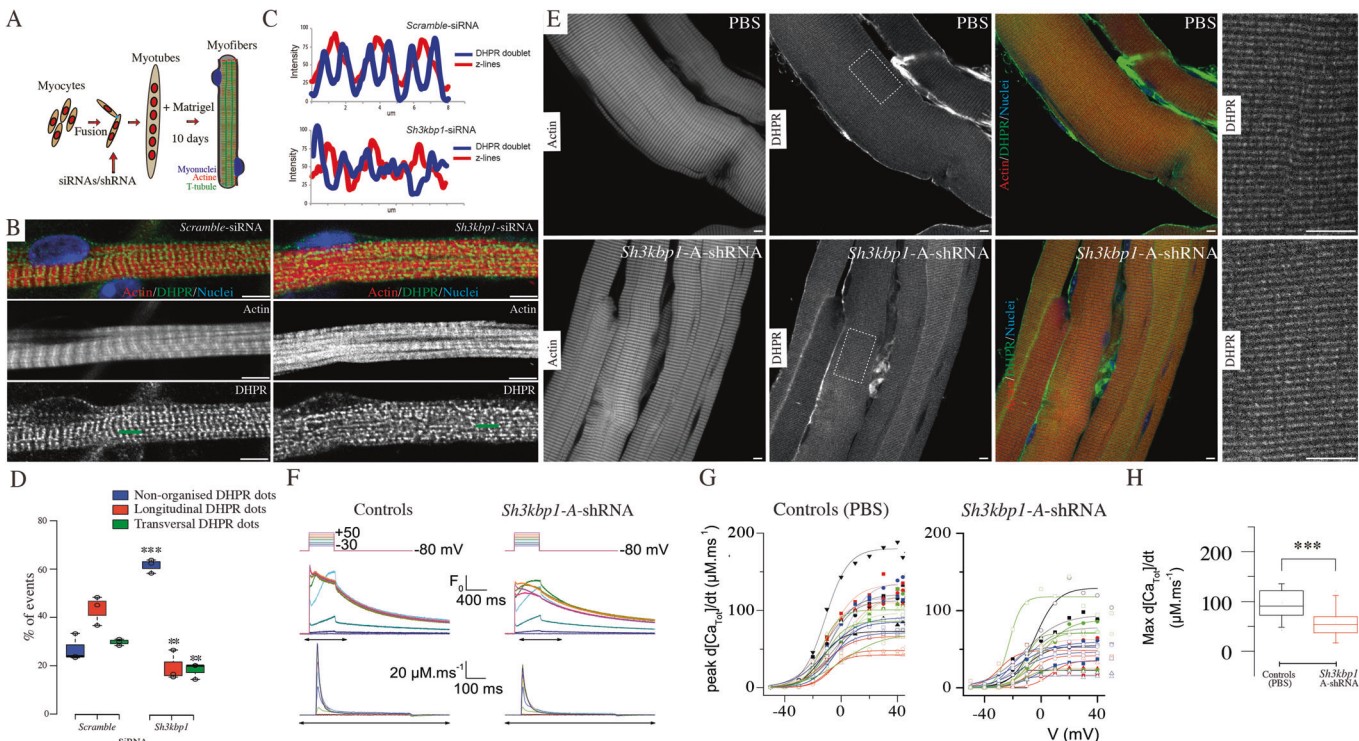

**Figure 6. SH3KBP1 alters T-tubule formation and the excitation–contraction coupling (ECC) in mature myofibers.**

(A) Sequential steps performed to obtain in vitro "mature" primary myofibers. (B) Representative immunofluorescent image of DHPR1-α (green), Actin (red) and myonuclei (blue) staining in 10 days cultured primary myotubes after scramble siRNA or *Sh3kbp1* siRNA transfection. Scale bars, 5 μm. (C) Line scan of the green line depicted in (B) to visualize the organization of transversal triad doublets, according to DHPR-α staining (blue) and of Z-line, according to actin staining (red). (D) Quantification of DHPR1-α aspects in mature myofibers treated with either scramble siRNA or a pool of 2 individual siRNAs targeting *Sh3kbp1* mRNA (n = 3; biological replicates) for the "non-organized DHPR dots" category P = 0.00066, for the "longitudinal DHPR dots" category P = 0.0089 and for the "transversal DHPR dots" category P = 0.0055. Statistical analysis performed using unpaired *t* tests where ***P < 0.001, **P < 0.01. Details for different categories are available in Fig. EV4A. (E) Representative images of extracted myofibers of *Tibialis Anterior* muscle electroporated with either PBS or AAV-shRNAs targeting *Sh3kbp1* and stained for Actin (red), DHPR-α (green) and nucleus (blue). Scale bars, 20 μm. (F) Representative rhod-2 Ca²⁺ transients in FDB fibers injected with either PBS or AAV cognate vector expressing shRNA targeting SH3KBP1 mRNA (Sh3kbp1 shRNA) for 3 months, in response to 0.5-s-long depolarizing pulses from −80 mV to the range of indicated values, with intervals of 10 mV. (Bottom) Corresponding Ca²⁺ release flux (d[CaTot]/dt) traces calculated as described in the Materials and methods. (G) Mean voltage-dependence of the peak rate of SR Ca²⁺ release in control and in *Sh3kbp1* knockdown fibers. (H) Inset shows the mean values for maximal rate for SR Ca²⁺ release in the two groups of fibers, as assessed from Boltzmann fits to data from each fiber. Data are pooled from four independent repeats (n > 21 cells by condition). P = 0.00009. Statistical analysis performed using unpaired *t* tests where ***P < 0.001. (D, H) Boxplot whiskers represent the maximum and minimum data values. Center lines show the medians; box limits indicate the 25th and 75th percentiles as determined by R software and represents the middle 50% of observed values.

depolarization followed by a slower phase at low and intermediate voltages and by a slowly decaying phase for the largest depolarizing steps (Kutchukian et al, 2017). Rhod-2 transients from *Sh3kbp1*-depleted fibers exhibited an overall similar time course, not distinguishable from control muscle fibers (Fig. 6F). In each tested fiber, the rate of SR Ca²⁺ release was calculated from the rhod-2 Ca²⁺ transients. Traces for the calculated rate of SR Ca²⁺ release corresponding to the transients shown in Fig. 6F are shown at the bottom of Fig. 6F. In both control and *Sh3kbp1*-depleted fibers, the rate exhibited an early peak, whose amplitude increased with that of the pulse, followed by a spontaneous decay down to a low level. The voltage-dependency of the peak rate of SR Ca²⁺ release is shown in Fig. 6G for both groups of fibers. Fitting the relationship in each fiber with a Boltzmann function showed that the mean value for maximal rate of SR Ca²⁺ release (Max d[Catot]/dt) was significantly reduced in *Sh3kbp1*-depleted fibers while mid-activation voltage (V₀.₅) and slope factor (k) of the voltage-dependency were statistically unchanged (Figs. 6H and EV4E,F). The reduction of

only the maximum rate of Ca2+ release indicates that the voltage sensing properties of the process, carried by the DHPR, are unaffected by SH3KBP1 depletion, and that the SR Ca2+ release channels activation and inactivation properties during voltage activation are also maintained. Thus, triadic disruption is most likely reducing the density of activatable Ca2+ release channels, resulting in a reduction of maximum Ca2+ release with unaffected voltage-dependent and kinetic properties. Overall, these results suggest that SH3KBP1 is required for proper ECC process through triads integrity maintenance.

### *Sh3kbp1* is upregulated in a murine model of AD-CNM and contributes to CNM phenotypes

Due to SH3KBP1 interaction with DNM2, we next thought to study *sh3kbp1* in pathological context associated with DNM2 dysfunction. Heterozygous expression of the most frequent human mutation causing the dominant centronuclear myopathy in a

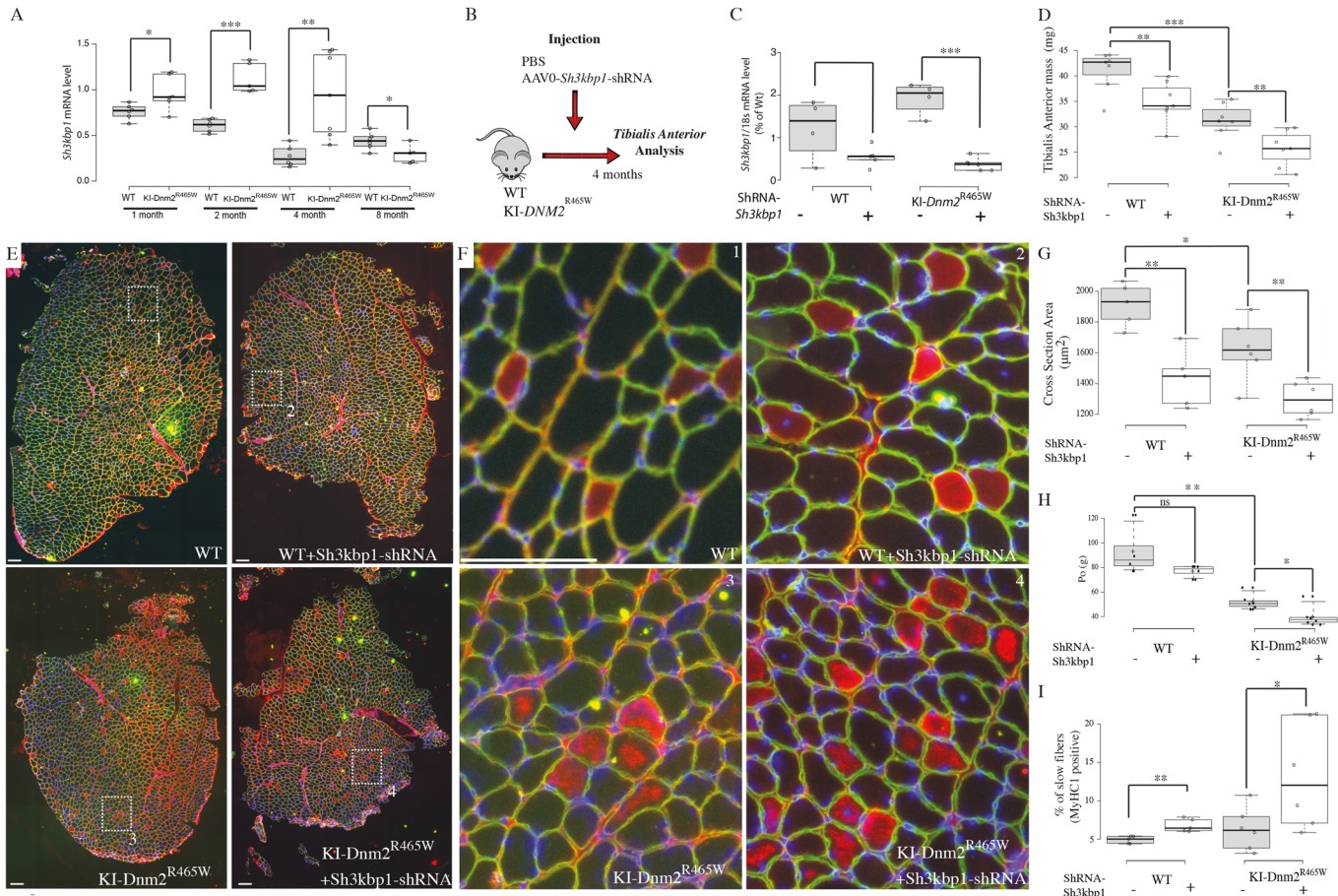

**Figure 7. *Sh3kbp1* silencing is related to an "atrophic phenotype" in WT or in KI-*Dnm2*^R465W/+ mice.**

(A) RT-qPCR analysis of *Sh3kbp1* gene expression relative to *Nat10* gene in *Tibialis Anterior* muscles from WT or KI-*Dnm2*^R465W/+ mice at 1, 2, 4, and 8 months of age. Number of mice per group $n > 5$; comparison between WT and KI-DNM2^R465W at 1 month $P = 0.035$, at 2-month $P = 0.00006$, at 4-month $P = 0.0096$ and at 8-month $P = 0.037$. Statistical analysis performed using unpaired $t$ tests where ***$P < 0.001$, **$P < 0.01$ and *$P < 0.05$. (B) Sequential steps performed to obtain *Sh3kbp1* knockdown fibers. (C) RT-qPCR analysis of *Sh3kbp1* gene expression relative to *Nat10* gene in 4 months *Tibialis Anterior* muscles from WT mice ($n = 5$; biological replicates) or KI-*Dnm2*^R465W/+ mice ($n > 4$; biological replicates, $P = 0.000014$) injected with either PBS or AAV cognate vector expressing shRNA targeting *SH3KBP1* mRNA. Statistical analysis performed using unpaired $t$ tests where ***$P < 0.001$. (D) Quantification of *Tibialis Anterior* muscle mass from WT or KI-*Dnm2*^R465W/+ mice injected with either PBS or shRNA targeting *sh3kbp1* mRNA. ($n > 7$; biological replicates). Comparison between WT and WT-ShRNA-*sh3kbp1* $P = 0.004$, WT and KI-DNM2^R465W $P = 0.00006$, and KI-DNM2^R465W-ShRNA-*sh3kbp1* $P = 0.006$. Statistical analysis performed using unpaired $t$ tests where ***$P < 0.001$, **$P < 0.01$. (E, F) Representative images of *Tibialis Anterior* muscle cross-sections from WT or KI-*Dnm2*^R465W/+ mice injected with either PBS or shRNA targeting *sh3kbp1* mRNA and stained for nucleus (DAPI, blue), Laminin (green) and MyHC1 (red). Scale bars = 150 µm. (F) Zooms of the white dots rectangles defined in (E) images. Scale bars = 150 µm. (G) Quantification of the mean cross-sectioned myofibers areas of *Tibialis Anterior* muscles from WT or KI-*Dnm2*^R465W/+ injected with either PBS or shRNA targeting *SH3KBP1* mRNA. ($n > 5$; biological replicates). Comparison between WT and WT-ShRNA-*sh3kbp1* $P = 0.0015$, WT and KI-DNM2^R465W $P = 0.02$, and KI-DNM2^R465W-ShRNA-*sh3kbp1* $P = 0.005$. Statistical analysis performed using unpaired $t$ tests where **$P < 0.01$ and *$P < 0.05$. (H) Quantification of Absolute force P0 (g) of *Tibialis Anterior* muscles from WT or KI-*Dnm2*^R465W/+ injected with either PBS or shRNA targeting *SH3KBP1* mRNA. ($n > 7$; biological replicates). Comparison between WT and KI-DNM2^R465W $P = 0.0019$, KI-DNM2^R465W and KI-DNM2^R465W-ShRNA-*sh3kbp1* $P = 0.011$. Statistical analysis performed using unpaired $t$ tests where **$P < 0.01$ and *$P < 0.05$. (I) Quantification of the percentage of slow fibers (MyHC1-positives) of *Tibialis Anterior* muscles from WT or KI-*Dnm2*^R465W/+ injected with either PBS or shRNA targeting *SH3KBP1* mRNA ($n > 5$; biological replicates). Comparison between WT and WT-ShRNA-*sh3kbp1* $P = 0.002$, KI-DNM2^R465W and KI-DNM2^R465W-ShRNA-*sh3kbp1* $P = 0.04$. Statistical analysis performed using unpaired $t$ tests where **$P < 0.01$ and *$P < 0.05$. (A, C, D, G–I) Boxplot whiskers represent the maximum and minimum data values. Center lines show the medians; box limits indicate the 25th and 75th percentiles as determined by R software and represents the middle 50% of observed values.

mouse model (KI-*Dnm2*^R465W/+) results in progressive muscle atrophy, impairment of contractile properties, histopathological abnormalities including slight disorganization of T-tubules and reticulum, and elevated cytosolic calcium concentration (Durieux et al, 2010). We first addressed *Sh3kbp1* mRNA expression during mice development and aging. *Sh3kbp1* mRNA was progressively reduced by more than 50% in 4-month-old mice as compared to 1-month-old mice (Fig. 7A), suggesting a downward tendency for

SH3KBP1 expression along muscle aging. We found that in the KI-*Dnm2*^R465W/+ model, *Sh3kbp1* mRNA was significantly upregulated in mature myofibers, during the first 4 months of mice development to then reach a level significantly lower than in control conditions in 8 months old mice (Fig. 7A).

To address the role of SH3KBP1 in the long-term muscle homeostasis, we investigated in vivo depletion of SH3KBP1 protein in both wild-type and KI-*Dnm2*^R465W/+ mice. Downregulation of

*Sh3kbp1* was achieved using intramuscular TA muscle injections of an AAV cognate vector expressing shRNA targeting *Sh3kbp1* mRNA (AAV-shSh3kbp1) both in wild-type and KI-*Dnm2*^R465W/+ mice at 5 weeks of age, an age where muscle mass is nearly fully developed (Fig. 7B). In wild-type mice, *Sh3kbp1* mRNA level was decreased by 2.7-fold compared to PBS-injected muscles while in KI-*Dnm2*^R465W/+ mice, expression level showed a 5.4-fold decrease compared to PBS-injected TA muscles, 3 months after injection (Fig. 7C). No significant change of body weight was observed between AAV-shSh3kbp1-injected and PBS-injected conditions in both genotypes (Fig. EV5A). However, we observed a significant decrease of TA muscle mass of about 20%, when *Sh3kbp1* is depleted, in both WT and KI-*Dnm2*^R465W/+ mice (Fig. 7D) revealing a role of SH3KBP1 in muscle fibers homeostasis maintenance (Fig. EV5B). Transverse sections of TA muscles were performed to determine muscle fibers cross-sectional areas (Fig. 7E–G). A strong decrease in the median myofibers area was observed in *Sh3kbp1*-depleted conditions, when compared both to control muscles in WT (-30%) or KI-*Dnm2*^R465W/+ (−20%) mice (Fig. 7G). Analysis of muscle fibers area repartition showed a significant increase in atrophic fibers in AAV-shSh3kbp1 injected muscles (Fig. EV5C). In addition, we observed a significant reduction of about 25% in the total number of fibers specifically in KI-*Dnm2*^R465W/+ model depleted for *Sh3kbp1* (Fig. EV5D). As expected, we observed a significant decrease in the absolute force (g) developed by TA muscles specifically in KI-*Dnm2*^R465W/+ mice depleted for *Sh3kbp1*, suggesting a more drastic impact of *Sh3kbp1* downregulation when DNM2 activity is altered (Fig. 7H).

CNM patients with the p.R464W mutation exhibit centralized myonuclei and a predominance of type 1 muscle fibers (Romero and Bitoun, 2011). Our in vitro data showed that *Sh3kbp1* depletion increases fusion and alters myonuclei spreading (Figs. 1–4). No significant changes in internalized myonuclei in *Sh3kbp1*-depleted conditions in both WT and KI-*Dnm2*^R465W/+ mice were noticed (Fig. EV5AE), showing that SH3KBP1 involvement in myonuclei spreading occurs mainly during the early phases of myotubes formation. Interestingly, we observed a myonuclear accretion tendency in *Sh3kbp1*-depleted conditions (Fig. EV5F), suggesting a role of SH3KBP1 on the regulation of fusion events. Finally, we investigated the ratio of slow fiber types in AAV-shSh3kbp1 injected TA muscles, reflected by the expression of myosin heavy chain type 1. In wild-type conditions, slow fiber type accounted for only 5% of total muscle fibers, whereas in AAV-sh-Sh3kbp1 injected muscles, this ratio was slightly increased to reach 6.4% of total muscle fibers, which is the same as in KI-*Dnm2*^R465W/+ mice. However, when *Sh3kbp1* was depleted in KI-*Dnm2*^R465W muscle, this ratio increased drastically to reach 12% of the total fibers (Fig. 7I). These experiments show that SH3KBP1 is a key factor in the maintenance of muscle fibers homeostasis and particularly in the pathological context of CNM disease induced by DNM2 mutation.

## SH3KBP1 is an endoplasmic reticulum scaffolding protein that supports the autophagy pathway

As SH3KBP1 harbors a strong accumulation at the vicinity of myonuclei, we thus wondered what additional compartment could be modulated by SH3KBP1. To answer this question, we first used stable C2C12 cell line depleted for *Sh3kbp1* (Fig. 3C) to investigate the scaffolding of the Golgi Apparatus (GA) and the Endoplasmic

Reticulum (ER) (Fig. 8A,B). In both control- and *Sh3kbp1*-depleted-myotubes, the GA, labeled with RCAS1 antibody is mainly localized around myonuclei and dispersed in the cytoplasm, as previously described (Fig. 8A) (Ralston, 1993). On the opposite, the ER, labeled with ERp72 antibody, that is highly concentrated at the perinuclear zone in control myotubes failed to remain accumulated at the vicinity of myonuclear membrane and was dispersed in the cytosol in *Sh3kbp1*-depleted myotubes (Fig. 8B). This result suggests a supportive role of SH3KBP1 in the conservation of the ER at the vicinity of myonuclei, independently of the GA. To assess if SH3KBP1 is an ER-associated protein, we conducted co-immunoprecipitation experiments using SH3KBP1 deletion mutants tagged with GFP construct (Fig. 8C–E). This approach showed that the full-length SH3KBP1 protein was able to coimmunoprecipitate with both ERP72 and Calnexin and that this interaction is mediated by the C-terminal part of SH3KBP1 (Fig. 8C–E). ERP72 is a disulfide isomerase that acts as a folding chaperone in the intraluminal ER compartment and that interacts with Calnexin, an ER- transmembrane chaperone, which exhibits a cytoplasmic C-terminus (Satoh et al, 2005; Penga et al, 2014). Our results point to the "Proline-and Serine-Rich domain" of SH3KBP1 as the main sequence mediating the interaction with ERP72-Calnexin containing ER, as the coiled-coiled domain in the C-terminus part of SH3KBP1 failed to interact with both proteins (Fig. 8E). ER distribution in mature skeletal muscle fiber is mainly found around myonuclei and forms striated pattern at the I-band/Z-line zone (Ralston, 1993; Zhang et al, 2021). In *Tibialis Anterior*, SH3KBP1 also exhibited an accumulation at the vicinity of myonuclei and a striated pattern at the Z-line, in-between the staining of the voltage-dependent calcium channel, DHPRα, thanks to its interaction with DNM2 (Fig. 5). We confirmed this particular transversal patterning in vitro in "mature-like" myofibers over-expressing either full-length, N-terminal or C-terminal part of SH3KBP1 (Fig. 8F). As expected, in isolated myofibers, full-length SH3KBP1 was present as small aggregation patches close to myonuclei, combined with striated pattern and was also accumulated at the I-band with no overlap with actin staining (Figs. 8F, SH3KBP1-FL and 5F). The N-terminal fragment of SH3KBP1 containing SH3 domains shown to interact with DNM2, was only present as striated patterns with no particular accumulation at myonuclei vicinity or at the periphery of myofibers. On the opposite, the C-terminal SH3KBP1 fragment was strongly accumulated at the periphery of muscle fibers, at the vicinity of myonuclei and at the I-band with striated patterns (Fig. 8F, SH3KBP1-Cter) suggesting a bridge-like role of SH3KBP1 between the ER and proteins localized at the I-band in skeletal muscle fibers.

Since perinuclear ER architecture is altered in *Sh3kbp1*-depleted myotubes, we asked whether it could affect the autophagy, as ER is an important organelle for the functioning of the autophagic process (Mochida and Nakatogawa, 2022). Autophagy is characterized by the formation of double-membrane vesicles called autophagosomes that engulf parts of the cytoplasm/organelles and fuse with acidic lysosome to form autolysosomes whose content will be degraded. Autophagy is an important actor of proteostasis and muscle homeostasis (Xia et al, 2021). To decipher the involvement of SH3KBP1 in the autophagy regulation, we quantified LC3 protein (Microtubule-associated protein 1A/1B-light chain 3), commonly used to address the autophagic activity due to its conjugation to the phosphatidylethanolamines of the

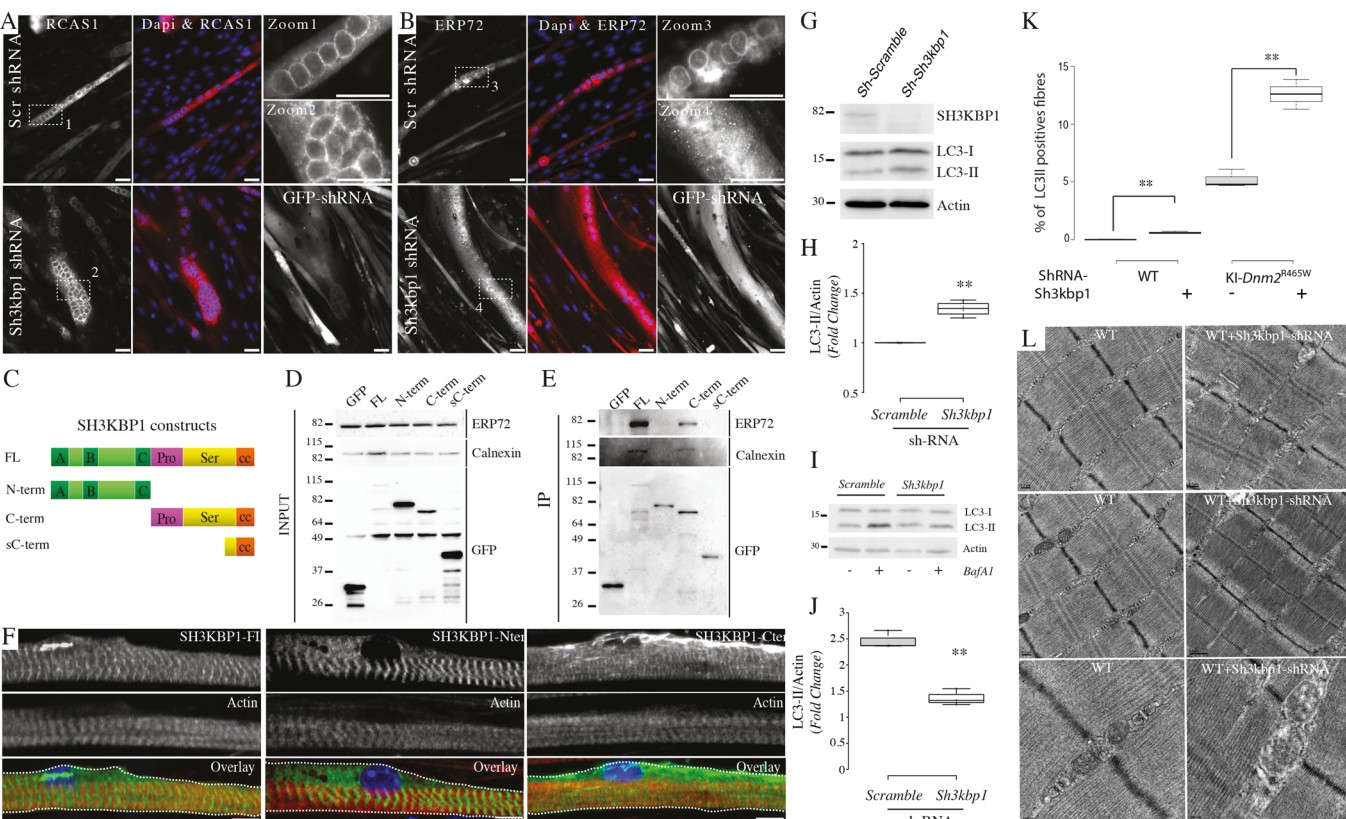

**Figure 8. SH3KBP1 is an endoplasmic reticulum scaffolding protein and supports the autophagic pathway in skeletal muscle fibers.**

(A, B) Representative Immunofluorescent staining of Golgi (RCAS1, red) (A) or Endoplasmic Reticulum (ERP72, red) (B) and myonuclei (DAPI, blue) in 6 days differentiated C2C12 cells expressing either scramble-GFP-shRNA or GFP-shRNA targeting *Sh3kbp1* gene. Scale bars, 100 μm. Zooms 1–4 are magnifications of the images in white dots. Scale bars: 100 μm. (C) GFP-tagged SH3KBP1 constructs used in the experiment. (D) Representative western blot performed on crude extracts of C2C12 cells expressing GFP-SH3KBP1 constructs and stained with anti-ERP72 (Top), anti-Calnexin (middle) and anti-GFP (bottom) antibodies. (E) Representative western blot performed after SH3KBP1-GFP construct immunoprecipitation (GFP trap assay) of C2C12 cells extracts (D) and stained with anti-ERP72 (top), anti-Calnexin (middle) and anti-GFP (bottom) antibodies. Blots were repeated more than three times. (F) Representative immunofluorescent images of GFP-SH3KBP1 constructs expression in 10 days cultured primary myofibers. (GFP, green; Actin, red and myonuclei, blue) Scale bars, 5 μm. (G) Control (Sh-Scramble) and SH3KBP1-depleted (Sh-*Sh3kbp1*) C212 myotubes, differentiated for 6 days, were analyzed for their content of LC3-I/LC3-II and SH3KBP1 proteins by western blot; Actin labeling was used as a loading control. (H) Fold change quantification of LC3-II/Actin ratios reported to the Scramble condition. (*n* = 3; biological replicates) *P* = 0.002. Statistical analysis performed using unpaired *t* tests where **$P < 0.01$. (I) Control (Sh-Scramble) and SH3KBP1-depleted (Sh-*Sh3kbp1*) C212 myotubes differentiated for 6 days were either left untreated or treated with 100 nM of bafilomycin-A1 during 6 h. After total protein extraction, LC3-I/LC3-II and Actin levels were analyzed by immunoblot. (J) Fold change quantification of LC3-II/ Actin ratios reported to the untreated condition in each condition (Scramble or Sh3kbp1) (*n* = 3; biological replicates) *P* = 0.0011. Statistical analysis performed using unpaired *t* tests where **$P < 0.01$. (K) Quantification of the percentage of highly LC3-II positive myofibers in *Tibialis Anterior* muscles from WT or KI-*Dnm2*^R465W/+ injected with either PBS or shRNA targeting *SH3KBP1* mRNA (*n* > 3; biological replicates). Comparison between WT and WT-ShRNA-*sh3kbp1* *P* = 0.0059, KI-DNM2^R465W and KI-DNM2^R465W-ShRNA-*sh3kbp1* *P* = 0.0056. Statistical analysis performed using unpaired *t* tests where **$P < 0.01$. Boxplot whiskers represent the maximum and minimum data values. Center lines show the medians; box limits indicate the 25th and 75th percentiles as determined by R software and represents the middle 50% of observed values. (L) Representative electron microscopy images of myofibrils and triads organization within *Flexor digitalis Brevis* muscles from WT mice injected with either PBS or AAV cognate vector expressing shRNA targeting *SH3KBP1* mRNA. Scale bar = 0.2 μm or 100 nm.

autophagosomes membrane (LC3-II form). First, we observed a higher level of LC3-II in the total protein extracts of *Sh3kbp1*-depleted myotubes compare to control myotubes, indicating an increase of the autophagosome content (Fig. 8G,H). To determine if this increased number of autophagosome is the consequence of an increase or blockade of the autophagic flux, we treated myotubes with bafilomycin-A1, an inhibitor of the maturation step of the autophagic flux. We show that the inhibition of autophagy by bafilomycin-A1 induced a 2.5-fold increase of the LC3-II level in control myotubes while in SH3KBP1-depleted myotubes the increase was only of 1.4-fold (Fig. 8I,J). Thus, our results indicate that the fusion of autophagosomes with lysosomes is altered in

SH3KBP1-underexpressing myotubes. Of interest, the KI-Dnm2^R465W/+ mice model is also associated to an autophagy impairment (Puri et al, 2020; Puri and Rubinsztein, 2020). LC3 labeling was thus also realized in cross-sections of *Sh3kbp1*-depleted fibers in both WT and KI-*Dnm2*^R465W/+ mice. In WT condition, the quantification of densely-labeled LC3 myofibers showed a slight increase when we compared *Sh3kbp1*-depleted muscles to control ones, validating our previous *in cellulo* data (Fig. 8K, WT condition). In the KI-*Dnm2*^R465W/+ mice, the number of dense positive-LC3 fibers was directly increased up to 5% of total fibers, in accordance to previous study (Rabai et al, 2019). Moreover, the downregulation of SH3KBP1 was sufficient to

increase by more than 2.5-fold the ratio of LC3-dense fibers, showing a preserved role of SH3KBP1 in the support of the dynamics of the autophagic flux in skeletal muscle fibers.

Finally, as we implicated SH3KBP1 in the formation/maintenance of T-tubule and on the efficiency of the ECC process, we next questioned if ER/SR/triads structures were perturbed in the absence of SH3KBP1. Electron microscopy was used to visualize the ultrastructure of myofibrils in *Sh3kbp1*-depleted *Flexor Digitalis Brevis* muscle using AAV-shSh3kbp1 (Fig. 8L). This approach showed that in the absence of SH3KBP1, the I-band area seems perturbed as it appears much brighter compared to control condition and as triads show a "swelling-like" appearance. To better appreciate these modifications, we first quantified the distance between adjacent t-tubules that reveal a 10% decrease (Fig. EV5G), in accordance with the immunofluorescence quantification using DHPR staining (Fig. EV4C). In addition, individual T-tubule width quantification was performed and showed a 2.4-fold increase, suggesting a T-tubule "dilatation" phenotype (Fig. EV5H). Altogether, our data suggest that SH3KBP1 is a key factor of muscle fibers homeostasis through an ER/SR/Triads structure maintenance.

## Discussion

The present study highlights the role of the SH3KBP1 protein in the regulation of an endoplasmic reticulum-related pathway that control muscle fusion, myonuclei dynamics and allows skeletal muscle fibers to maintain their functionality through the formation/maintenance/functioning of triads.

One challenging process that myotubes have to face is their ability to manage myotube elongation and myonuclei spreading during the limited time when myoblasts are ready to fuse. This fusion time window is set and limited and thus any alteration in the fusion capacity/velocity will have consequences on myotube elongation (Hansson et al, 2020). Moreover, myonuclei spreading capacity depends largely on an interplay between microtubules (MTs), microtubules-associated-proteins such as MAP4, MAP7 and motors proteins such as Dynein and Kif5B and also contributes to myotube elongation process by sustaining MTs organization. Indeed, failure in the maintenance of MTs leads to myonuclei aggregation and are often associated with failure in the elongation capacity (Gimpel et al, 2017; Mogessie et al, 2015; Metzger et al, 2012; Wang et al, 2013a). Our limited siRNA screen shows a clear correlation between the capacity of myotubes to elongate and the ability of myonuclei to escape from the center of the myotube, highlighting the interplay between correct myonuclei spreading and myotube elongation. Interestingly, the absence of selected proteins related to endomembrane organelles such as GA and ER tends to enhance myotube length, in an extensive manner for the proteins related to the ER compared to GA (Fig. 1B). These results suggest that these organelles form various dense networks in the cytoplasmic compartment, which influence the organization/dynamic/polarization of the MTs network that is anchored at the membrane of myotube myonuclei. Thus, this endomembrane-related network can be considered as a brake for the myotube elongation process. Interestingly, the selected proteins related to the plasma membrane or to the cytosolic compartment do not contribute to a preferential behavior regarding elongation and spreading correlation. Indeed, some contribute to an extensive elongation associated to an enhancement in myonuclei spreading

such as HOOK3, MAP6, and MAST2, but others such as KIF5B, ANHAK1, KINECTIN, JAKMIP3, and KIFAP3 contribute to a noticeable aggregation phenotype associated to a slight effect on myotube length.

Our data show in vitro that myotubes length elongation and myonuclei fusion/spreading in developing myofibers are dependent on SH3KBP1 protein (Figs. 1–3). In vivo, when *Sh3kbp1* is depleted from mature myofibers in a period of time characterized by minimal myonuclei accretion in muscle fibers, we observe a slight increase in the percentage of myonuclei per myofibers in WT condition (Fig. EV5F). These data suggest that SH3KBP1 could contribute to the fusion efficiency by a mechanism involving ER remodeling. Indeed, myotubes elongation can contribute to increase myotubes/myoblasts contacts that thus will modulate membrane fusion and ultimately modify myonuclei accretion speed (Kim et al, 2015). As for the role of the ER on myoblast fusion, it is poorly documented and its effect seems to be more indirectly linked to pathways related to physiologic ER stress signaling and SARC (stress-activated response to $Ca^{2+}$) body formation more than a direct impact on myoblast fusion (Bohnert et al, 2017; Nakanishi et al, 2007, 2015). Moreover, there is no evidence for a clear specific role of the MTs network on the fusion potential, even if the modification of the MTs dynamics/orientation by different MAPs is known to alter fusion potential (Mogessie et al, 2015; Straube and Merdes, 2007; Gache et al, 2017). Fusion processes require remodeling of membranes and actin cytoskeleton polymerization at the site where the two membranes will fuse (Sampath et al, 2018). One can hypothesize that SH3KBP1 could contribute to the fusion process through the control of membrane remodeling on specific PM-ER sites.

Alternatively, our in vitro data suggest that the movements of organelles related to the MTs network are improved, reflected by an increase of the myonuclei motion in absence of SH3KBP1 (Fig. 4). We show that SH3KBP1 does not impact the global organization/orientation of the microtubules network in developing myotubes and that the recruitment of microtubules nucleators such as pericentrin or PCM1 to the membrane of myonuclei is not altered, suggesting alternative processes in myonuclear displacement independently from the microtubule network. The ER has been implicated in governing both MTs alignment and cytoplasmic streaming (Kimura et al, 2017). Kimura et al propose that local cytoplasmic flow generated along MTs is transmitted to neighboring regions through the ER and in turn, aligns MTs and self-organizes the collective cytoplasmic flow. SH3KBP1, by shaping and maintaining ER clustering during myofibers development, could contribute to the stability of the MTs organization at the vicinity of myonuclei and thus decrease forces applied on myotubes tips extremity that, in turn, will control both myotubes elongation and fusion events specifically at their tips.

SH3KBP1 staining is diffuse in dividing cells, while it progressively accumulates at the vicinity of myonuclei during the differentiation process and at the I-band/Z-line (Fig. 5). A few proteins have been shown to form a flexible perinuclear shield that can protect myonuclei from extrinsic forces (Wang et al, 2013b; Ghasemizadeh et al, 2021). Consequently, we suggest that SH3KBP1 belongs to the group of proteins that contribute to the stability of this perinuclear shield, through the maintenance of ER at the vicinity of myonuclei. In accordance with this preferential localization at the proximity of myonuclei, SH3KBP1 staining in transversal *Tibialis Anterior* muscle section is very intense at myonucleus site compare to nuclei belonging to other cells type

outside the fiber (Fig. EV3). The three SH3 domains of SH3KBP1 have been shown to cluster multiple proteins and protein complexes that can also contributes to the stability of those interactions. Havrylov et al identified, using mass spectrometry, several MTs-binding-proteins such as MAP7 and MAP4 that can potentially interact with SH3 domains of SH3KBP1 and that have already been shown to control myonuclei positioning in myotubes (Metzger et al, 2012; Mogessie et al, 2015; Havrylov et al, 2009). We show in the present study that SH3KBP1 role in myonuclei spreading is not related to its interaction with MAP7 (Fig. EV2G–I) and thus suggesting that additional partners of SH3KBP1 will have to be determined in the future. Alternatively, SH3KBP1 also interacts with dynamin-2 (DNM2) and has been shown to participate in the dynamic's instability of MTs (Tanabe and Takei, 2009) and in MTs nucleation (Thompson et al, 2004). Interestingly, in CNM-KI-$Dnm2^{R465W/+}$ mouse model, Fongy et al showed that myonuclei move and spread properly in heterozygous myotubes but hypothesized a defect in nuclear anchoring at the periphery (Fongy et al, 2019). Several studies pointed the importance of cytoskeleton, including MAPs, MTs and intermediate filaments, in the nuclear anchorage in mature muscle (Roman et al, 2017; Ghasemizadeh et al, 2021). SH3KBP1-dependent ER scaffolding could participate in myonuclei anchoring at the periphery of myofibers and thus in the recruitment and stabilization of a network of proteins at the vicinity of myonuclei. This hypothesis is supported by our data showing an increase in the percentage of the time in motion of myonuclei in the absence of SH3KBP1 (Fig. 4).

We also observed that SH3KBP1 depletion induces T-tubule formation impairment in vitro and a perturbation in their maintenance in vivo (Figs. 6 and 8). Myofibrils provide the contractile force under the control of the 'excitation-contraction coupling' (ECC) system that includes two membranous organelles: the sarcoplasmic reticulum (SR) and Transverse (T)-tubules (Al-Qusairi et al, 2009). These two-membrane systems are structurally associated to form the triads of skeletal muscle cells. The SR is a complex network of specialized smooth endoplasmic reticulum, essential to the storage of calcium ions. The distribution of the ER within the SR membranes is still pending, but the SR and the ER represent a continuous membrane system with different specialized subdomains (Rossi et al, 2022). T-tubule network is continuous with the muscle cell plasma membrane (PM) and begins from the invagination of PM in a repeated pattern at each sarcomere (Barone et al, 2015). These two-membrane systems are structurally associated to maintain their typical organization in muscle cells. SH3KBP1 depletion alters in vitro the organization of the ER and the formation of T-tubules. In vivo, SH3KBP1 depletion affect T-tubules maintenance, triads morphology and reduce the capacity of triads to perform efficient ECC (Figs. 6 and 8). Thus, it appears that SH3KBP1 is a node protein between the ER, SR, and PM, allowing correct triads maintenance and functions.

SH3KBP1 depletion in vivo in the *Tibialis Anterior* muscle reduces by more than 30% the average cross- section areas of myofibers (Fig. 7). Moreover, SH3KBP1-depleted mature myofibers show disorganized perinuclear ER (Fig. 8). Of interest, ER is the major initiation site for the autophagic process and ER selective autophagy (called ER-phagy or reticulophagy) has been described to control ER shape and dynamics via ER-phagy receptors that address ER portions to the autophagosomes (Grumati et al, 2018). SH3KBP1 depletion increases LC3 detection in myofibers in *Tibialis Anterior* muscle (Fig. 8G–K). The use of bafilomycin-A1 allowed us to conclude that this increase is a consequence of an alteration of the maturation step of the autophagic process and thus to a decreased autophagy dynamic. This autophagy-dependent modulation of muscle homeostasis could explain the decrease of the Cross Section Area of myofibers (Fig. 7G) and ultimately the reduction of the numbers of muscle myofibers by muscle (Fig. EV5D). Interestingly, autophagy genes have also been involved in muscle myonuclei positioning during *Drosophila* metamorphosis (Fujita et al, 2017). Moreover, SH3KBP1 interacts with dynamin-2 (our study), which is also involved in the autophagic lysosome reformation step of the process (Schulze et al, 2013). Whether this process is involved in SH3KBP1-dependent myonuclei positioning will be the subject of further investigations.

Autosomal dominant CNM is caused by heterozygous mutations in the *DNM2* gene, which encodes the Dynamin-2 (DNM2) GTPase enzyme (Romero, 2010). *DNM2*-related autosomal dominant (AD)-CNM was initially characterized as a slowly progressive muscle weakness affecting distal muscles with onset in early adulthood. DNM2-R465W missense mutation represents the most frequent mutation in humans and a knock-in (KI) mouse model expressing this mutation has been generated that develops a progressive muscle weakness (Durieux et al, 2010). Expressing DNM2-R465W mutation in mice leads to contractile impairment that precedes muscle atrophy and structural disorganization that mainly affects mitochondria and endo/sarcoplasmic reticulum. Interestingly, CNM-KI-$Dnm2^{R465W}$ mouse model exhibits twice time more *Sh3kbp1* mRNA amount in the first 4 months than control mice after 8 months (Fig. 7A). Of note, this increase is concomitant with transient transcriptional activation of regulatory genes of both ubiquitin–proteasome and autophagy pathways at 2 months of age in the TA muscle of the KI-*Dnm2* mice (Durieux et al, 2010). One can hypothesize that this elevated amount of *sh3kbp1* is one of the factors that strengthen the autophagy pathway, proteostasis and the maintenance of ER architecture via ER-phagy, thus slowing-down CNM-associated phenotypes. In accordance with this hypothesis, we found that *sh3kbp1* depletion aggravates the CNM phenotype (Fig. 7G–I). In conclusion, an increased amount of SH3KBP1 could delay the CNM phenotype development by stabilizing triads and perinuclear ER and supporting autophagy activity.

Altogether, these data are in agreement with an involvement of *Sh3kbp1* in muscle fibers formation and maintenance. In this study, we show that the adapter protein SH3KBP1, through its C-terminus part, is able to interact and scaffold the ER, that in turn control fusion events, myotubes elongation and myonuclei velocity while it interacts with DNM2 via its N-terminal part and contributes to the formation/maintenance of T-tubules and triads functions in skeletal muscle.

# Methods

**Reagents and tools table**

| Reagent/resource | Reference or source | Identifier or catalog number |
|---|---|---|
| **Experimental models** | | |
| C57BL/6J | Jackson Laboratory | #000664 |
| CNM-KI-$Dnm2^{R465W}$ | Dr. Marc Bitoun | N/A |
| **Recombinant DNA** | | |
| pcDNA-SH3KBP1-FL-GFP | Dr. Vladimir L. Buchman | N/A |

| Reagent/resource | Reference or source | Identifier or catalog number |
|---|---|---|
| pcDNA-SH3KBP1-Nter-GFP | Dr. Vladimir L. Buchman | N/A |
| pcDNA-SH3KBP1-Cter-GFP | Dr. Vladimir L. Buchman | N/A |
| pcDNA-SH3KBP1-sCter-GFP | Dr. Vladimir L. Buchman | N/A |
| pcDNA-SH3KBP1-FL-Flag | Dr. Vladimir L. Buchman | N/A |
| pcDNA-SH3KBP1-Nter-Flag | Dr. Vladimir L. Buchman | N/A |
| pcDNA-SH3KBP1-Cter-Flag | Dr. Vladimir L. Buchman | N/A |
| pcDNA-SH3KBP1-sCter-Flag | Dr. Vladimir L. Buchman | N/A |
| pcDNA-DNM2-GFP | Dr. Marc Bitoun | N/A |
| **Antibodies** | | |
| Anti-SH3KBP1 | Sigma-Aldrich | HPA003355 |
| Anti-SH3KBP1 | Santa-Cruz | sc-48746 |
| Anti-myosin heavy chain | DSHB | MF20 |
| Anti-DHPR-a | DSHB | IIID5E1 |
| Anti-RCAS1 | Cell Signaling | #12290 |
| Anti-Erp72 | Cell Signaling | #5033 |
| Anti-calnexin | Cell Signaling | #2679 |
| Anti-LC3-II/LC3-I | Sigma-Aldrich | L7543 |
| Anti-alpha tubulin | Sigma-Aldrich | T6074 |
| Anti-Myogenin | Santa-Cruz | sc-12732 |
| Anti-GFP | Chromotek | 3H9 |
| Anti-LC3 | Sigma-Aldrich | L7543 |
| Anti Actin clone C4 | Millipore | MAB1501R |
| Anti-Flag | Sigma-Aldrich | F1804 |
| Anti-mouse-HRP | Invitrogen | 62-6520 |
| Anti-rabbit-HRP | Invitrogen | 65-6120 |
| Dapi-brilliant blue | Thermo Fisher Scientific | D1306 |
| Phalloidin-Alexa Fluor 647 | Thermo Fisher Scientific | A22287 |
| Anti-rat- Alexa Fluor 488 | Thermo Fisher Scientific | A-21208 |
| Anti-rabbit- Alexa Fluor 647 | Thermo Fisher Scientific | A-21245 |
| Anti-mouse- Alexa Fluor 647 | Thermo Fisher Scientific | A-21240 |
| **Oligonucleotides and other sequence-based reagents** | | |
| siRNA-Kif5b | Thermo Fisher Scientific | #155041 |
| siRNA-Kinectin | Thermo Fisher Scientific | #62858 |
| siRNA-Ahnak | Thermo Fisher Scientific | #288166 |
| siRNA-Jackmip | Thermo Fisher Scientific | #508234 |
| siRNA-Calreticulin | Thermo Fisher Scientific | #60376 |
| siRNA-Akap9 | Thermo Fisher Scientific | #224020 |
| siRNA-Clasp2 | Thermo Fisher Scientific | #262252 |

| Reagent/resource | Reference or source | Identifier or catalog number |
|---|---|---|
| siRNA-Mast1 | Thermo Fisher Scientific | #74635 |
| siRNA-Stahmin4 | Thermo Fisher Scientific | #74162 |
| siRNA-Bicd1 | Thermo Fisher Scientific | #174597 |
| siRNA-Arl3 | Thermo Fisher Scientific | #74292 |
| siRNA-Clasp1 | Thermo Fisher Scientific | #508887 |
| siRNA-Akap12 | Thermo Fisher Scientific | #175072 |
| siRNA-DLG4 | Thermo Fisher Scientific | #61245 |
| siRNA-Stau1 | Thermo Fisher Scientific | #69550 |
| siRNA-Elmo1 | Thermo Fisher Scientific | #258892 |
| siRNA-Fez1 | Thermo Fisher Scientific | #154805 |
| siRNA-DLG1 | Thermo Fisher Scientific | #61157 |
| siRNA-Obscurin | Thermo Fisher Scientific | #223870 |
| siRNA-DLG3 | Thermo Fisher Scientific | #72408 |
| siRNA-Mark1 | Thermo Fisher Scientific | #169860 |
| siRNA-Mast4 | Thermo Fisher Scientific | #503640 |
| siRNA-Mast3 | Thermo Fisher Scientific | #299839 |
| siRNA-Mast2 | Thermo Fisher Scientific | #164709 |
| siRNA-Cdk5RAP2 | Thermo Fisher Scientific | #90312 |
| siRNA-Map6 | Thermo Fisher Scientific | #68385 |
| siRNA-Mark2 | Thermo Fisher Scientific | #157320 |
| siRNA-Hook3 | Thermo Fisher Scientific | #166825 |
| siRNA-KifAP3 | Thermo Fisher Scientific | #67656 |
| siRNA-Sh3KBP1 | Thermo Fisher Scientific | #75484 |
| shRNA-Sh3kbp1 (AAV) | GeneCopoeia | MSH032547-2-CU6 |
| shRNA-Sh3kbp1 (cells) | MERCK | TRCN0000088508 |
| shRNA-ctrl | MERCK | SHC002 |

| RT-qPCR primers | Forward sequence | Reverse sequence |
|---|---|---|
| Sh3kbp1 | ATTGATTCACCGATACAGGCCC | ATCTTCTGCATCTAGCAGTCGG |
| Gapdh | AACTTTGGCATTGTGGAAGG | ACACATTGGGGGTAGGAACA |
| Myogenin | CAATGCACTGGAGTTCGGTC | ACAATCTCAGTTGGGCATGG |
| Gusb | GAGGATTGCCAACGAAACCG | GTGTCTGGGGACCACCTTTGA |
| RpL4 | GCCATGAGAGCGAAGTGG | CTCCTGCAGGCGTCGTAG |

| Software | | |
|---|---|---|
| imagej | | |
| BoxPlotR | http://shiny.chemgrid.org/boxplotr | |
| GraphPad PRISM 5.0 | | |
| Metamorph | | |
| SkyPad | Cadot et al, 2014 | |

## Cell culture and cell lines

Primary myoblasts were collected from wild-type C57BL6 mice as described before (Falcone et al, 2014; Pimentel et al, 2017). Briefly, hindlimb muscles from 6 days pups were extracted and digested with collagenase (Sigma, C9263-1G) and dispase (Roche, 04942078001). After a pre-plating step to discard contaminant cells such as fibroblasts, myoblasts were cultured on matrigel-coated dishes (Corning, 356231) and induced to differentiate in myotubes for 2–3 days in differentiation medium (DM: IMDM

(Gibco, 21980-032) + 2% of horse serum (Gibco, 16050-122) + 1% penicillin–streptomycin (Gibco, 15140-122)). Myotubes were then covered by a concentrated layer of matrigel and maintained for up to 10 days in long differentiation culture medium (LDM: IMDM (Gibco, 21980-032) + 2% of horse serum (Gibco, 16050-122) + 0.1% Agrin + 1% penicillin–streptomycin (Gibco, 15140-122)) until the formation of mature and contracting myofibers. LDM was changed every two days. Cell lines were regularly tested for mycoplasma contamination.

Mouse myoblast C2C12 cells were cultured in Dulbecco's modified Eagle's medium (DMEM (Gibco, 41966029) + 15% fetal bovine serum (FBS) (Gibco, 10270-106) + 1% penicillin–streptomycin (Gibco, 15140-122)) and were plated on 0.1% matrigel-coated dishes for 1–2 days before differentiation. Differentiation was induced by switching to differentiation media (DMEM + 1% horse serum). Inhibition of autophagy maturation was performed by treating C2C12 myotubes with Bafilomycin-A1 (MedChemExpress, HY-100558) at 100 nM during 6 h.

To generate C2C12 stably expressing shRNAs (shControl or shCin85), cells were transfected with plasmids encoding control shRNA (Mission pLKO.1-Puro non-mammalian shRNA control plasmid, Merck SHC002) or shRNA directed against SH3KBP1 (mission shRNA clone TRCN0000088508 targeting the 3'UTR sequence CCCACCACTCTAAGAGAAATT) and selected using 2 µg/mL of Puromycin (Gibco, A1113803) for 2 weeks. The clones were then amplified and analyzed for their content of Cin85 protein, proliferation and differentiation capacities.

## Cell transfection

For C2C12 cells, three different Silencer siRNAs per gene were transfected using Lipofectamine 2000 (Thermo Fisher Scientific, 11668-019) at the final concentration of 10 nM, following the manufacturer's instructions, 2 days before differentiation. For shRNA cDNA (Geneocopia) transfections, Lipofectamine 2000 (Thermo Fisher Scientific, 11668-019) was used following the manufacturer's instructions.

For primary cells, siRNAs were transfected using Lipofectamine 2000 (Thermo Fisher Scientific, 11668-019) at the final concentration of 2 nM. shRNA (Geneocopia), Eb1 or RFP-Lamin-chromobody (Chromotek) cDNA were transfected using Lipofectamine 3000 (Thermo Fisher Scientific, L3000-008).

## Protein sample preparation and muscle protein extraction

For primary cultured cells or C2C12 cell line, cells were harvested using Trypsin for 5 min at 37 °C and centrifuged at 1500 rpm for 5 min at 4 °C. Cell pellets were diluted and incubated in the optimal volume of RIPA lysis buffer containing phosphatases inhibitors (Sigma, P5726-5mL) and proteases inhibitors (Sigma, P8340) for 10 min at 4 °C. Following a sonication and a centrifugation at 12,000 rpm for 10 min at 4 °C, protein samples were collected for further use.

To obtain protein extracts from dissected muscles or other organs, the samples were finely dilacerated into tubes containing ceramic beads (MP Biomedicals Lysing Matrix D, 6913-100) with an optimal volume of RIPA lysis buffer completed with phosphatases inhibitors (Sigma, P5726-5mL) and proteases inhibitors (Sigma, P8340). Tubes

were subjected to harsh shaking (6500 rpm, 3 cycles of 15 s with 30 s pause intervals at 4 °C) and rested at 4 °C for 1 h. Following a sonication and a centrifugation at 12,000 rpm for 10 min at 4 °C, protein samples were collected for further use.

The concentration of proteins was determined using BCA protein assay kit (Thermo Fisher Scientific, 23225) as described by the manufacturer.

## Western blot

After protein transfer from SDS-Page gels, western blot membranes were saturated in 5% milk or BSA (depending on the antibody) in 0.1% Tween-20 TBS1X for 1 h at room temperature (RT) and were incubated in primary antibodies overnight at 4 °C or during 1 h at RT. Following washes by 0.1% Tween-20-1× TBS, the membranes were incubated in HRP-conjugated secondary antibodies in 5% milk or 0.1% BSA in TBS-0.1% Tween for 1 h at RT. Following washes by 0.1% Tween-20 in TBS and TBS, the detection of the target proteins was carried out using Super Signal West Femto (Thermo Fisher Scientific, 34095) or ECL prime western blotting detection reagent (Cytiva, RPN2232) and ChemiDoc imaging system (BioRad).

## Isolation of myofibers and immunofluorescence staining

Following the dissection of the whole muscle from the mice, muscle blocks were fixed in 4% PFA in PBS for 2 h at RT. After several washes, 30–50 mono-myofibers were isolated per staining from each muscle. Myofibers were then permeabilized using 1% Triton-X100 in PBS for 15 min at 37 °C and saturated in 1% BSA in PBS for 30 min at RT. Then, myofibers were incubated in desired primary antibodies at 4 °C for two nights. Following washes with 0.05% Triton-X100 in PBS, myofibers were incubated in secondary antibodies or dyes for 2 h at RT and washed several times with 0.05% Triton-X100 in PBS before mounting on slides, using fluoromount Aqueous mounting medium (Sigma, F4680-25mL). Fibers were kept at 4 °C until image acquisition.

## Analysis of intracellular Ca²⁺ in voltage-clamped fibers

Single fibers were isolated from *FDB* muscles as described previously (Jacquemond and Allard, 1998). In brief, muscles were incubated for 60 min at 37 °C in the presence of external Tyrode containing 2 mg/mL collagenase (Sigma, type 1). Single fibers were obtained by triturating the collagenase-treated muscles within the experimental chamber.

Isolated muscle fibers were handled with the silicone voltage-clamp technique (Lefebvre et al, 2014). Briefly, fibers were partly insulated with silicone grease so that only a short portion (50–100 µm long) of the fiber extremity remained out of the silicone. Fibers were bathed in a standard voltage-clamp extra-cellular solution containing (in mM) 140 TEA-methanesulfonate, 2.5 CaCl₂, 2 MgCl₂, 1 4-aminopyridine, 10 HEPES and 0.002 tetrodotoxin. An RK-400 patch-clamp amplifier (Bio-Logic, Claix) was used in whole-cell configuration in combination with an analog-digital converter (Axon Instruments, Digidata 1440 A) controlled by pClamp 9 software (Axon Instruments). Voltage-clamp was performed with a micropipette filled with a solution containing (in mM) 120 K-glutamate, 5 Na2-ATP, 5 Na2-phosphocreatine, 5.5 MgCl₂, 15 EGTA, 6 CaCl₂, 0.1 rhod-2, 5

glucose, 5 HEPES. The tip of the micropipette was inserted through the silicone within the insulated part of the fiber and was gently crushed against the bottom of the chamber to ease intracellular equilibration and decrease the series resistance. Intracellular equilibration of the solution was allowed for 30 min before initiating measurements. Membrane depolarizing steps of 0.5 s duration were applied from -80mV. Confocal imaging was conducted with a Zeiss LSM 5 Exciter microscope equipped with a 63× oil immersion objective (numerical aperture 1.4). Rhod-2 fluorescence was detected in line-scan mode ($x,t$, 1.15 ms per line) above 560 nm, upon excitation from the 543 nm line of a HeNe laser. Rhod-2 fluorescence transients were expressed as F/F0 where F0 is the baseline fluorescence. The $Ca^{2+}$ release flux (rate of SR $Ca^{2+}$ release) was estimated from the time derivative of the total released cytoplasmic $Ca^{2+}$ ([Catot]) calculated from the occupancy of intracellular calcium binding sites following a previously described procedure (Kutchukian et al, 2017).

## Video microscopy

Time-lapse images were acquired using Z1-AxioObserver (Zeiss) with intervals of 20 min. Final videos were analyzed using Metamorph (Zeiss) and SkyPad plugin as described before (Cadot et al, 2014).

## Adeno-associated virus production and in vivo transduction

A cassette containing the small hairpin (sh) RNA under the control of H1 RNA polymerase III promoter was inserted in a pSMD2 expression plasmid. AAV vectors (serotype 1) were produced in HEK293 cells after transfection of the pSMD2-shRNA plasmid, the pXX6 plasmid coding for viral helper genes essential for AAV production and the pRepCap plasmid (p0001) coding for AAV1 capsid as described previously (Riviere et al, 2006). Viral particles were purified on iodixanol gradients and concentrated on Amicon Ultra-15 100 K columns (Merck-Millipore). The concentration of viral genomes (vg/ml) was determined by quantitative real-time PCR on a Light-Cycler480 (Roche diagnostic, France) by using TaqMan probe. A control pSMD2 plasmid was tenfold serially diluted (from $10^7$ to $10^1$ copies) and used as a control to establish the standard curve for absolute quantification. Male wild-type and heterozygous KI-$Dnm2^{R465W/+}$ mice were injected under isoflurane anesthesia. Two intramuscular injections of 30 µl within 24 h interval were performed using 29 G needle in TA muscles corresponding to $10^{11}$ viral genomes per muscle. All the experiments and procedures were conducted in accordance with the guidelines of the local animal ethics committee of the University Claude Bernard—Lyon 1/and Groupe Hospitalier Universitaire La Pitié-Salpêtrière, in accordance with French and European legislation on animal experimentation and approved by the ethics committee CECCAPP and the French ministry of research (APAFIS#21085-2019060719395546 v5).

## Muscle contractile properties

The isometric contractile properties of TA muscles were studied in situ on mice anesthetized with 60 mg/kg pentobarbital. The distal tendon of the TA muscle was attached to a lever arm of a servomotor system (305B Dual-Mode Lever, Aurora Scientific). The sciatic nerve was stimulated by a bipolar silver electrode using a supramaximal (10 V) square wave pulse of 0.1 ms duration. Absolute maximal isometric tetanic force was measured during isometric contractions in response to electrical stimulation (frequency of 25–150 Hz; train of stimulation of 500 ms). All isometric contraction measurements were made at optimal muscle length. Forces are expressed in grams (1 g = 9.8 mNewton). Mice were sacrificed by cervical dislocation and TA muscles were weighted.

## RNA extraction

After the addition of Trizol (Sigma, T9424-200mL) on each sample, lysing matrix D and fast prep system (MPbio, 6913-100) were used for sample digestion and pre-RNA extraction. In order to extract RNA, samples were incubated in chloroform for 5 min at RT, centrifuged for 15 min at 12,000 rcf at 4 °C and incubated in the tubes containing isopropanol (precipitation of RNA) for 10 min at RT. Following a centrifuge of samples for 15 min at 12,000 rcf at 4 °C, samples were washed 2 times with 70% ethanol and the final RNA pellets were diluted in ultra-pure RNase-free water (Invitrogen, 10977-035). RNA concentration was calculated using Nanodrop (Thermo Fisher Scientific).

## RT-qPCR on cells

Goscript Reverse Transcriptase System (Promega, A5001) was used, as described by the manufacturer to produce the cDNA. Fast Start Universal SYBR Green Master (Rox) (Roche, 04913914001) and CFX Connect™ Real-Time PCR Detection System (BioRad) were used to carry out the quantitative PCR using the following primer sets. The CT of target genes were normalized on 3 control genes. For the list of primers used, see below.

## RT-qPCR on muscle samples

In total, 50 longitudinal sections (12 µm) of TA muscles were cut and used for RNA isolation and RT-qPCR. Total RNA was extracted from muscle by using NucleoSpin kit (Macherey-Nagel). RNA (200 ng) was reverse transcribed using Reverse Transcription Core Kit (Eurogentec). Real-time PCR was performed in a 20 µL final volume using the Takyon No Rox SYBR kit (Eurogentec). Fluorescence intensity was recorded using a CFX96 Real-Time PCR Detection System (BioRad) and the data analyzed using the ΔΔCt method of analysis. Reference gene 18 s was used to normalize the expression level of the gene of interest. The selected forward and reverse primer sequences are listed below. Statistical analyses were performed using GraphPad PRISM 5.0 (La Jolla). Data were analyzed for normal distribution using Shapiro–Wilk test. Non-parametric one-way ANOVA ($n = 4$–6) was used to determine transcripts expression level. Primers were designed using Primer 3 software from gene sequences obtained from Genebank. Primer specificity was determined using a BLAST search.

## Histological staining and analysis

*Tibialis anterior* muscles were collected, embedded in tragacanth gum, and quickly frozen in isopentane cooled in liquid nitrogen. Cross-sections (10-µm thick) were obtained from the middle portion of frozen muscles and processed for histological, immunohistochemical

according to standard protocols. The fibers cross-sectional area and the number of centrally nucleated fibers were determined using Laminin and Dapi-stained sections. Fluorescence microscopy and transmission microscopy were performed using Axioimager Z1 microscope with CP Achromat 5x/0.12, 10x/0.3 Ph1, or 20x/0.5 Plan NeoFluar objectives (Zeiss). Images were captured using a charge-coupled device monochrome camera (Coolsnap HQ, Photometrics) or color camera (Coolsnap colour) and MetaMorph software. For all imaging, exposure settings were identical between compared samples. Fiber number and size, central nuclei and peripheral myonuclei were calculated using ImageJ software.

## Quantification methods for myonuclei spreading in myotubes

Quantifications in immature myotubes were assessed using a home-made analysis tool (see Ghasemizadeh et al, 2021). An image analysis performed in ImageJ® software is combined with a statistical analysis in RStudio® software. This provides quantifications of parameters, ranked by myonuclei content per myotubes, regarding phenotype of myotubes (area, length) and their respective myonuclei positioning compare to centroid of myotubes (DiMycMio).

MSG diagrams were obtained through the normalization of lengths of all analyzed myotubes (independently to their myonuclei content) to 100%. White lines represent myonuclei density curves assessing the statistical frequency for myonuclei positioning along myotubes. Each color group reflects statistical estimation of myonuclei clustering along myotubes.

## Quantification methods for the orientation of microtubules bundles inside myotubes

Myotubes were treated in live with sir-tubulin® according to the manufacturer protocol (spirochrome). Fluorescence microscopy was performed using Nikon AX confocal microscope with a 60X oil objective. Images obtained were cropped as shown in Fig. EV2A (zooms) and image analysis was performed in ImageJ® software using the directionality plugin (https://imagej.net/plugins/directionality). This provides quantifications of parameters regarding the orientation of Microtubules bundles classified by angles categories, that were then plotted using BoxPlotR (http://shiny.chemgrid.org/boxplotr/).

## Electron microscopy

Tissues were cut into small pieces and fixed in 2% glutaraldehyde for 2 h at 4 °C. Samples were washed three times for 1 h at 4 °C and post-fixed with 2% OsO$_4$ 1 h at 4 °C. Then tissues were dehydrated with an increasing ethanol gradient (5 min in 30%, 50%, 70%, 95%) and three times for 10 min in absolute ethanol. Impregnation was performed with Epon A (75%) plus Epon B (25%) plus DMP30 (1.7%). Inclusion was obtained by polymerization at 60 °C for 72 h. Ultrathin sections (~70 nm thick) were cut on a UC7 (Leica) ultra-microtome, mounted on 200 mesh copper grids coated with 1:1000 polylysine, and stabilized for 1 day at room temperature and contrasted with uranyl acetate and lead citrate. Sections were acquired with a Jeol 1400JEM (Tokyo, Japan) transmission electron microscope, 80 Kv, equipped with a Orius 600 camera and Digital Micrograph.

## Statistical analysis

All the results were analyzed using Microsoft Excel and BoxPlotR (http://shiny.chemgrid.org/boxplotr/). The results are representative of at least three biological replicate experiments. Center lines show the medians; box limits indicate the 25th and 75th percentiles as determined by R software; whiskers extend 1.5 times the interquartile range from the 25th and 75th percentiles, outliers are represented by dots as indicated in the figure legends. We used the two-tailed Student's $t$ test to calculate the statistical significance. $P$ values < 0.05 were considered statistically significant. The researchers were not blinded during data collection and analysis.

## Data availability

The datasets produced in this study are available in the following databases: https://www.ebi.ac.uk/biostudies/studies/S-BSST1788?key=2bd0a76e-9c39-4588-8d8e-db9b986883b7.

The source data of this paper are collected in the following database record: biostudies:S-SCDT-10_1038-S44319-025-00413-9.

## Peer review information

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

## Acknowledgements

The authors thank the Penn Vector Core, Gene Therapy Program (University of Pennsylvania, Philadelphia, USA) for providing pAAV1 plasmid (p0001), and Sofia Benkhelifa-Ziyyat for AAV production. We acknowledge the contributions of the CELPHEDIA Infrastructure (http://www.celphedia.eu/), especially the center AniRA of Lyon in addition to members of CIQLE imaging center (Faculté de Médecine Rockefeller, Lyon-Est). This work is supported by the ATIP-AVENIR Program (R14074CS), Association Française contre les Myopathies (MyoNeurAlp Alliance), and the ANR Atrorescue (ANR-21-CE14-0064-01).

## Author contributions

**Alexandre Guiraud**: Conceptualization; Formal analysis; Investigation; Methodology; Writing—original draft; Project administration. **Nathalie Couturier**: Formal analysis; Investigation; Methodology. **Emilie Christin**: Formal analysis; Methodology. **Léa Castellano**: Formal analysis; Investigation; Methodology. **Marine Daura**: Formal analysis; Investigation; Methodology. **Carole Kretz-Remy**: Conceptualization; Resources; Formal analysis; Funding acquisition; Validation; Investigation; Visualization; Methodology; Writing—original draft; Writing—review and editing. **Alexandre Janin**: Resources; Formal analysis; Methodology. **Alireza Ghasemizadeh**: Formal analysis; Investigation; Methodology. **Peggy del Carmine**: Formal analysis. **Laloe Monteiro**: Formal analysis; Investigation. **Ludivine Rotard**: Formal analysis; Investigation. **Colline Sanchez**: Formal analysis; Investigation. **Vincent Jacquemond**: Resources; Formal analysis; Investigation; Methodology; Writing—original draft. **Claire Burny**: Software. **Stéphane Janczarski**: Software. **Anne-Cécile Durieux**: Resources; Formal analysis; Investigation. **David Arnould**: Formal analysis; Methodology. **Norma Beatriz Romero**: Resources. **Mai Thao Bui**: Resources. **Vladimir L Buchman**: Resources; Writing—original draft. **Laura Julien**: Formal analysis; Methodology. **Marc Bitoun**: Resources; Formal analysis; Investigation; Methodology; Writing—original draft. **Vincent Gache**: Conceptualization; Formal analysis; Supervision; Funding acquisition; Validation; Investigation; Visualization; Methodology; Writing—original draft; Project administration; Writing—review and editing.

Source data underlying figure panels in this paper may have individual authorship assigned. Where available, figure panel/source data authorship is listed in the following database record: biostudies:S-SCDT-10_1038-S44319-025-00413-9.

## Disclosure and competing interests statement

The authors declare no competing interests.

# Expanded View Figures

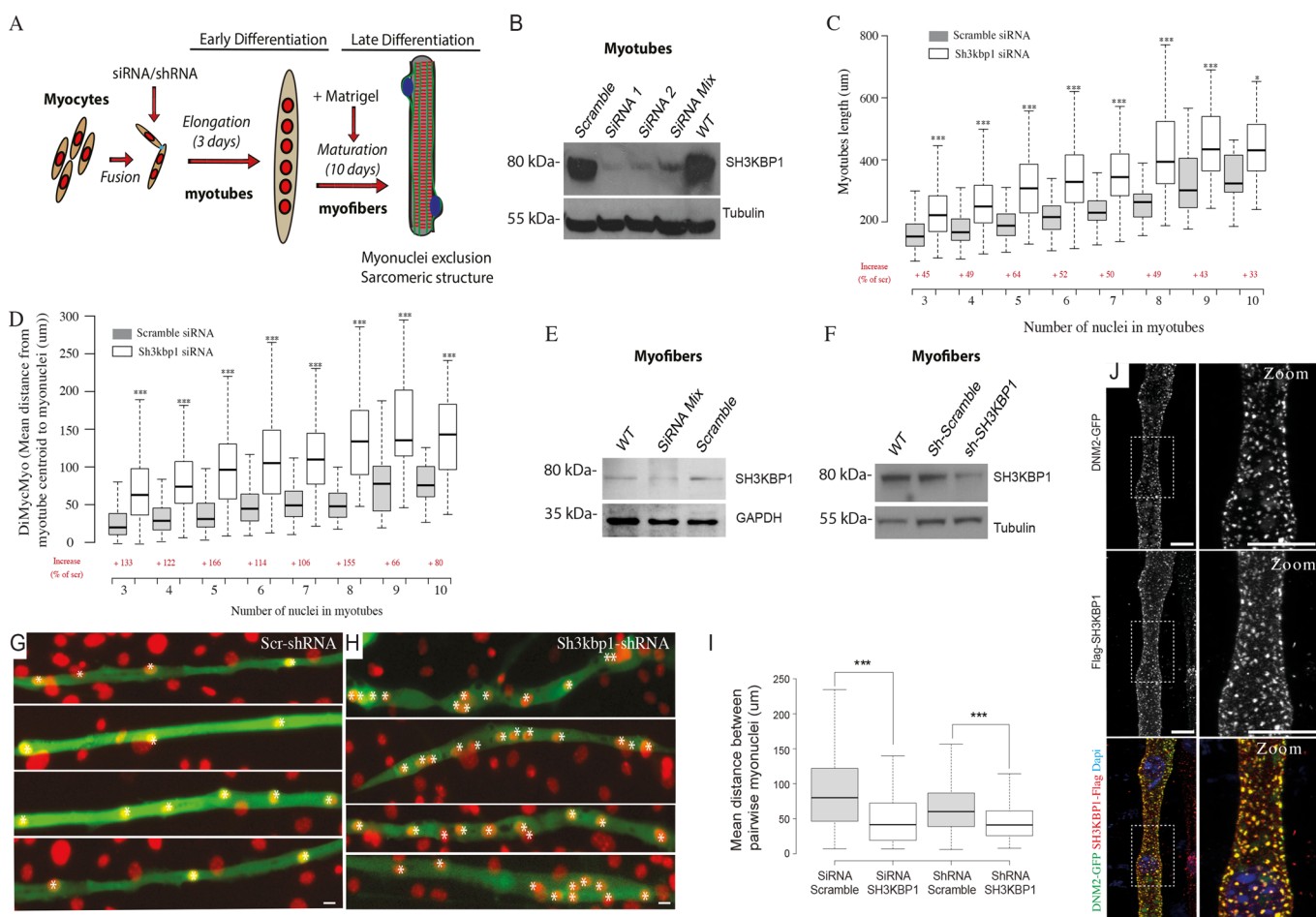

**Figure EV1.   SH3KBP1 affects myotubes elongation and myonuclei spreading.**

(A) Sequential steps performed to obtain mature myofibers from primary myoblasts. siRNAs & shRNAs were transfected 24 h before myoblasts fusion. (B) Representative western blot analysis of SH3KBP1 protein levels in total protein extracts obtained after sh3kbp1 depletion using either 2 individual siRNA (1 & 2) or a pool of siRNA (Mix) after 3 days of differentiation of primary myoblasts; Tubulin used as loading control. (C) Distribution of myotubes length ranked by myonuclei content per myotubes quantified after 3 days of differentiation in cells treated with scramble or *Sh3kbp1* siRNAs. Comparison between Scramble siRNA and *sh3kbp1* siRNA for "3-nuclei" $P = 1,7E^{-26}$, "4-nuclei" $P = 9,9E^{-28}$, "5-nuclei" $P = 3,2E^{-22}$, "6-nuclei" $P = 6,35E^{-22}$, "7-nuclei" $P = 4,5E^{-15}$, "8-nuclei" $P = 2,1E^{-11}$, "9-nuclei" $P = 9,1E^{-7}$, "10-nuclei" $P = 0,03$. Statistical analysis performed using unpaired $t$ tests where ***$P < 0.001$ and *$P < 0.05$. (D) Distribution of the mean distances between each myonuclei and myotube centroids (DiMycMyo) ranked by myonuclei content per myotubes were quantified after 3 days of differentiation in cells treated with scramble or *Sh3kbp1* siRNAs. Comparison between Scramble siRNA and *sh3kbp1* siRNA for "3-nuclei" $P = 1,9E^{-34}$, "4-nuclei" $P = 4,5E^{-38}$, "5-nuclei" $P = 3,7E^{-27}$, "6-nuclei" $P = 3,5E^{-23}$, "7-nuclei" $P = 1,2E^{-17}$, "8-nuclei" $P = 7,5E^{-15}$, "9-nuclei" $P = 1,3E^{-7}$, "10-nuclei" $P = 0,0006$. Statistical analysis performed using unpaired $t$ tests where ***$P < 0.001$. (C, D) Data from five independent experiments were combined. Scramble siRNA cells ($n = 1010$ cells) and *Sh3kbp1* siRNA cells ($n = 1093$ cells). Boxplot whiskers represent the maximum and minimum data values. Center lines show the medians; box limits indicate the 25th and 75th percentiles as determined by R software and represents the middle 50% of observed values. (E, F) Representative western blot analysis of SH3KBP1 protein levels in total protein extracts obtained after SH3KBP1 depletion using a pool of siRNA (Mix) or a shRNA targeting *sh3kbp1* after 10 days of differentiation of primary myoblasts (E, F, respectively); Tubulin or GAPDH used as loading control (G, H) Four representative images of 10 days differentiated myofibers transfected with either scramble (G) or *Sh3kbp1* shRNA tagged with GFP (H); shRNA (green) myonuclei (DAPI, red). Scale bar: 10 μm (asterisks are individual myonuclei). (I) Quantification of the mean distance between pairwise myonuclei in 10 days differentiated myofibers treated with scramble siRNA or shRNA, or with a pool of 2 individual siRNAs or of one individual shRNAs targeting *Sh3kbp1* ($n > 3$; biological replicates). Comparison between Scramble siRNA and *sh3kbp1* siRNA, $P = 3,1E^{-34}$; Comparison between Scramble shRNA and *sh3kbp1* shRNA, $P = 4,9E^{-34}$ Statistical analysis performed using unpaired $t$ tests where ***$P < 0.001$. Boxplot whiskers represent the maximum and minimum data values. Center lines show the medians; box limits indicate the 25th and 75th percentiles as determined by R software and represents the middle 50% of observed values (J) Representative immunofluorescent staining of in vitro myofibers co-expressing GFP-DNM2 (green), SH3KBP1-Flag (red) and myonuclei (blue) in primary myotubes differentiated for 10 days. (Max intensity of Z stacks plans) Scale bars, 10 μm.

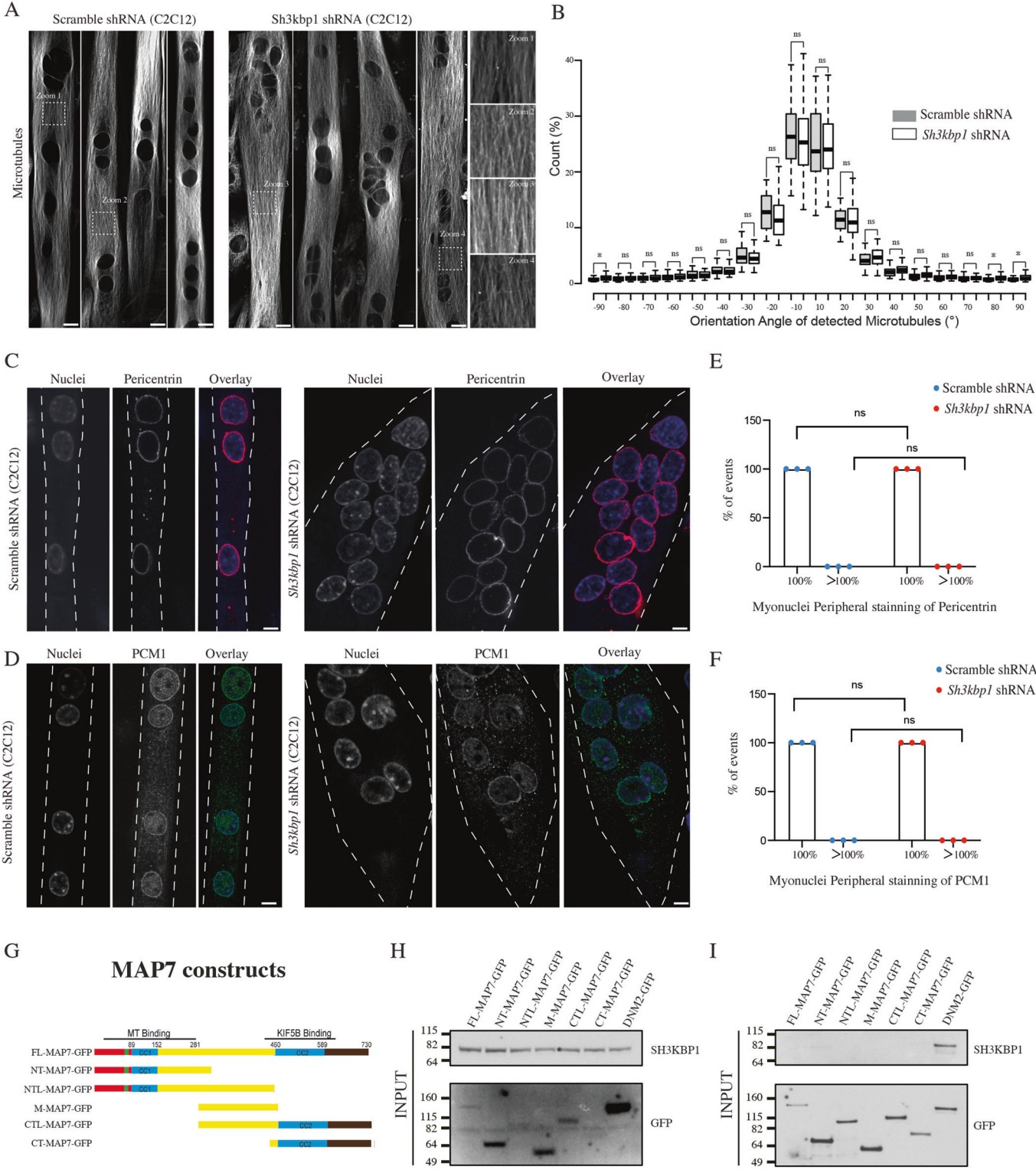

◀   **Figure EV2.   SH3KBP1 is not affecting microtubule nucleation and organization in developing myotubes.**

(A) Representative immunofluorescence staining of 5 days C2C12 myotubes expressing either scramble or *Sh3kbp1* shRNA and stained with sirTubulin˚. Scale bars, 10 μm.
(B) Quantification of the microtubule bundle directionality (orientation angle normalized according to myotubes longitudinal axis) in myotubes using the "directionality plugin" of ImageJ˚. Data are pooled from three independent repeats ($n = 41$ cells in scramble condition and $n = 61$ cells in *Sh3kbp1* shRNA condition). "$-90°$" category $P = 0.03$; "$+80°$" category $P = 0.02$"$+90°$" category $P = 0.017$. Statistical analysis performed using unpaired *t* tests where $*P < 0.05$. Boxplot whiskers represent the maximum and minimum data values. Center lines show the medians; box limits indicate the 25th and 75th percentiles as determined by R software and represents the middle 50% of observed values. (C, D) Representative images of immunofluorescent staining of Pericentrin (red), PCM1 (green) and nuclei (Blue) in 5 days differentiated C2C12 myotubes expressing either scramble or *Sh3kbp1* shRNA. Scale bars, 10 μm. (E, F) Quantification of the myonuclei peripheral staining of Pericentrin (E) or PCM1 (F) in 5 days differentiated C2C12 myotubes expressing either scramble or *Sh3kbp1* shRNA. Data are pooled from three independent repeats. Error bars represent SD (G) MAP7 constructs used in the experiment. (H) Representative western blot of crude extracts of C2C12 cells expressing various GFP-MAP7 constructs (FL: Full length, NT: N-terminal part of MAP7; NTL: N-terminal long part of MAP7; M: Middle part of MAP7, CT: C-terminal part of MAP7 and CTL: C-terminal long part of MAP7) and GFP-DNM2 and stained for endogenous SH3KBP1 (top) or with anti-GFP (bottom) antibodies. (I) Representative western blot after GFP immunoprecipitation (MAP7 and DNM2 constructs) using GFP-Trap in C2C12 cell extracts (H). The membrane was revealed with anti-GFP (bottom) and anti-SH3KBP1 (Top) antibodies *n.*>3.

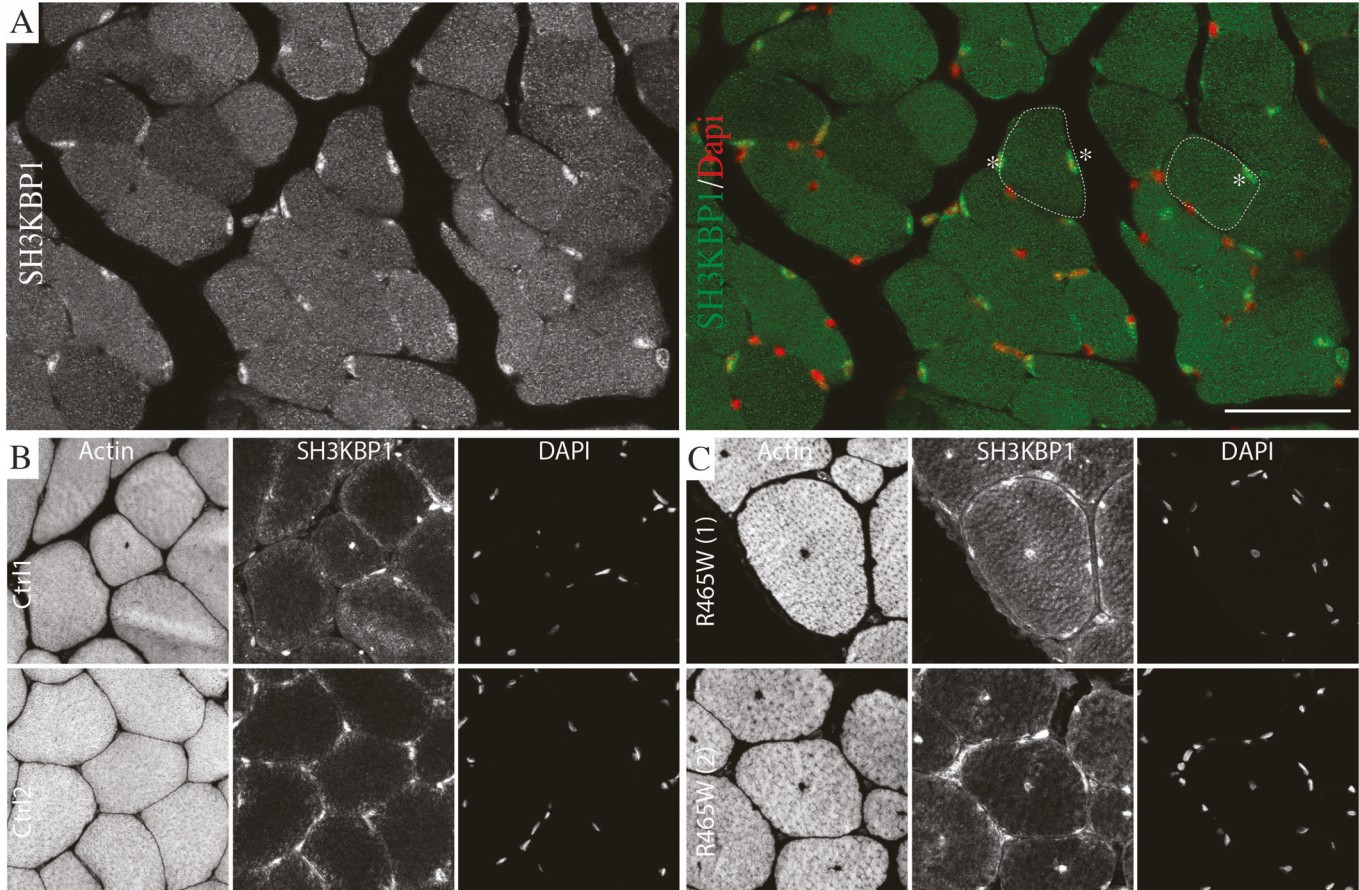

**Figure EV3. SH3KBP1 is localized at myonuclei vicinity in mouse and human skeletal muscle fibers.**

(A) Representative images of *Tibialis Anterior* muscle transversal cross-sections stained for SH3KBP1 (green) and myonuclei (Dapi, red). Asterisks show myonuclei inside myofibers. Scale bars, 150 μm. (B, C) Representative images of immunofluorescent staining of SH3KBP1, Actin and nuclei (Dapi) in transversal cross-sections of control human subject or of CNM patient harboring the p.R465W mutation. Scale bars, 150 μm.

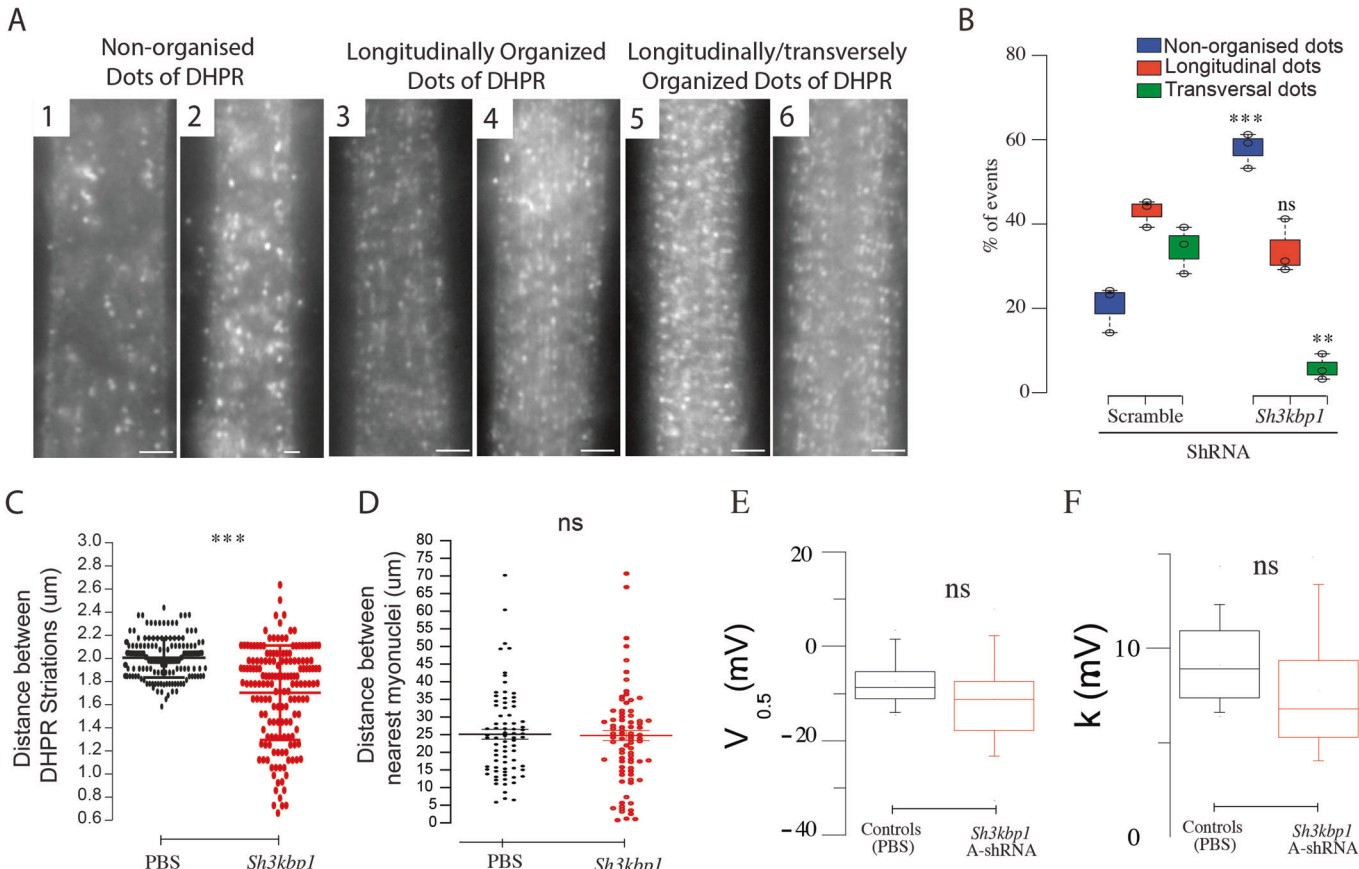

**Figure EV4. SH3KBP1 contributes to T-tubule formation and Triads function**

(A) Representative immunofluorescent images used for the classification of DHPR1-α staining patterns in 10 days cultured primary myotubes after scramble siRNA transfection. (1–2: non-organized DHPR1-α staining dots; 3-4: longitudinally organized DHPR1-α staining dots; 5–6: longitudinally and transversally organized DHPR1-α staining dots) Scale bars, 5 μm. (B) Quantification of DHPR1-α aspects in mature myofibers treated with either scramble shRNA or shRNA targeting *Sh3kbp1* mRNA. ($n = 3$; biological replicates) for the "non-organized DHPR dots" category $P = 0.0007$ and for the "transversal DHPR dots" category $P = 0.0015$. Statistical analysis performed using unpaired $t$ tests where ***$P < 0.001$, **$P < 0.01$. Boxplot whiskers represent the maximum and minimum data values. Center lines show the medians; box limits indicate the 25th and 75th percentiles as determined by R software and represents the middle 50% of observed values. (C) Quantification of the mean distances between adjacent DHPR striations in *Tibialis Anterior* extracted muscle fibers from WT mice injected with either PBS or shRNA targeting *SH3KBP1* mRNA. ($n = 3$; biological replicates) with 166 measurements in scramble condition and 165 measurements in *Sh3kbp1* shRNA condition). $P = 4\ E^{-27}$. Error bars represent SD. Statistical analysis performed using unpaired $t$ tests where ***$P < 0.001$. (D) Quantification of the distances between nearest myonuclei in *Tibialis Anterior* extracted muscle fibers from WT mice injected with either PBS or shRNA targeting *SH3KBP1* mRNA. (E, F) Half-activation (E) and steepness factor (F) for SR Ca²⁺ release in the two groups of fibers, as assessed from Boltzmann fits to data from each fiber. Boxplot whiskers represent the maximum and minimum data values. Center lines show the medians; box limits indicate the 25th and 75th percentiles as determined by R software and represents the middle 50% of observed values.

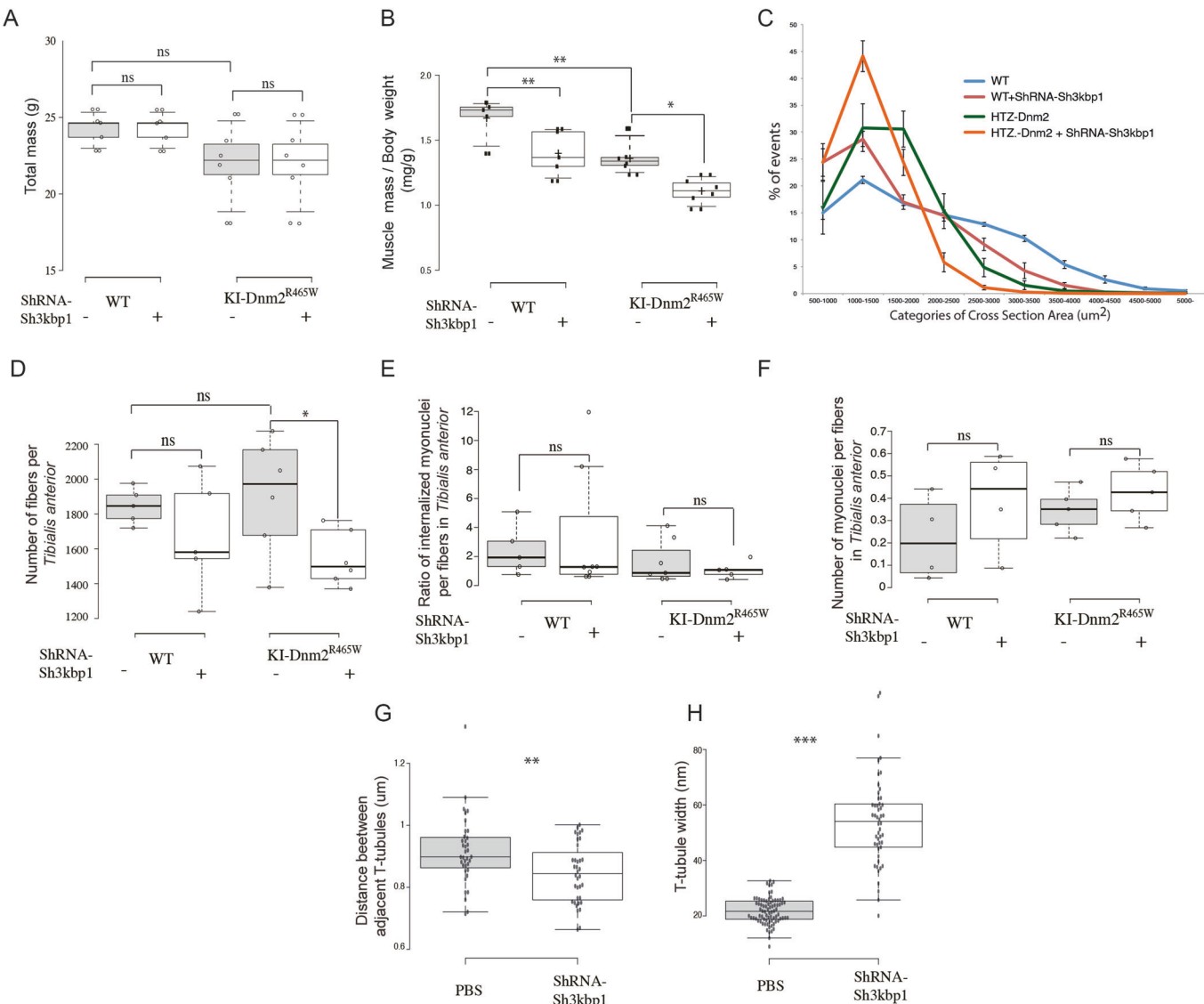

**Figure EV5.** *Sh3kbp1* silencing affects myofibers parameters in WT or in KI-*Dnm2*R465W/+ mice model.

(**A**) Quantification of total body mass from WT or KI-*Dnm2*R465W/+ mice at the age of 4 months, injected with either PBS or AAV cognate vector expressing shRNA targeting SH3KBP1 mRNA (AAV-SH3KBP1) for 3 months. Number of mice per group: $n = 8$. (**B**), Quantification of muscle mass normalized to respective mice body weight from WT or KI-*Dnm2*R465W/+ mice injected with either PBS or AAV-SH3KBP1 for 3 months. ($n > 7$; biological replicates) Comparison between WT and WT-ShRNA-*sh3kbp1* $P = 0.003$, WT and KI-DNM2R465W $P = 0.004$, KI-DNM2R465W and KI-DNM2R465W-ShRNA-*sh3kbp1* $P = 0.01$. Statistical analysis performed using unpaired *t* tests where \*\**P* < 0.01 and \**P* < 0.05. (**C**) Distribution of cross-sectional myofibers areas in *Tibialis Anterior* muscles from WT or KI-*Dnm2*R465W/+ mice injected with either PBS or AAV-SH3KBP1 for 3 months. ($n = 4$; biological replicates) Error bars represent SD. (**D**) Quantification of the number of fibers per *Tibialis Anterior* muscles from WT or KI-*Dnm2*R465W/+ mice injected with either PBS or AAV-SH3KBP1 for 3 months. ($n > 5$; biological replicates) Statistical analysis performed using unpaired *t* tests where \**P* < 0.05. (**E**) Quantification of internalized myonuclei in *Tibialis Anterior* muscles myofibers from WT or KI-*Dnm2*R465W/+ mice injected with either PBS or AAV-SH3KBP1 for 3 months. ($n > 5$; biological replicates). (**F**) Quantification of the number of myonuclei per fibers in *Tibialis Anterior* muscles myofibers from WT or KI-*Dnm2*R465W/+ mice injected with either PBS or AAV-SH3KBP1 for 3 months. ($n > 5$; biological replicates). (**G, H**) Quantification of the distances between adjacent T-tubules structures in *Tibialis Anterior* muscles extracted from WT or KI-*Dnm2*R465W/+ mice injected with either PBS or AAV-SH3KBP1 for 3 months. $P = 0.004$. Two mice were combined for each condition ($n = 36$ measurements per condition), Statistical analysis performed using unpaired *t* tests where \*\**P* < 0.01. (**G**) Quantification of individual T-tubule width in *Tibialis Anterior* muscles extracted from WT or KI-*Dnm2*R465W/+ mice injected with either PBS or AAV-SH3KBP1 for 3 months. $P = 1{,}6\ E^{-34}$. Two mice were combined for each condition ($n = 79$ measurements in Control condition and $n = 50$ in AAV-SH3KBP1 condition) Statistical analysis performed using unpaired *t* tests where \*\*\**P* < 0.001. (**A, B, D–H**) Boxplot whiskers represent the maximum and minimum data values. Center lines show the medians; box limits indicate the 25th and 75th percentiles as determined by R software and represents the middle 50% of observed values.

