## [Peer Review File · EMBO Reports]

SH3KBP1 promotes skeletal myofiber formation and functionality through ER/SR architecture integrity

Alexandre Guiraud, Nathalie Couturier, Emilie Christin, Léa Castellano, Marine Daura, Carole Kretz-Remy, Alexandre Janin, alireza ghasemizadeh, Peggy Del Carmine, Laloé Monteiro, Ludivine ROTARD, Colline Sanchez, Vincent Jacquemond, Claire Burny, Stéphane Janczarski, Anne-cécile Durieux, David Arnould, Norma Beatriz Roméro, Mai Thao Bui, Vladimir Buchman, Laura Julien, Marc Bitoun, and vincent GACHE

Corresponding author(s): vincent GACHE (vincent.gache@inserm.fr)

Review Timeline:

Submission Date:	6th May 20
Editorial Decision:	19th Jun 20
Appeal Received:	7th Mar 24
Editorial Decision:	2nd Jul 24
Revision Received:	19th Dec 24
Editorial Decision:	6th Feb 25
Revision Received:	13th Feb 25
Accepted:	18th Feb 25

Transaction Report:

Dear Dr. GACHE

Thank you for the submission of your research manuscript to EMBO reports. We have now received the two enclosed reports on it.

I am sorry to say, that the evaluation of your manuscript is not a positive one. As you will see, although both referees acknowledge that the findings are potentially interesting, they raise important, partially overlapping criticisms of the manuscript and consider the data insufficient to support the main conclusions. The referees are concerned that the effect of SH3KBP1 on myonuclear positioning is only observed in cell culture but not in vivo, questioning the physiological relevance of this finding. Moreover, the referees are concerned that the role of SH3KBP1 in CNM pathogenesis is not sufficiently supported by the data presented.

Given these concerns, the amount of work required to address them, the uncertain outcome of these experiments, and the fact that EMBO reports can only invite revision of papers that receive enthusiastic support from a majority of referees, I am sorry to say that we cannot offer to publish your manuscript.

I am sorry to disappoint you on this occasion, and hope that the referee comments will be helpful in your continued work in this area.

Yours sincerely

Martina Rembold, PhD
Editor
EMBO reports

Referee #1:

Myonuclei positioning is an important but not well-understood question in skeletal muscle field, which is directly linked to muscle function and diseases. In this manuscript, the authors performed a siRNA screen for new factors that affect myonuclear spreading, and identified SH3KBP1 as a novel regulator of myonuclear positioning. Knockdown of SH3KBP1 resulted in an increase in myonuclear spreading along myotubes and an increase in total number of myonuclei in myotubes. The authors then went on to perform a series of experiments to study SH3KBP1's subcellular location, interacting partners, and its involvement in T-tubules. Lastly, the authors showed that SH3KBP1 is upregulated in a mouse model of AD-CNM and its knockdown via AAV injection affected muscle histology and function. This is an interesting study, as it provided novel insights into the mechanism of myonuclei positioning. However, there are several major concerns of the study that should be addressed.

1. SH3KBP1 knockdown in primary myoblasts resulted in a significant decrease in myofiber width (Figs. 1C, 2G and 2I). However, in C2C12 cells, knockdown resulted in dramatic increases in myofiber width (Fig. 3E). How can the same knockdown have opposite effects?
2. The conclusion that "SH3KBP1 is required in mature fibers for the maintenance of T-tubules" should be supported by electron microscopic images demonstrating defective T-tubules and triads in SH3KBP1 depleted myofibers. Total T-tubule numbers and T-tubule numbers per Z-line should be quantified. T-tubules control excitation-contraction coupling and muscle contraction. Therefore, it is necessary to show that EC coupling is affected in SH3KBP1 depleted myofibers. Does knockdown affect expression of proteins such as DHPR α , RyR1, and others that are involved calcium homeostasis? Since DHPR showed abnormal expression pattern in SH3KBP1 depleted myofibers, is its function in regulating calcium channel affected? This should be addressed by measuring calcium current in single myofibers. DHPR staining in SH3KBP1 depleted myofibers is not clear in demonstrating abnormal expression pattern (Fig. 6G), please provide images with better resolution.
3. The authors showed in SH3KBP1 expression is dramatically increased in a mouse model of AD-CNM (KI-Dnm2R465W/+). However knockdown of SH3KBP1 in this mouse model did not rescue the CNM phenotype. Instead knockdown resulted in further decreases in muscle mass, fiber area, and muscle force compared to KI-Dnm2R465W/+ mice, without affecting the number of centralized nuclei. These results mean that SH3KBP1 function in the muscle is unrelated to CNM, at least the upregulation of SH3KBP1 is not the cause of CNM. The authors argued that this is a compensatory mechanism, but if one wants to prove that SH3KBP1 plays a role in CNM, should it be necessary to either rescue the disease or recapitulate the disease phenotype by manipulating SH3KBP1 levels in mice?
4. SH3KBP1 expression level is decreased in aged mice (Fig. 7A). The question is whether aged mice showed abnormal myonuclear positioning. If yes, this will provide physiological implications of the in vitro cell culture studies.

5. It is difficult to appreciate the subcellular localization of SH3KBP1 in vivo (Fig. 5). The authors described "a cage around myonuclei" in Fig. 5B, but with the magnification and image quality, it is unclear what the authors meant. Along the same line, Fig. 2E also needs zoom-in images.

6. Fig. S2F and S2G are mentioned in the text, but missing in the figures.

Referee #2:

The authors set out to identify factors that regulate myonuclear positioning, which is an important and unknown aspect of skeletal muscle. They do not present the details of the screen but nonetheless decide to further study Sh3kbp1. Reduction of Sh3kbp1 in culture increases the fusion index and impacts myonuclear positioning in cultured cells. However, in vivo there is no impact on these indices, at least based on conditions used in the experiments presented. They then further investigate the role for Sh3kbp1 in the maintenance of T-tubules and see effects in vitro and in vivo. Overall, I think it is clear that Sh3kbp1 plays a role in muscle integrity but based on the data presented it is unlikely to play that role through effects on myonuclear positioning, as posited in the first half of the paper. I think the paper could be improved by focusing on the in vivo experiments to guide major interpretations, or it is possible that new in vivo experiments (shRNA treatment during development or regeneration in vivo) could be designed to potentially reveal an effect on myonuclear positioning in vivo.

1. The authors mention a 'large siRNA screen' and 'candidate genes' but only show data for sh3kbp1. It is not clear why this particular gene was selected, and the authors should consider showing the data from the screen. The design of the screen would help the reader understand if it is identifying genes directly regulating myonuclear positioning or indirectly regulating positioning due to an independent effect on fusion or another myogenic process. Moreover, a positive control (a factor known to regulate myonuclear positioning) should be employed to validate the screen.

2. In their interpretation of figure 1, the authors indicate that sh3kbp1 strongly modifies myonuclear positioning. How can they be sure that myonuclear position is altered and not fusion, which then secondarily impacts nuclear position?

3. From these experiments, I don't think it can be concluded that SH3KBP1 acts as an 'anti-elongation' factor. Since the design of the system has sh3kbp1 being reduced in both myoblasts and myotubes, the precise reason for an altered MSG cannot be determined. It is possible that the KO myotubes are constantly searching for new fusing partners, which impacts the observed MSG. The authors should consider an inducible shRNA plasmid where sh3kbp1 would be reduced in myotubes after formation.

4. Could the result of increased nuclear movement after sh3kbp1 reduction be due to altered actin-cytoskeleton? One way to test this is to test if manipulation of other regulators of actin-cytoskeleton also result in a similar phenotype.

5. The C2C12 experiments in Fig. 4 suffer from the same interpretation issues as the primary myoblasts. It is unclear whether the effect is directly on fusion or directly on nuclear positioning.

6. In the experiments described in Fig. 6D, what is the knockdown of Sh3kbp1 achieved with shRNA in vivo?

7. The experiments and interpretations within the manuscript are contradictory. Initial interpretations are that Sh3kbp1 regulates myonuclear positioning and/or fusion but the in vivo results presented in Fig. S3 do not support those interpretations as there is no nuclear abnormality found in vivo. Thus, the conclusion for the role of Sh3kbp1 is not clear.

8. The writing needs to be improved, especially the sentence structure and grammar. For instance, lines 38-38 says 'early phases of myofibers formation'. Should say 'early phases of myofiber formation' or 'early phases of formation of myofibers'. There are many of issues throughout the text.

** As a service to authors, EMBO Press provides authors with the ability to transfer a manuscript that one journal cannot offer to publish to another journal, without the author having to upload the manuscript data again. To transfer your manuscript to another EMBO Press journal using this service, please click on Link Not Available

Pathophysiology & Genetics of Neuron and Muscle

Dr. Vincent GACHE, PhD, CRCN Inserm

Principal Investigateur INSERM chez INMG-PGNM (INMG)

Équipe « Muscle Nuclear & Cytoskeleton Architecture » Équipe Dr. Gache

Email : vincent.gache@inserm.fr

Tel : +33 4 78 77 70 31- Cellular phone : + 33 6 83 27 83 16

Martina Rembold
Editor EMBO reports

Lyon, March 7 2024

Dear editor,

We are pleased to send you a revised version of our manuscript “SH3KBP1 controls skeletal myofibers formation and functionality through ER/SR architecture integrity” for publication in EMBO reports (EMBOR-2020-50819-T) (related preprint in bioRxiv, doi:10.1101/2020.05.04.076208).

We have addressed 100% of reviewer’s requests and the new data generated further strengthen and consolidate the conclusions of the study. In the first version of the manuscript, we showed that SH3KBP1 contribute, in the muscle fibers, to myonuclei spreading and ER architecture as well as the formation of transverse (T-) tubules, that contribute to skeletal myofibers homeostasis. The reviewers underlined the novelty of the findings and suggested to consolidate SH3KBP1 role in the modulation of the Excitation-Contraction Coupling (ECC) process.

Thanks to the infection of the *Flexor Digitalis Brevis* mouse muscle with AAVs expressing shRNA targeting *Sh3kbp1* mRNA, we now demonstrated the involvement of SH3KBP1 in the proper ECC process, showing that it regulates voltage-activated SR-Ca²⁺ release. Moreover, these experiments were associated to electron microscopy, showing that the ultrastructure of triads were modified in *Sh3kbp1* depleted conditions, reinforcing the ECC impairment discovery. Importantly, our new data demonstrated a new implication of SH3KBP1 in the myofiber integrity by controlling both endoplasmic and sarcoplasmic reticulum’s behaviors. Finally, as requested by the reviewers, we provided the complete siRNA screen results, performed on primary murine myotubes, measuring the involvement of 30 candidates on myotubes elongation and myonuclei spreading, thus giving new additional information’s on their respective role, especially during early steps of muscle fibers formation.

To our knowledge, our work still describes for the first time the implication of SH3KBP1 protein in the muscle field and show that acting on the ER/SR architecture formation and maintenance impact myoblast fusion and myonuclei movement capacity in vertebrate myofibers. Finally, our data suggest that SH3KBP1 is an anti-atrophic muscle protein and it raise the potentially to be used as a preventing factor in the development of myopathies associated with progressive muscle atrophy as in congenital centronuclear myopathies, Limb-Girdle Muscular Dystrophy and Sarcopenia.

Altogether, our data suggest a multimodal implication of SH3KBP1 in the formation and functionality of post-mitotic cells, that we feel will be of interest for a large audience of researchers interested in cytoskeleton, nucleus, ER/SR formation/maintenance and ECC process efficiency during muscle development.

We feel that our revised manuscript now complies with EMBO Reports requirements and we hope that you will find the novel conclusions suitable for publication in EMBO Reports.

I remain at your disposal for any further information needed.

Vincent GACHE

We first would like to thank the referees for their constructive comments that we addressed in the resubmitted manuscript. We adapted our manuscript to take into account the referees' comments and according to the format of EMBO reports.

Please find below a point-by-point response to the referees' comments:

Referee #1:

Myonuclei positioning is an important but not well-understood question in skeletal muscle field, which is directly linked to muscle function and diseases. In this manuscript, the authors performed a siRNA screen for new factors that affect myonuclear spreading, and identified SH3KBP1 as a novel regulator of myonuclear positioning. Knockdown of SH3KBP1 resulted in an increase in myonuclear spreading along myotubes and an increase in total number of myonuclei in myotubes. The authors then went on to perform a series of experiments to study SH3KBP1's subcellular location, interacting partners, and its involvement in T-tubules. Lastly, the authors showed that SH3KBP1 is upregulated in a mouse model of AD-CNM and its knockdown via AAV injection affected muscle histology and function. This is an interesting study, as it provided novel insights into the mechanism of myonuclei positioning. However, there are several major concerns of the study that should be addressed.

→ We thank the referee for these positive comments.

"1. SH3KBP1 knockdown in primary myoblasts resulted in a significant decrease in myofiber width (Figs. 1C, 2G and 2I). However, in C2C12 cells, knockdown resulted in dramatic increases in myofiber width (Fig. 3E). How can the same knockdown have opposite effects?"

→ There is actually no opposite effect as *Sh3kbp1* knockdown significantly enhances the fusion index of both C2C12 and primary myoblasts. As quantified in Figure 2 of the updated manuscript, the knock-down of *Sh3kbp1* in primary myoblasts significantly increases their fusion index, the average number of nuclei per myotubes and the percentage of myotubes with more than 10 nuclei. Similarly, C2C12 cells that express constitutively a shRNA targeting *Sh3kbp1* mRNA, exhibit an enhancement in the number of myonuclei per myotubes (Figure PBP-1) and the number of myonuclei in clusters (Figure 3D-H).

Figure PBP-1 Distribution of 3 days differentiated C2C12 myotubes formed by differentiation of stable C2C12 cell line expressing either scramble or *Sh3kbp1* shRNAs and classified according to nuclei content inside myotubes, after 5 days of differentiation.

However, we would like to highlight that the fusion kinetics of primary myoblasts and C2C12 cells are dramatically different. Indeed, while C2C12 start to fuse 3 days after differentiation induction (Figure 3E), primary myoblasts are *per se* “ready to fuse” myoblasts and rapidly form multinucleated myotubes in less than one day after differentiation induction, before myofibrils genesis begin (Ghasemizadeh *et al*, 2021; Falcone *et al*, 2014; Cadot *et al*, 2012). Thus, a decrease in primary myotube width is observed in the first day of differentiation, mainly resulting from new forces applied at the tips of myotubes due to the failure in the endoplasmic reticulum organization, as discussed in our revised article (line 377-401). However, this parameter is difficult to appreciate in C2C12 as myonuclei rapidly form clusters that directly impact myotube width parameters. The same phenomenon on width myofibers appears in primary cells, depending on the presence of myonuclei clusters.

To avoid any misunderstanding on the real effect of SH3KBP1 on myotube elongation, we added SH3KBP1 effect on fusion capabilities as a main panel of Figure 2. The analysis of primary myofibers focuses now only on the distance between myonuclei and is presented in supplementary data (Figure Supplement 1D-F).

2. “The conclusion that “SH3KBP1 is required in mature fibers for the maintenance of T-tubules” should be supported by electron microscopic images demonstrating defective T-tubules and triads in SH3KBP1 depleted myofibers. Total T-tubule numbers and T-tubule numbers per Z-line should be quantified. T-tubules control excitation-contraction coupling and muscle contraction. Therefore, it is necessary to show that EC coupling is affected in SH3KBP1 depleted myofibers. Does knockdown affect expression of proteins such as DHPRalpha, RyR1, and others that are involved calcium homeostasis? Since DHPR showed abnormal expression pattern in SH3KBP1 depleted myofibers, is its function in regulating calcium channel affected? This should be addressed by measuring calcium current in single myofibers. DHPR staining in SH3KBP1 depleted myofibers is not clear in demonstrating abnormal expression pattern (Fig. 6G), please provide images with better resolution”.

→ We agree with the comment of the referee and we therefore included in the manuscript:

- Electron microscopy images allowing the visualization of the myofibrils and triads organization in *Flexor digitoralis brevis* muscles from WT mice injected with either PBS or AAV cognate vector expressing shRNA targeting *SH3KBP1* mRNA (Figure 8G).
- High magnification images of T-tubule organization (Figure 6F) and quantification of the distance between DHPR striations (supplementary figure 3).
- *In vivo* analysis of the excitation contraction coupling (ECC) in mature myofibers (Figure 6F-G and supplementary figure 3E-F).
- Co-immunolabeling of SH3KBP1 and DHPR α in isolated mature myofibers (Figure 5F).

These additional data strengthen the conclusion that SH3KBP1 is required for T-tubules formation/maintenance and muscle excitability.

3. “The authors showed in SH3KBP1 expression is dramatically increased in a mouse model of AD-CNM (KI-Dnm2R465W/+). However knockdown of SH3KBP1 in this mouse model did not rescue the CNM phenotype. Instead knockdown resulted in further decreases in muscle mass,

fiber area, and muscle force compared to KI-Dnm2^{R465W/+} mice, without affecting the number of centralized nuclei. These results mean that SH3KBP1 function in the muscle is unrelated to CNM, at least the upregulation of SH3KBP1 is not the cause of CNM. The authors argued that this is a compensatory mechanism, but if one wants to prove that SH3KBP1 plays a role in CNM, should it be necessary to either rescue the disease or recapitulate the disease phenotype by manipulating SH3KBP1 levels in mice?"

→ We respectfully disagree with the referee as we did not propose that SH3KBP1 upregulation could be causal of CNM or knockdown could rescue the phenotype.

We observed an increase of *SH3KBP1* mRNA during the 4 first months of life in AD-CNM (KI-Dnm2^{R465W/+}) mouse models (Figure 7A). We thus hypothesized that *SH3KBP1* up-regulation limits the activation/progression of the CNM related-pathways in the AD-CNM (KI-Dnm2^{R465W/+}) mouse models and thus, prevents the appearance of CNM phenotypes.

In both control and KI-Dnm2^{R465W/+} mouse models, SH3KBP1 protein depletion triggers a decrease in the total muscle mass, the myofiber cross-sectional area, the muscle force associated with an increase in the proportion of fibers with high content of autophagosomes (LC3 positives) and slow fibers type (MHC1 positives) (Figure 7D-H & 8H). Therefore, SH3KBP1 depletion rather dampens than rescues the muscle function and *SH3KBP1* up-regulation during the first months of postnatal and adult development may have a beneficial effect regarding the evolution of the human CNM pathology.

Although we did not observe a drastic alteration of myonuclei localization after 3-4 months of *Sh3kbp1* down-regulation in the *Tibialis anterior* muscle fibers, we found that nearly 15% of myonuclei were present in myonuclei cluster in *Sh3kbp1* depleted conditions compared to controls (Supplementary Figure 3D).

Regarding myonuclei internalization in humans, the Dnm2^{R465W/+} mutation leads to the accumulation of myofibers with centrally located myonuclei to above 60% of the total myofibers (Mori-Yoshimura *et al*, 2012). However, the KI-Dnm2^{R465W/+} mouse model does not recapitulate the centralization phenotype observed in CNM human tissue samples ((Durieux *et al*, 2010) and Supplementary Figure 4E).

Our results suggest that SH3KBP1 is not involved in the peripheral myonuclei anchoring pathways in mature myofibers but rather in myonuclei spreading, directly in the early steps of myofiber formation. Of note, we recently attributed myonuclei internalization to the modulation of the microtubule network stability (Ghasemizadeh *et al*, 2021).

4." SH3KBP1 expression level is decreased in aged mice (Fig. 7A). The question is whether aged mice showed abnormal myonuclear positioning. If yes, this will provide physiological implications of the in vitro cell culture studies".

→ We thank the reviewer for this excellent comment, as aged mice progressively display central/disorganized myonuclei in myofibers, especially in sarcopenia conditions (Brooks *et al*, 2009; Malatesta *et al*, 2009). Moreover, Murgia *et al* also observed a downregulation of SH3KBP1 in aging human muscles, especially in fast fibers (Murgia *et al*, 2017). However, the role of SH3KBP1 in the maintenance of myonuclei peripherization in mature myofibers seems very limited in our *in vivo*

experiments (Supplementary figures 3D & 4E). Moreover, from our point of view, it seems difficult to argue in favor of a precocious aging of our *in vitro* muscle fibers with regard to our results. Nevertheless, this long-term effect of SH3KBP1 in aging muscle will certainly require future investigations.

5. "It is difficult to appreciate the subcellular localization of SH3KBP1 in vivo (Fig. 5). The authors described "a cage around myonuclei" in Fig. 5B, but with the magnification and image quality, it is unclear what the authors meant. Along the same line, Fig. 2E also needs zoom-in images".

→ We agree with this comment and we have now included in the manuscript a clear image of the co-staining of SH3KBP1 and DHPR α in isolated mature myofibers (Figure 5F in the new manuscript). This staining around myonuclei is also found in transversal sections of human muscle samples (Supplementary figure 2). We also provided a zoom-in for the image of Figure 2E that is now relocated to figure 4C-F of the new manuscript.

6. "Fig. S2F and S2G are mentioned in the text, but missing in the figures".

→ We apologize for this oversight. These figures have been now relocated in figure 2F-G.

Referee #2:

The authors set out to identify factors that regulate myonuclear positioning, which is an important and unknown aspect of skeletal muscle. They do not present the details of the screen but nonetheless decide to further study Sh3kbp1. Reduction of Sh3kbp1 in culture increases the fusion index and impacts myonuclear positioning in cultured cells. However, in vivo there is no impact on these indices, at least based on conditions used in the experiments presented. They then further investigate the role for Sh3kbp1 in the maintenance of T-tubules and see effects in vitro and in vivo. Overall, I think it is clear that Sh3kbp1 plays a role in muscle integrity but based on the data presented it is unlikely to play that role through effects on myonuclear positioning, as posited in the first half of the paper. I think the paper could be improved by focusing on the in vivo experiments to guide major interpretations, or it is possible that new in vivo experiments (shRNA treatment during development or regeneration in vivo) could be designed to potentially reveal an effect on myonuclear positioning in vivo.

→ We thank the reviewer for these positive comments. We reformatted the paper in accordance to the referee wishes, notably by adding new *ex-vivo* experiments demonstrating the involvement of SH3KBP1 in muscle excitability (see also answer to reviewer 1, question 2).

1. "The authors mention a 'large siRNA screen' and 'candidate genes' but only show data for sh3kbp1. It is not clear why this particular gene was selected, and the authors should consider showing the data from the screen. The design of the screen would help the reader understand if it is identifying genes directly regulating myonuclear positioning or indirectly regulating positioning due to an independent effect on fusion or another myogenic process. Moreover, a

positive control (a factor known to regulate myonuclear positioning) should be employed to validate the screen”.

→ The referee suggestion is very good. We thus included in the new manuscript the siRNA screen that we performed (Figure 1A-B, table 1-3), and commented on the selection of SH3KBP1 as the best candidate (lines 94-132).

2. “In their interpretation of figure 1, the authors indicate that sh3kbp1 strongly modifies myonuclear positioning. How can they be sure that myonuclear position is altered and not fusion, which then secondarily impacts nuclear position?”

→ SH3KBP1 functions on both myonuclear positioning (Figure 1,3-4) and myoblast fusion (see figure 2 of the new manuscript and Figure PBP-1 above). To specifically decipher the effect of SH3KBP1 on myonuclei positioning and to discard any possible effect of the fusion process, we classified primary myotubes according to their myonuclei content (Figure supplement 1B-C). Thus, myotubes sharing the same low myonuclei content can be analyzed, allowing to exclude fusion-related effects. We observed that myotube length and myonuclei spreading are directly affected by SH3KBP1 over-expression in myotubes containing 3-4 myonuclei (Figure supplement 1B-C). We thus concluded that although SH3KBP1 acts on the fusion capacity, myonuclei mispositioning in developing myotubes occurs first and independent of the fusion process and will further contribute to aggravate the long-term myonuclei clustering (Figure 3D-H).

3. “From these experiments, I don't think it can be concluded that SH3KBP1 acts as an 'anti-elongation' factor. Since the design of the system has sh3kbp1 being reduced in both myoblasts and myotubes, the precise reason for an altered MSG cannot be determined. It is possible that the KO myotubes are constantly searching for new fusing partners, which impacts the observed MSG. The authors should consider an inducible shRNA plasmid where sh3kbp1 would be reduced in myotubes after formation”

→ We would like to mention that our experimental set up actually allows for analyzing Sh3kbp1 knockdown in developing myotubes with shRNA (Figure 4). Indeed, primary myoblasts are transfected the day of differentiation induction, right at the time of myoblast fusion (Figure 4A). Thus, Sh3kbp1 shRNA expression will occur during differentiation, specifically downregulating Sh3kbp1 in myotubes. In this condition, we also found aggregation of myonuclei (Figure 4C-F and figure supplement 1E-G), associated with an increase in myonuclei velocity and time in motion (Figure 4 G-I).

4. “Could the result of increased nuclear movement after sh3kbp1 reduction be due to altered actin-cytoskeleton? One way to test this is to test if manipulation of other regulators of actin-cytoskeleton also result in a similar phenotype”.

→ We already address this point in a previous article (Gache *et al*, 2017), in which we showed that precocious myonuclei spreading primarily relies on the microtubule network (Figure PBP-2).

Figure 1B from Gache V et al., Mol Biol Cell 2017

5. “The C2C12 experiments in Fig. 4 suffer from the same interpretation issues as the primary myoblasts. It is unclear whether the effect is directly on fusion or directly on nuclear positioning”.

→ We thank the reviewer for this comment. This figure is now relocated in figure 8A-B in the new manuscript. The aim of this experiment is to address which compartment is controlled by SH3KBP1; here we clearly show that Golgi apparatus is not affected while ER is dispersed. To our knowledge and our experience, increasing fusion as never been so far linked to ER disorganization.

6. “In the experiments described in Fig. 6D, what is the knockdown of *Sh3kbp1* achieved with *shRNA in vivo*?”

→ Plasmid electroporation only affects a few myofibers and does not permit to measure precisely the knockdown level of *Sh3kbp1*. Therefore, we replaced this experiment for an analysis performed following intramuscular injections of AAV-sh*Sh3kbp1* that provide a much stronger and significant knockdown. Downregulation levels are presented in Figure 7C.

7. “The experiments and interpretations within the manuscript are contradictory. Initial interpretations are that *Sh3kbp1* regulates myonuclear positioning and/or fusion but the *in vivo* results presented in Fig. S3 do not support those interpretations as there is no nuclear abnormality found *in vivo*. Thus, the conclusion for the role of *Sh3kbp1* is not clear”.

→ We hope the revised version of our manuscript will clarify our conclusions. In brief, our experiments show that in the early phases of myofiber formation, SH3KBP1, through ER-scaffolding, controls myonuclei motion and positioning. However, in mature myofibers (already formed myofibers), SH3KBP1 is not the main protein involved in the modulation of myonuclei anchoring at the periphery of muscle fibers. Therefore, SH3KBP1 acts in a more complex interplay of proteins during the maturation process of myofibers and to control myonuclei positioning.

8. “The writing needs to be improved, especially the sentence structure and grammar. For instance, lines 38-38 says 'early phases of myofibers formation'. Should say 'early phases of myofiber formation' or 'early phases of formation of myofibers'. There are many of issues throughout the text”.

→ We apologize for this and we asked for external proofreading to improve the grammar of our revised manuscript.

References

- Brooks NE, Schuenke M & Hikida R (2009) Ageing influences myonuclear domain size differently in fast and slow skeletal muscle of rats. *Acta physiologica (Oxford, England)* 197: 55–63
- Cadot B, Gache V, Vasyutina E, Falcone S, Birchmeier C & Gomes ER (2012) Nuclear movement during myotube formation is microtubule and dynein dependent and is regulated by Cdc42, Par6 and Par3. *EMBO reports* 13: 741–749
- Durieux A-C, Vignaud A, Prudhon B, Viou M, Beuvin M, Vassilopoulos S, Fraysse B, Ferry A, Lainé J, Romero NB, *et al* (2010) A centronuclear myopathy-dynamamin 2 mutation impairs skeletal muscle structure and function in mice. *Human molecular genetics* 19: 4820–4836
- Falcone S, Roman W, Hnia K, Gache V, Didier N, Lainé J, Auradé F, Marty I, Nishino I, Charlet-Berguerand N, *et al* (2014) N-WASP is required for Amphiphysin-2/BIN1-dependent nuclear positioning and triad organization in skeletal muscle and is involved in the pathophysiology of centronuclear myopathy. *EMBO molecular medicine* 6: 1455–1475
- Gache V, Gomes E & Cadot B (2017) Microtubule motors involved in nuclear movement during skeletal muscle differentiation. *Molecular biology of the cell: mbc*.E16-06-0405
- Ghasemizadeh A, Christin E, Guiraud A, Couturier N, Abitbol M, Risson V, Girard E, Jagla C, Soler C, Laddada L, *et al* (2021) MACF1 controls skeletal muscle function through the microtubule-dependent localization of extra-synaptic myonuclei and mitochondria biogenesis. *Elife* 10: e70490

Malatesta M, Perdoni F, Muller S, Zancanaro C & Pellicciari C (2009) Nuclei of aged myofibres undergo structural and functional changes suggesting impairment in RNA processing. *European journal of histochemistry : EJH* 53: 97–106

Mori-Yoshimura M, Okuma A, Oya Y, Fujimura-Kiyono C, Nakajima H, Matsuura K, Takemura A, Malicdan MCV, Hayashi YK, Nonaka I, *et al* (2012) Clinicopathological features of centronuclear myopathy in Japanese populations harboring mutations in dynamin 2. *Clin Neurol Neurosurg* 114: 678–683

Murgia M, Toniolo L, Nagaraj N, Ciciliot S, Vindigni V, Schiaffino S, Reggiani C & Mann M (2017) Single Muscle Fiber Proteomics Reveals Fiber-Type- Specific Features of Human Muscle Aging. *Cell reports* 19: 2396–2409

Dear Dr. GACHE

Thank you for the submission of your research manuscript to our journal. I apologize for the delay in handling your manuscript. Unfortunately, the referees who had evaluated your manuscript when it was originally submitted to EMBO Reports in 2020 were not available anymore, except for former referee #1. I therefore asked two additional referees to review your manuscript taking your response to the previous referee concerns into account. As you will see, former referee #1 considers your response to his/her concerns adequate. Referee #5 considers your response to the previous concerns overall valid but also comments on the missing analysis of microtubules in the SH3KBP1-mediated process. Referee #4 also acknowledges that you invested a lot of work in the revised manuscript but considers some conclusions not sufficiently supported by the data. Similar to referee #5 the reviewer raises concerns regarding the involvement of the MT network in nuclear spreading. Given that 2 referees raised a similar concern, this aspect should be addressed experimentally. The effect of SH3KBP1 depletion on DNM2 and vice versa, the effect of Dnm2 mutation on SH3KBP1, should be analysed as well. The lasting effect of Sh3kbp1 depletion in myofibers at Day 10 needs to be tested. EM images need to be quantified and the questions on autophagic flux need to be answered as indeed, accumulation of LC3 alone is not sufficient to conclude on flux. Please address all other concerns either experimentally or textual. All minor requests need to be addressed as well.

Given these constructive comments, we would like to give you the opportunity you to revise your manuscript for potential publication at EMBO Reports, with the understanding that the referee concerns (as detailed above and in their reports) must be fully addressed and their suggestions taken on board. I feel that the experiments required are feasible but please let me know whether you want to pursue this revision at our journal. If so, please address all referee concerns in a complete point-by-point response. Acceptance of the manuscript will depend on a positive outcome of a second round of review. The manuscript will be seen again by referee #4. It is EMBO Reports policy to allow a single round of revision only and acceptance or rejection of the manuscript will therefore depend on the completeness of your responses included in the next, final version of the manuscript.

We realize that it is difficult to revise to a specific deadline. In the interest of protecting the conceptual advance provided by the work, we recommend a revision within 3 months (October 2nd). Please discuss the revision progress ahead of this time with the editor if you require more time to complete the revisions.

I am also happy to discuss the revision further via e-mail or a video call, if you wish.

*****IMPORTANT NOTE:

We perform an initial quality control of all revised manuscripts before re-review. Your manuscript will FAIL this control and the handling will be delayed IN CASE the following APPLIES:

- 1) A data availability section providing access to data deposited in public databases is missing. If you have not deposited any data, please add a sentence to the data availability section that explains that.
- 2) Your manuscript contains statistics and error bars based on $n=2$. Please use scatter blots in these cases. No statistics should be calculated if $n=2$.

When submitting your revised manuscript, please carefully review the instructions that follow below. Failure to include requested items will delay the evaluation of your revision.*****

- 1) a .docx formatted version of the manuscript text (including legends for main figures, EV figures and tables). Please make sure that the changes are highlighted to be clearly visible.
- 2) individual production quality figure files as .eps, .tif, .jpg (one file per figure). Please download our Figure Preparation Guidelines (figure preparation pdf) from our Author Guidelines pages <https://www.embopress.org/page/journal/14693178/authorguide> for more info on how to prepare your figures.
- 3) a .docx formatted letter INCLUDING the reviewers' reports and your detailed point-by-point responses to their comments. As part of the EMBO Press transparent editorial process, the point-by-point response is part of the Review Process File (RPF), which will be published alongside your paper.
- 4) a complete author checklist, which you can download from our author guidelines (<<https://www.embopress.org/page/journal/14693178/authorguide>>). Please insert information in the checklist that is also reflected in the manuscript. The completed author checklist will also be part of the RPF.

5) Please note that all corresponding authors are required to supply an ORCID ID for their name upon submission of a revised manuscript (<<https://orcid.org/>>). Please find instructions on how to link your ORCID ID to your account in our manuscript tracking system in our Author guidelines (<<https://www.embopress.org/page/journal/14693178/authorguide#authorshipguidelines>>)

6) We replaced Supplementary Information with Expanded View (EV) Figures and Tables that are collapsible/expandable online. A maximum of 5 EV Figures can be typeset. EV Figures should be cited as 'Figure EV1, Figure EV2' etc... in the text and their respective legends should be included in the main text after the legends of regular figures.

7) Please note that a Data Availability section at the end of Materials and Methods is now mandatory. In case you have no data that requires deposition in a public database, please state so instead of refereeing to the database. See also < <https://www.embopress.org/page/journal/14693178/authorguide#dataavailability>>. Please note that the Data Availability Section is restricted to new primary data that are part of this study.

Additional information on source data and instruction on how to label the files are available <<https://www.embopress.org/page/journal/14693178/authorguide#sourcedata>>.

10) Figure legends and data quantification:
The following points must be specified in each figure legend:

- the name of the statistical test used to generate error bars and P values,
 - the number (n) of independent experiments (please specify technical or biological replicates) underlying each data point,
 - the nature of the bars and error bars (s.d., s.e.m.)
- If the data are obtained from n {less than or equal to} 5, show the individual data points in addition to the SD or SEM.
- If the data are obtained from n {less than or equal to} 2, use scatter blots showing the individual data points.

See also the guidelines for figure legend preparation:
<https://www.embopress.org/page/journal/14693178/authorguide#figureformat>

11) Our journal encourages inclusion of *data citations in the reference list* to directly cite datasets that were re-used and obtained from public databases. Data citations in the article text are distinct from normal bibliographical citations and should directly link to the database records from which the data can be accessed. In the main text, data citations are formatted as follows: "Data ref: Smith et al, 2001" or "Data ref: NCBI Sequence Read Archive PRJNA342805, 2017". In the Reference list, data citations must be labeled with "[DATASET]". A data reference must provide the database name, accession number/identifiers and a resolvable link to the landing page from which the data can be accessed at the end of the reference. Further instructions are available at <<https://www.embopress.org/page/journal/14693178/authorguide#referencesformat>>.

12) All Materials and Methods need to be described in the main text using our 'Structured Methods' format, which is required for all research articles. According to this format, the Methods section includes a Reagents and Tools Table (listing key reagents, experimental models, software and relevant equipment and including their sources and relevant identifiers) followed by a Methods and Protocols section describing the methods using a step-by-step protocol format. The aim is to facilitate adoption of

the methodologies across labs. More information on how to adhere to this format as well as a downloadable template (.docx) for the Reagents and Tools Table can be found in our author guidelines:
<https://www.embopress.org/page/journal/14693178/authorguide#structuredmethods>.

An example of a Method paper with Structured Methods can be found here:
<https://www.embopress.org/doi/10.15252/msb.20178071>.

13) As part of the EMBO publication's Transparent Editorial Process, EMBO Reports publishes online a Review Process File to accompany accepted manuscripts. This File will be published in conjunction with your paper and will include the referee reports, your point-by-point response and all pertinent correspondence relating to the manuscript.

Yours sincerely,

Referee #1:

In this revised manuscript, the authors performed key experiments that were brought up in my previous comments, included images with higher resolutions, reorganized the results and toned down some previous conclusions. These changes resulted in a much-improved manuscript that still holds novelty regarding SH3KBP1 function in myofiber formation. The authors addressed most of my major concerns of the original manuscript. I appreciate authors' responsiveness to my previous comments and their efforts in bringing the story to a higher level. I recommend publication of this beautiful study.

One minor suggestion: in Figure 5D and E, it is better to include the color labeling in the figure panels, as they are different stainings from Figure 5F.

Referee #4:

In the manuscript by Guiraud et al., the authors sought to identify novel factors that might be involved in myonuclear positioning. Using a limited siRNA screen, then identified Sh3kbp1 as a potential candidate. In vitro, depletion of Sh3kbp1 lead to aberrant nuclear positioning, increased myoblast fusion and altered t-tubule formation. In vivo, Sh3kbp1 depletion induced muscle wasting and weakness in both wild-type animals and a mouse model of centronuclear myopathy (Dnm2R465W). While the manuscript contains many interesting findings, the results are somewhat superficial and the link between the different results are not always obvious. For example, are defects in nuclear positioning related to changes in contractility/fiber type/atrophy/autophagy in vivo? If these mechanisms are distinct, my suggestion would be to focus on either the role of SH3KBP1 in nuclear positioning/myogenesis or the role in muscle maintenance/homeostasis. In addition, the role of Sh3kbp1 in ER/SR/triad structure maintenance in vivo still needs additional support. While clearly a lot of work went into the revised manuscript, for some aspects of the manuscript, there are still too strong of conclusions being drawn based on the current data.

Major Comments

- One shortcoming of the paper is that there is no attempt to link the SH3KBP1 to known important regulators of nuclear spreading. Could it be that many of the phenotypes that you observe are due to disruption in the microtubule network? The

authors acknowledge in the discussion that SH3KBP1 may interact with microtubule binding partners, so ruling out the direct involvement of microtubules would be important

- For many outcomes, it is difficult to decouple the changes in fusion from the changes in nuclear spreading. For example, in Fig. Suppl 1G, is this distance reduced because of reduced spreading or are there more nuclei in those myofibers because of enhanced fusion (Fig. 2)? It could be that this reduced distance is because there are simply more nuclei in the Sh3kbp1 depleted myofibers. Further, it would be very informative to perform the DiMycMyo analysis at later time points to show whether the defect in nuclear spreading persists or is just delayed with Sh3kbp1 depletion.

- I'm not sure I agree with the conclusion for Figure 4 that "...SH3KBP1 restrains myonuclei movements and motion during muscle fibers maturation, which contributes to correct myonuclear spreading.", as it appears that nuclear motion is not affected (or even enhanced) with Sh3kbp1 depletion.

- One critical experiment that is missing is whether SH3KBP1 depletion has any functional effect on DNM2. The authors do a very nice job showing that SH3KBP1 interacts with DNM2 in myofibers, but fail to show whether SH3KBP1 depletion affects DNM2 localization. If the proposed mechanism is that SH3KBP1 interacts with DNM2 and this interaction is important for t-tubule formation via DNM2, then the prediction would be that DNM2 localization should be affected with SH3KBP1 depletion. If this is not the case, then it would suggest that SH3KBP1 is involved in t-tubule formation via some other mechanism. Similarly, the authors claim that "Sh3kbp1 exhibits a DNM2-like localization" but do not show any DNM2 immunostaining. Performing these experiments would strengthen the authors proposed mechanism.

- While I appreciate the inclusion of the Ca²⁺ transient studies, the interpretation of the results are difficult for this non-electrophysiologist. Many parameters were not affected by Sh3kbp1 depletion (transient time course, amplitude increase with voltage, mid-activation voltage, and slope factor), except for maximal rate of SR Ca²⁺ release. Is this an expected result if triads are disrupted?

- One thing that is not clear is the lasting effect of Sh3kbp1 depletion when myofibers are matured for 10 days. Since this is a transient transfection early in differentiation, is Sh3kbp1 still depleted at day 10? Added rigor would include showing immunofluorescent staining for Sh3kbp1 in the siRNA depleted fibers at Day 10 (Fig. 6A) to effectively interpret at what point Sh3kbp1 expression is required for t-tubule formation. Currently, Sh3kbp1 depletion is only shown using western blot and the time point is not described

- The rationale for looking at Sh3kbp1 in myopathies caused by Dnm2 is not clear, as it is still not known what Sh3kbp1 depletion does to Dnm2 in vivo, or vice versa. One possibility could be that the Dnm2 mutation affects its interaction with Sh3kbp1. This should be tested, otherwise modulation the expression of Sh3kbp1 does not make sense.

- Again, I'm not sure I understand the conclusion that ".....in pathological context such as in the CNM diseases, SH3KBP1 expression could counteract the progression of related phenotype." SH3KBP1 expression is already elevated in Dnm2R465W mice, and knocking it down actually made the phenotype worse in the mutant mice. So, would the idea be to increase SH3KBP1 levels even further in the diseased context?

- Fig. 8G. While the EM images are the best way to visualize triads, I'm not sure what information should be gathered from just representative EM images, as any differences are not obvious to the untrained eye. Some quantification is warranted.

- The rationale for looking at LC3 is not well supported and the method of quantification is not clear. In theory, every fiber will have some level of LC3+ structures at any time, but conclusions about autophagic flux cannot be determined from a single timepoint. An increase in LC3 could mean either enhanced autophagic flux or impaired flux, depending on what is causing the increase in LC3. Additional studies/control are needed to draw any conclusion about the role of Sh3kbp1 in autophagy.

Minor Comments

- Figure 1B is very difficult to interpret at first glance. I guess it should be "% change relative to scramble" and not "% of scramble", otherwise I would think that scramble would be equal to 100%. I think the way that it is shown now, it is actually calculated as a fold increase not percent increase. For example, based on Table 1, by my calculation Sh3kbp1 has either a 1.48-fold increase or 47.23% increase in length compared to scramble control.

- Similarly, the wording in line 110 and 111 is confusing. I think authors mean that every candidate appears to be a negative regulator of myotube length expansion, as knocking them down lead to an increase in myotube length relative to the scramble control. I suggest rewording for clarity.

- Suppl Fig 1A. - at what time was knockdown assessed?

- In Fig 3H, it is not clear which conditions are being compared for the statistics.

- I suggest adding labels for the different antibodies in Fig 5 D and E, similar to how it is in 5F. It will make the figure easier to follow.

- Perhaps it's because it's a maximum intensity projection, but I do not see SH3KBP1 as a "nuclear cage" in Fig. 5F and Suppl. Fig. 2. It appears to be clearly inside the nucleus in vivo. Do the authors have any explanation for this localization?

- Double check line 207 - it seems some text may be missing.

- How were triads quantified? Was it an automated analysis? This analysis would seem to be highly subjective

- What does GFP represent in Fig 6E? It is not described in the methods whether the AAV plasmid also contains a GFP.

- Similarly, it is difficult to interpret the representative images in 6E without any sarcomeric marker.

- Fig. 7A. Normalizing to Nat10 seems a bit odd. Is there a reason for using this somewhat obscure gene?

This revised manuscript by Guiraud and colleagues provides strong evidence that SH3KBP1 is an adapter protein that has important roles in muscle formation, nuclear positioning, and possibly homeostatic maintenance. This reviewer finds this to be an interesting work, and the authors have responded well to previous reviews.

I only have one comment on the work:

It is interesting that the authors comment on the importance of microtubule organization in muscle, however did not investigate how microtubules are involved in the the SH3KBP1-mediated processes. Particularly in regards to nuclear positioning, the work of Wilson & Holtzbaaur (2012) suggest that microtubule association with the nuclear membrane may be disrupted by SH3KBP1- thus contributing to the observations of this report.

SH3KBP1 controls skeletal myofibers formation and functionality through ER/SR architecture integrity

Submission for publication to Embo Reports

Manuscript ID: EMBOR-2020-50819V2

Thank you for the submission of your research manuscript to our journal. I apologize for the delay in handling your manuscript. Unfortunately, the referees who had evaluated your manuscript when it was originally submitted to EMBO Reports in 2020 were not available anymore, except for former referee #1. I therefore asked two additional referees to review your manuscript taking your response to the previous referee concerns into account. As you will see, former referee #1 considers your response to his/her concerns adequate. Referee #5 considers your response to the previous concerns overall valid but also comments on the missing analysis of microtubules in the SH3KBP1-mediated process. Referee #4 also acknowledges that you invested a lot of work in the revised manuscript but considers some conclusions not sufficiently supported by the data.

Similar to referee #5 the reviewer raises concerns regarding the involvement of the MT network in nuclear spreading. Given that 2 referees raised a similar concern, this aspect should be addressed experimentally.

The effect of SH3KBP1 depletion on DNM2 and vice versa, the effect of Dnm2 mutation on SH3KBP1, should be analysed as well. The lasting effect of Sh3kbp1 depletion in myofibers at Day 10 needs to be tested. EM images need to be quantified and the questions on autophagic flux need to be answered as indeed, accumulation of LC3 alone is not sufficient to conclude on flux. Please address all other concerns either experimentally or textual. All minor requests need to be addressed as well.

Given these constructive comments, we would like to give you the opportunity you to revise your manuscript for potential publication at EMBO Reports, with the understanding that the referee concerns (as detailed above and in their reports) must be fully addressed and their suggestions taken on board. I feel that the experiments required are feasible but please let me know whether you want to pursue this revision at our journal. If so, please address all referee concerns in a complete point-by-point response. Acceptance of the manuscript will depend on a positive outcome of a second round of review. The manuscript will be seen again by referee #4. It is EMBO Reports policy to allow a single round of revision only and acceptance or rejection of the manuscript will therefore depend on the completeness of your responses included in the next, final version of the manuscript.

We realize that it is difficult to revise to a specific deadline. In the interest of protecting the conceptual advance provided by the work, we recommend a revision within 3 months (October 2nd). Please discuss the revision progress ahead of this time with the editor if you require more time to complete the revisions.

I am also happy to discuss the revision further via e-mail or a video call, if you wish.

Responses to the Editor and the reviewers

We would like to express our sincere gratitude to you and the reviewers for dedicating time and effort to thoroughly review our manuscript. We appreciate the constructive feedback and valuable insights provided during the review process.

We have carefully considered all the comments and suggestions made by the reviewers, and we are pleased to submit the revised version of our manuscript. In the revised document, you will find that we have addressed each concern raised during the review process. To facilitate the assessment of our revisions, we have highlighted all changes in red throughout the manuscript.

Referee #1:

In this revised manuscript, the authors performed key experiments that were brought up in my previous comments, included images with higher resolutions, reorganized the results and toned down some previous conclusions. These changes resulted in a much-improved manuscript that still holds novelty regarding SH3KBP1 function in myofiber formation. The authors addressed most of my major concerns of the original manuscript. I appreciate authors' responsiveness to my previous comments and their efforts in bringing the story to a higher level. I recommend publication of this beautiful study.

We would like to express our sincere thankfulness to Referee #1 for the review of our manuscript and the positive evaluation of our work.

One minor suggestion: in Figure 5D and E, it is better to include the color labeling in the figure panels, as they are different staining's from Figure 5F.

We thank the referee for this suggestion, the Figure 5D-E and the Figure legend have been modified accordingly.

Referee #4:

In the manuscript by Guiraud et al., the authors sought to identify novel factors that might be involved in myonuclear positioning. Using a limited siRNA screen, then identified Sh3kbp1 as a potential candidate. In vitro, depletion of Sh3kbp1 lead to aberrant nuclear positioning, increased myoblast fusion and altered t-tubule formation. In vivo, Sh3kbp1 depletion induced muscle wasting and weakness in both wild-type animals and a mouse model of centronuclear myopathy (Dnm2R465W). While the manuscript contains many interesting findings, the results are somewhat superficial and the link between the different results are not always obvious. For example, are defects in nuclear positioning related to changes in contractility/fiber type/atrophy/autophagy in vivo? If these mechanisms are distinct, my suggestion would be to focus on either the role of SH3KBP1 in nuclear positioning/myogenesis or the role in muscle maintenance/homeostasis. In addition, the role of Sh3kbp1 in ER/SR/triad structure maintenance in vivo still needs additional support. While clearly a lot of work went into the revised manuscript, for some aspects of the manuscript, there are still too strong of conclusions being drawn based on the current data.

We thank the referee for her/his insight and appreciate this constructive feedback. We clarified some aspects on the revised manuscript and performed additional experiments/quantifications to strengthen our conclusions. You will find below our answers to the specific comments.

Major Comments

- *One shortcoming of the paper is that there is no attempt to link the SH3KBP1 to known important regulators of nuclear spreading. Could it be that many of the phenotypes that you observe are due to disruption in the microtubule network? The authors acknowledge in the discussion that SH3KBP1 may interact with microtubule binding partners, so ruling out the direct involvement of microtubules would be important.*

We agree with this comment and we now provide in our revised manuscript an additional figure (New Expanded View Figure 2) that aims at clarifying the role of SH3KBP1 on the organization of the microtubule network in developing muscle fibers. To this end, we used stable C2C12 cell lines expressing either a scramble shRNA or a shRNA against *sh3kbp1* to decipher the organization of the microtubule network using live-Sir-tubulin® staining (Figure EV2A). This labeling indicates that the overall organization of microtubules seems not to be affected. For a more detailed analysis of this organization, we extracted from these experiments the directionality of the microtubules bundles in-between non-aggregated myonuclei and showed that the global orientation of the microtubule network is not impaired in the absence of SH3KBP1, excepted for a small pool of perpendicular microtubules (Figure EV2B, 90°). As shown in Figure 8B, the perinuclear Endoplasmic Reticulum architecture is disrupted in myotubes downregulated for SH3KBP1; we thus asked whether the recruitment of proteins able to drive/initiate the polymerization of microtubules such as Pericentrin or PCM1 at the nuclear membrane was affected. To do so, we quantified the number of myonuclei with a correct recruitment of both proteins at the vicinity of the myonuclei membrane and showed that Pericentrin and PCM1 were correctly recruited (Figure EV2C-F), suggesting that SH3KBP1 under-expression is not affecting microtubules anchoring at the myonuclei membrane. Finally, as we observed a slight increase in the number of perpendicular microtubules in myotubes in the absence of SH3KBP1 and as *Havrylov et al* identified using mass spectrometry, that the MTs-binding-proteins MAP7 was a potential interacting partner of SH3KBP1, we conducted immunoprecipitation assay using various constructs of MAP7 showing that SH3KBP1 does not interact with MAP7, excluding the possibility of a direct role of SH3KBP1 on the microtubule network organization *via* MAP7 interaction. **With these new experiments, we can conclude that SH3KBP1 has no direct role on the Microtubule network organization in myotubes, reinforcing the “Microtubule-independent” role of SH3KBP1 on myonuclei spreading.**

The modifications of the paper regarding this point are listed below:

Expanded View Figure 2

Figure legend, lines 697 to 714: **Expanded View Figure 2: SH3KBP1 is not affecting microtubule nucleation and organization in developing myotubes.** (A) Representative immunofluorescence staining of 5 days C2C12 myotubes expressing either scramble or *Sh3kbp1* shRNA and stained with sirTubulin®. Scale bars, 10 µm. (B) Quantification of the microtubule bundle directionality (orientation angle normalized according to myotubes longitudinal axis) in myotubes using the “directionality plugin” of ImageJ®. Unpaired t-test, *p < 0.05. (C-D) Representative images of immunofluorescent staining of Pericentrin (red), PCM1 (green) and nuclei (Blue) in 5 days differentiated C2C12 myotubes expressing either scramble or *Sh3kbp1* shRNA. Scale bars, 10 µm. (E-F) Quantification of the myonuclei peripheral staining of Pericentrin (E) or PCM1 (F) in 5 days differentiated C2C12 myotubes expressing either scramble or *Sh3kbp1* shRNA. Unpaired t-test, ***p < 0.001. (G) MAP7 constructs used in the experiment. (H) Representative western blot of crude extracts of C2C12 cells expressing various GFP-MAP7 constructs (FL: Full length, NT: N-terminal part of MAP7; NTL: N-terminal long part of MAP7; M: Middle part of MAP7, CT: C-terminal part of MAP7 and CTL: C-terminal long part of MAP7) and GFP-DNM2 and stained for endogenous SH3KBP1 (top) or with anti-GFP (bottom) antibodies. (I) Representative western blot after GFP immunoprecipitation (MAP7 and DNM2 constructs) using GFP-Trap in C2C12 cell extracts (see H). The membrane was revealed with anti-GFP (bottom) and anti-SH3KBP1 (Top) antibodies n.>3.

Results, lines 190 to 213: As myonuclei movement and spreading in developing myotubes have been largely related to the microtubule network (Cadot *et al*, 2012; Metzger *et al*, 2012; Ghasemizadeh *et al*, 2021; Wilson & Holzbaaur, 2012), we investigated the impact of *sh3kbp1* depletion on the microtubule network organization. Microtubule network was revealed by Sir-tubulin® stainings in 5 days differentiated myotubes obtained from stable C2C12 cell lines expressing either a scramble or an shRNA against *sh3kbp1*. In these conditions, we did not observe a dramatic impairment in the global microtubule network architecture/orientation, with the exception of the myotubes zones containing aggregated myonuclei (Figure EV2A). The quantification of the microtubule bundles orientation in-between individual myonuclei showed that the “microtubules bundle orientation” was not impaired in the absence of SH3KBP1, with the exception of a small pool of microtubules bundles perpendicular to the length of myotubes (Figure EV2B). As microtubule network emanates from the membrane of the nucleus in myotubes (Bugnard *et al*, 2005; Winje *et al*, 2018), we asked whether the recruitment of specific proteins able to drive/initiate the polymerization of microtubules at the nuclear membrane such as Pericentrin or PCM1 was affected (Figure EV2C-F). This approach shows that the re-localization of these proteins at the membrane of myonuclei was not affected by the absence of SH3KBP1. Finally, as we observed a small increase of the number of perpendicular microtubules (with respect to the longitudinal axis of myotubes) and that Havrylov *et al* previously identified the MTs-binding-proteins MAP7 as a potential interactor of SH3KBP1 protein, we performed immunoprecipitation assay using various constructs of MAP7 to evaluate the MAP7-SH3KBP1 interaction (Figure EV2G-I). Various GFP-tagged-MAP7 constructs were expressed in C2C12 cells and immunoprecipitated using GFP as a trap. However, we failed to identify an interaction with MAP7 in our conditions (Figure EV2I). **All together, these results suggested that the expression of SH3KBP1 during muscle fibers formation and maturation contribute to slow down myonuclei movements (time in motion and speed), which could contribute to a relative myonuclei motion stability during myotube formation, that in turn will facilitate myonuclei spreading, independently from any microtubule network organization impairment.**

Discussion, lines 449 to 454: Alternatively, our *in vitro* data suggest that the movements of organelles related to the MTs network are improved, reflected by an increase of the myonuclei motion in absence of SH3KBP1 (Figure 4). We show that SH3KBP1 does not impact the global organization/orientation of the microtubules network in developing myotubes and that the recruitment of microtubules nucleators such as pericentrin or PCM1 to the membrane of myonuclei is not altered, suggesting alternative processes in myonuclear displacement independently from the microtubule network.

Discussion, lines 470 to 475: Havrylov *et al* identified, using mass spectrometry, several MTs-binding-proteins such as MAP7 and MAP4 that can potentially interact with SH3 domains of SH3KBP1 and that have already been shown to control myonuclei positioning in myotubes (Metzger *et al*, 2012; Mogessie *et al*, 2015; Havrylov *et al*, 2009). We show in the present study that SH3KBP1 role in myonuclei spreading is not related to its interaction with MAP7 (Figure EV2 G-I) and thus suggest that additional partners of SH3KBP1 will have to be determined in the future.

Materials and methods lines 934 to 940: **Quantification methods for the orientation of microtubules bundles inside myotubes.** Myotubes were treated in live with sir-tubulin® according to the manufacturer protocol (spirochrome). Fluorescence microscopy was performed using Nikon AX confocal microscope with a 60X oil objective. Images obtained were cropped as shown in Figure EV2A (zooms) and image

analysis was performed in ImageJ® software using the directionality plugin (<https://imagej.net/plugins/directionality>). This provides quantifications of parameters regarding the orientation of Microtubules bundles classified by angles categories, that were then plotted using BoxPlotR (<http://shiny.chemgrid.org/boxplotr/>).

- *For many outcomes, it is difficult to decouple the changes in fusion from the changes in nuclear spreading. For example, in Fig. Suppl 1G, is this distance reduced because of reduced spreading or are there more nuclei in those myofibers because of enhanced fusion (Fig. 2)? It could be that this reduced distance is because there are simply more nuclei in the Sh3kbp1 depleted myofibers. Further, it would be very informative to perform the DiMyocMyo analysis at later time points to show whether the defect in nuclear spreading persists or is just delayed with Sh3kbp1 depletion.*

The reviewer rightly points out the difficulty to discriminate between myonuclei spreading failure and the consequences of an enhancement of myonuclei content in myotubes. Unfortunately, the extraction of DiMyocMyo parameter at later time points of differentiation is not possible as this method of analysis is restricted to “small” myotubes (no more than 15 myonuclei), relatively straight (not curved), to allow a correct extraction of both myotubes length and of DiMyocMyo parameters. Our *in vitro* mature myofibers are very long (millimeters of long), branched and curved as shown in Figure 4 C-D. Moreover, these myofibers are also quite heterogeneous regarding their length, orientation and maturity, making the use of the DiMyocMyo analysis method quite hazardous and thus explaining why we decided to quantify the mean distance between pairwise myonuclei in figure EV1G.

However, part of the answer can be extracted from our analysis in figure EV1G showing that, in control condition, the evolution of both myotube length and DiMyocMyo regarding myonuclei addition is relatively linear, with a myotube length elongation of about 25 µm each time a myonucleus is added to a myotube and a mean corresponding increase in the DiMyocMyo of about 7,3 µm. From this dataset, one can conclude that myonuclei spreading is not dependent (at least until 15 myonuclei) on the myonuclei content and thus on myonuclei fusion events. Also, depletion of SH3KBP1 enhances both the elongation and the DiMyocMyo but with a higher proportion for the DiMyocMyo parameter, and both myonuclei velocity and time in motion are enhanced and lead to myonuclei aggregation at a time in which fusion events has ended (Figure 4); it is thus tempting to conclude from our data that the initial defect brought by the depletion of SH3KBP1 is myonuclei spreading, that is then after worsened by the enhancement of myonuclei addition in myotubes.

- *I'm not sure I agree with the conclusion for Figure 4 that "...SH3KBP1 restrains myonuclei movements and motion during muscle fibers maturation, which contributes to correct myonuclear spreading.", as it appears that nuclear motion is not affected (or even enhanced) with Sh3kbp1 depletion.*

As mentioned in lines 187-188, in *sh3kbp1* depleted conditions “*myonuclei outside clusters were easily trackable and showed an increase of more than 20% of the percentage of time in motion and of more than 30% of the median speed*”. In this view, we can consider that a normal expression of SH3KBP1 minors the displacement of myonuclei and thus contributes to a relative myonuclei stability regarding both time in motion and velocity and thus may facilitate myonuclei spreading. We rephrase the conclusion to be more specific on this point.

Results lines 209 to 213: All together, these results suggested that the expression of SH3KBP1 during muscle fibers formation and maturation contribute to slow down myonuclei movements (time in motion and speed), which could contribute to a relative myonuclei motion stability during myotube formation, that in turn will facilitate myonuclei spreading, independently from any microtubule network organization impairment.

- One critical experiment that is missing is whether SH3KBP1 depletion has any functional effect on DNM2. The authors do a very nice job showing that SH3KBP1 interacts with DNM2 in myofibers, but fail to show whether SH3KBP1 depletion affects DNM2 localization. If the proposed mechanism is that SH3KBP1 interacts with DNM2 and this interaction is important for t-tubule formation via DNM2, then the prediction would be that DNM2 localization should be affected with SH3KBP1 depletion. If this is not the case, then it would suggest that SH3KBP1 is involved in t-tubule formation via some other mechanism. Similarly, the authors claim that "Sh3kbp1 exhibits a DNM2-like localization" but do not show any DNM2 immunostaining. Performing these experiments would strengthen the authors proposed mechanism.

We agree and we now provide in our revised manuscript an additional panel in Figure 5 illustrating the co-localization of DNM2 and SH3KBP1 by performing immunostaining in *ex vivo* myofiber extracted from *Tibialis Anterior* (new Figure 5D). Stainings confirm the interaction of DNM2 with SH3KBP1. However, DNM2 staining in our *in vitro* mature myofibers is not working in our hands, making the quantification of any SH3KBP1 depletion effect on DNM2 localization in developing myofibers not possible in our study. To try to counteract this problem, we conducted a co-expression experiment of both DNM2-GFP and full-length-SH3KBP1-Flag constructs in myofibers formed *in vitro* (see below the new Fig EV1J). With this experiment, we found that DNM2 over-expression in myofibers is detected as puncta that are predominantly positive for SH3KBP1 staining. Interestingly, DNM2 over-expression displaces SH3KBP1 as shown in Figure 8F, suggesting that DNM2 traps SH3KBP1 rather than the opposite. With these experiments, we reinforce our data showing that SH3KBP1 and DNM2 interact in muscle fibers despite the fact that we could not reveal the consequences of SH3KBP1 depletion on DNM2 localization. This will be investigated in the future.

Figure 5

Figure legends, lines 601 to 610: (D) Representative images of extracted *Tibialis Anterior* muscle fiber stained for SH3KBP1 (green), DNM2 (red) and myonuclei (blue) (single Z plan). Scale bars, 10 μ m. (E) Representative images of extracted *Tibialis Anterior* muscle fiber stained for SH3KBP1 (green), DHPR1 α (for

DyHydroPyridine Receptor alpha, red) and myonuclei (blue) (Max intensity of Z stacks plans). Scale bars, 10 μm . (F) Representative immunofluorescent staining of SH3KBP1 (green), Actin (Red) and myonuclei (blue) in the time course of C2C12 cells differentiation (proliferation (Prolif) and 3 or 5 days of differentiation (Diff Day3, Diff Day 5) are presented). (G) Representative immunofluorescent staining of SH3KBP1 (green), Actin (Red) and myonuclei (blue or red) along the time course of primary myoblasts cells differentiation (proliferation (Prolif) and 3 or 10 days of differentiation (Diff Day3, Diff Day 10) are presented). Scale bars, 10 μm .

Results, lines 224 to 245: This experiment confirmed that DNM2 interacts with SH3KBP1 through its N-terminal sequence, containing the SH3 domains of SH3KBP1 (Figure 5C). We conducted the same co-expression assay in 10 days *in vitro* primary myofibers and showed by immunofluorescence that DNM2 over-expression leads to the formation of small punctums along the length of myofibers, highly positive for SH3KBP1, confirming the association between these two proteins in myofibers (Figure EV1J). DNM2 is described as a component of the I-band in mature muscle fibers (Durieux *et al*, 2010) and more specifically localized at the Z-line (Neves *et al*, 2023). As expected, SH3KBP1 staining is present as transversal punctums, co-localizing in part with DNM2 staining (Figure 5D) and localized in-between the transversal DHPR staining (Figure 5E), validating SH3KBP1 as a component of the Z-line at the I-band zone. Interestingly, SH3KBP1 staining also strongly accumulated at the vicinity of myonuclei, forming a “cage” around myonuclei (Figure 5E). Moreover, SH3KBP1 localization related to myonuclei was confirmed using cross section of *Tibialis Anterior* mice muscle, highlighting that this localization seems specific to muscle cells (Figure EV3A).

We next addressed SH3KBP1 subcellular localization during the time-course of myotube formation using C2C12 myoblasts cell line and of myofiber formation using primary murine myoblasts (Figure 5 F-G). In proliferative conditions, SH3KBP1 is dispersed throughout the cytoplasmic compartment with an apparent higher concentration at the vicinity of myonuclei (Figure 5 F-G, Prolif). In myotubes, SH3KBP1 seems to spread along myotubes length and exhibits stronger labeling at the perinuclear zone (Figure 5 F-G, C2C12 Day 3 and 5; Primary Diff Day 3). Finally, in “mature-like” myofibers obtained using primary murine myoblasts, SH3KBP1 is still strongly accumulated at myonuclei vicinity and also exhibits longitudinal/transversal staining (Figure 5G, Diff Day 10). Myonuclei related localization of SH3KBP1 was also confirmed using staining of cross sections of human muscle biopsies (Figure EV3B-C). **These results show that SH3KBP1 binds to DNM2 in skeletal muscle fibers and localizes at the Z-line in the I-band zone and also at the vicinity of myonuclei.**

- While I appreciate the inclusion of the Ca^{2+} transient studies, the interpretation of the results are difficult for this non-electrophysiologist. Many parameters were not affected by *Sh3kbp1* depletion (transient time course, amplitude increase with voltage, mid-activation voltage, and slope factor), except for maximal rate of SR Ca^{2+} release. Is this an expected result if triads are disrupted?

Yes, the reduction of only the maximum rate of Ca^{2+} release is consistent with the disruption of triads. The coupling between t-tubule membrane voltage change and SR Ca^{2+} release relies on the interaction between two proteins: the voltage-sensing DHPR in the t-tubule membrane, and the Ca^{2+} release channel (type 1 ryanodine receptor, RyR1) in the SR membrane. The $\alpha 1$ subunit of the DHPR carries the voltage sensitivity of the process, so that unless the structure of $\alpha 1$ or its regulation (for instance by accessory subunits) is altered, no change in the voltage-dependence of the process should be anticipated. Since there is no indication whatsoever that SH3KBP1 interacts with the DHPR (no co-localization), the unchanged amplitude of the transients versus voltage, and the unchanged mid-activation voltage and slope parameters are consistent with the fact that the inherent functional properties of the DHPR are maintained. The rhod-2 Ca^{2+} transient time course is also not affected during the stimulations; this means that the time course of RyR1 channels open probability during a given voltage stimulation is unaffected, indicating that the regulation of opening and closure of those RyR1 channels that still work in the SH3KBP1-depleted fibers is unimpaired. Thus, we are essentially left with a reduced maximal amplitude of Ca^{2+} release, consistent with a reduced density of activatable RyR1 channels, due to loss of coupling with the DHPR because of triadic disruption.

Results, lines 284 to 293: Fitting the relationship in each fiber with a Boltzmann function showed that the mean value for maximal rate of SR Ca^{2+} release (Max $d[\text{Ca}^{2+}]/dt$) was significantly reduced in *Sh3kbp1* depleted fibers while mid-activation voltage ($V_{0.5}$) and slope factor (k) of the voltage-dependency were statistically

unchanged (Figure 6H, Figure EV4E-F). The reduction of only the maximum rate of Ca²⁺ release indicates that the voltage sensing properties of the process, carried by the DHPR, are unaffected by SH3KBP1 depletion, and that the SR Ca²⁺ release channels activation and inactivation properties during voltage activation are also maintained. Thus, triadic disruption is most likely reducing the density of activatable Ca²⁺ release channels, resulting in a reduction of maximum Ca²⁺ release with unaffected voltage-dependent and kinetic properties. **Overall, these results suggest that SH3KBP1 is required for proper ECC process through triads integrity maintenance.**

- One thing that is not clear is the lasting effect of Sh3kbp1 depletion when myofibers are matured for 10 days. Since this is a transient transfection early in differentiation, is Sh3kbp1 still depleted at day 10? Added rigor would include showing immunofluorescent staining for Sh3kbp1 in the siRNA depleted fibers at Day 10 (Fig. 6A) to effectively interpreted at what point Sh3kbp1 expression is required for t-tubule formation. Currently, Sh3kbp1 depletion is only shown using western blot and the time point is not described.

We agree with the referee and our revised manuscript contains two additional panels showing the down-regulation of SH3KBP1 protein, detected using western blotting techniques, in primary myofibers using siRNA (30% decrease compare to scramble siRNA, Figure EV1C) or ShRNA (50% decrease compare to scramble shRNA, Figure EV1D). The previous Western Blot was performed on primary myotubes for 3 days (Figure EV1B) and we now made it clear in the Figure legend.

Expanded View Figure 1

Figure legends, lines 676 to 697: Expanded View Figure 1: SH3KBP1 affects myotubes elongation and myonuclei spreading. (A) Sequential steps performed to obtain mature myofibers from primary myoblasts. siRNAs & shRNAs were transfected 24 hours before myoblasts fusion. (B-D) Western blot analysis of SH3KBP1 protein levels in total protein extracts obtained after sh3kbp1 depletion using either 2 individual siRNA (1 & 2) or a pool of siRNA (Mix) after 3 days of differentiation of primary myoblasts (B); a pool of siRNA (Mix) or a shRNA targeting SH3KBP1 after 10 days of differentiation of primary myoblasts (C and D respectively). Tubulin or GAPDH are used as loading control. (E-F) Myotubes length (E) and mean distances between each myonuclei and myotube centroids (DiMycMyo) (F) ranked by myonuclei content per myotubes were quantified after 3 days of differentiation in cells treated with scramble or *Sh3kbp1* siRNAs. Data from three independent experiments were combined. Scramble siRNA cells (n=1010) and *Sh3kbp1* siRNA cells (n=1093), Unpaired t-test, ***p < 0.001.

Center lines show the medians; box limits indicate the 25th and 75th percentiles as determined by R software; whiskers extend 1.5 times the interquartile range from the 25th and 75th percentiles, outliers are represented by dots. (G-H) Four representative images of 10 days differentiated myofibers transfected with either scramble (D) or *Sh3kbp1* shRNA tagged with GFP (E); shRNA (Green) myonuclei (DAPI, red). Scale Bar: 10 μ m. (Asterisks are individual myonuclei) (I) Quantification of the mean distance between pairwise myonuclei in 10 days differentiated myofibers treated with scramble siRNA or shRNA, or with a pool of 2 individual siRNAs or of one individual shRNAs targeting *Sh3kbp1*. Data from three independent experiments were combined. Center lines show the medians; box limits indicate the 25th and 75th percentiles as determined by R software; whiskers extend 1.5 times the interquartile range from the 25th and 75th percentiles, outliers are represented by dots. Unpaired t-test, ***p < 0.001. (J) Representative immunofluorescent staining of *in vitro* myofibers co-expressing GFP-DNM2 (green), SH3KBP1-Flag (red) and myonuclei (blue) in primary myotubes differentiated for 10 days. (Max intensity of Z stacks plans) Scale bars, 10 μ m.

- The rationale for looking at *Sh3kbp1* in myopathies caused by *Dnm2* is not clear, as it is still not known what *Sh3kbp1* depletion does to *Dnm2* *in vivo*, or vice versa. One possibility could be that the *Dnm2* mutation affects its interaction with *Sh3kbp1*. This should be tested, otherwise modulation the expression of *Sh3kbp1* does not make sense.

There are several reasons justifying the interest of looking at the status of SH3KBP1 in a mouse model of myopathy caused by DN2: (1) we identified for the first time SH3KBP1 as a modulator of myonuclei spreading/positioning in myofibers (Figure 1-4) whereas mispositioning of myonuclei is a hallmark of the DN2-related “centronuclear” myopathy”, (2) our study demonstrates the involvement of SH3KBP1 in the maintenance of the homeostasis of myofibers (Figure7) which is also disrupted in the DN2 myopathy, (3) we demonstrated that SH3KBP1 interacts with DN2 in muscle fibers (Figure 5), (4) DN2^{R465W/+} mutation is the most frequent mutation associated with autosomal dominant centronuclear myopathy and (5) *Sh3kbp1* mRNA expression is elevated in muscle from the DN2^{R465W/+} mouse model, suggesting a “protective” role of SH3KBP1 over-expression in the development of CNM associated phenotypes in this animal.

As explained above, DN2 staining in our *in vitro* mature myofiber system is not working in our hands, making the quantification of any defect of SH3KBP1 depletion on DN2 localization not possible. DN2 was previously described in HeLa cells as a SH3KBP1 interacting protein, through its proline-rich domain (C-terminus part of the protein) by Schroeder *et al* in 2010. As *Dnm2*^{R465W/+} mutation is in the middle domain of the DN2 protein, it is unlikely that its binding to SH3KBP1 will be modulated. To confirm this hypothesis, we conducted an *in vitro* co-expressing assay into myofibers with either GFP-DN2-WT or GFP-DN2-R465W with SH3KBP1-Flag in primary myotubes differentiated for 10 days (see below, figure PBP). We did not observe any changes in DN2-WT or DN2-R465W pattern in mature myofibers (puncta in myofibers) and all of those puncta were positive for SH3KBP1, suggesting (1) that both DN2-WT and DN2-R465W have the same ability to bind SH3KBP1 and (2) that over-expression of DN2 is “trapping” the vast majority of SH3KBP1 protein.

Figure Point by Point (PBP): DNM2 interaction with SH3KBP1. Representative immunofluorescent staining of *in vitro* myofibers co-expressing either GFP-DNM2-WT or GFP-DNM2-R465W (green) with SH3KBP1-Flag (red) in primary myotubes differentiated for 10 days. Myonuclei are stained by DAPI (blue). A-B is single Z plan. C-D is the Maximum intensity of Z stack plans. Scale bars, 10 μ m.

- Again, I'm not sure I understand the conclusion that ".....in pathological context such as in the CNM diseases, SH3KBP1 expression could counteract the progression of related phenotype." SH3KBP1 expression is already elevated in *Dnm2R465W* mice, and knocking it down actually made the phenotype worse in the mutant mice. So, would the idea be to increase SH3KBP1 levels even further in the diseased context?

Yes, this is exactly the idea but we agree with the referee that this hypothesis is not indicated in the appropriate location. We thus revised the conclusion of this paragraph.

Results, lines 336 to 338: These experiments show that SH3KBP1 is a key factor in the maintenance of muscle fibers homeostasis and particularly in the pathological context of CNM disease induced by DNM2 mutation.

- Fig. 8G. While the EM images are the best way to visualize triads, I'm not sure what information should be gathered from just representative EM images, as any differences are not obvious to the untrained eye. Some quantification is warranted.

We agree with the referee. Our revised manuscript contains now two quantifications regarding the EM images: in Figure EV5G, we quantified the distance between T-tubules from adjacent Z-line and in Figure EV5H, we quantified T-tubules width, allowing to better appreciate the modification observed.

Figure legends, lines 740 to 745: **Expanded View Figure 5:** (G-H) Quantification of the distances between adjacent T-tubules structures (G) and of individual T-tubule width (H) in *Tibialis Anterior* extracted muscle fibers from WT mice injected with either PBS or shRNA targeting *SH3KBP1* mRNA. 2 mice in each condition were combined. Center lines show the medians; box limits indicate the 25th and 75th percentiles as determined by R software; whiskers extend 1.5 times the interquartile range from the 25th and 75th percentiles, outliers are represented by dots. Unpaired t-test, *** $p < 0.001$, ** $p < 0.01$, * $p < 0.05$.

Results, lines 401 to 406: To better appreciate these modifications, we first quantified the distance between adjacent t-tubules that reveal a 10% decrease (Figure EV5G), in accordance with the immunofluorescence quantification using DHPR staining (Figure EV4C). Additionally, individual T-tubule width quantification was performed and showed a 2.4-fold increase, suggesting a T-tubule “dilatation” phenotype (Figure EV5H). **Altogether, our data suggest that SH3KBP1 is a key factor of muscle fibers homeostasis through an ER/SR/Triads structure maintenance.**

- *The rationale for looking at LC3 is not well supported and the method of quantification is not clear. In theory, every fiber will have some level of LC3+ structures at any time, but conclusions about autophagic flux cannot be determined from a single timepoint. An increase in LC3 could mean either enhanced autophagic flux or impaired flux, depending on what is causing the increase in LC3. Additional studies/control are needed to draw any conclusion about the role of Sh3kbp1 in autophagy.*

We clarified this point incorporating additional experiments performed on myotubes obtained from stable C2C12 myoblast clones expressing either Scramble or sh3kbp1 ShRNA (new Figure 8). We first quantified LC3-II levels in protein extracts and show a 1.4-fold increase of the LC3-II/Actin ratio in *Sh3kbp1*-depleted myotubes as compared to controls (New figure 8G-H), validating the results obtained using immunofluorescence approach on *Tibialis Anterior* cross-sections (Figure 8K). We then compared LC3-II levels in non-treated or bafilomycin A1-treated (100 nM, 6h) myotubes and showed that the BafA1 treatment increased by 2.5-fold the LC3-II/actin ratio in control myotubes, whereas in *Sh3kbp1*-depleted myotubes, the LC3-II/actin ratio induction was only increased by 1.4-fold, indicating an impaired maturation step of the autophagic process and thus an impaired autophagic flux (New figure 8I-J).

Figure 8

Results, lines 372 to 406: Since perinuclear ER architecture is altered in *Sh3kbp1*-depleted myotubes, we asked whether it could affect the autophagy, as ER is an important organelle for the functioning of the autophagic process (Mochida & Nakatogawa, 2022). Autophagy is characterized by the formation of double membrane vesicles called autophagosomes that engulf parts of the cytoplasm/organelles and fuse with acidic lysosome to form autolysosomes whose content will be degraded. Autophagy is an important actor of proteostasis and muscle homeostasis (Xia *et al*, 2021). To decipher the involvement of SH3KBP1 in the autophagy regulation, we quantified LC3 protein (Microtubule-associated protein 1A/1B-light chain 3), commonly used to address the autophagic activity due to its conjugation to the phosphatidylethanolamines of the autophagosomes membrane (LC3-II form). First, we observed a higher level of LC3-II in the total protein extracts of *Sh3kbp1*-depleted myotubes compare to control myotubes, indicating an increase of the autophagosome content (Figure 8G-H). To determine if this increased number of autophagosome is the consequence of an increase or blockade of the autophagic flux, we treated myotubes with bafilomycin-A1, an inhibitor of the maturation step of the autophagic flux. We show that the inhibition of autophagy by bafilomycin-A1 induced a 2.5-fold increase of the LC3-II level in control myotubes while in SH3KBP1-depleted myotubes the increase was only of 1.4-fold (Figure 8I-J). Thus, our results indicate that the fusion of autophagosomes with lysosomes is altered in SH3KBP1-underexpressing myotubes. Of interest, the *KI-Dnm2*^{R465W/+} mice model is also associated to an autophagy impairment (Puri *et al*, 2020; Puri & Rubinsztein, 2020). LC3 labeling was thus also realized in cross-sections of *Sh3kbp1* depleted fibers in both WT and *KI-Dnm2*^{R465W/+} mice. In WT condition, the quantification of densely-labeled LC3 myofibers showed a slight increase when we compared *Sh3kbp1*-depleted muscles to control ones, validating our previous *in cellulo* data (Figure 8K, WT condition). In the *KI-Dnm2*^{R465W/+} mice, the number of dense positive-LC3 fibers was increased up to 5% of total fibers, in accordance to previous study (Rabai *et al*, 2019). Moreover, down regulation of SH3KBP1 was sufficient to increase by more than 2.5-fold the ratio of LC3-dense fibers, showing a preserved role of SH3KBP1 in the support of the dynamics of the autophagic flux in skeletal muscle fibers.

Discussion, lines 505 to 513: The use of bafilomycin-A1 allowed us to conclude that this increase is a consequence of an alteration of the maturation step of the autophagic process and thus to a decreased autophagy dynamic. This autophagy-dependent modulation of muscle homeostasis could explain the decrease of the Cross Section Area of myofibers (Figure 7G) and ultimately the reduction of the numbers of muscle myofibers by muscle (Figure EV5D). Interestingly, autophagy genes have also been involved in muscle myonuclei positioning during *Drosophila* metamorphosis (Fujita *et al*, 2017). Moreover, SH3KBP1 interacts with dynamin-2 (our study), which is also involved in the autophagic lysosome reformation step of the process (Schulze *et al*, 2013). Whether this process is involved in SH3KBP1-dependent myonuclei positioning will be the subject of further investigations.

Figure legends, lines 659 to 672: **Figure 8: SH3KBP1 is an endoplasmic reticulum scaffolding protein and supports the autophagic pathway in skeletal muscle fibers** (G) Control (Sh-Scramble) and SH3KBP1-depleted (sh-Sh3kbp1) C2C12 myotubes, differentiated during 6 days, were analyzed for their content of LC3-I/LC3-II and SH3KBP1 proteins by western blot; Actin labeling was used as a loading control. (H) Fold change quantification: LC3-II/Actin ratios from 4 western blots were calculated and reported to the Scramble condition. (I) Control (Scramble) and SH3KBP1-depleted (Sh3kbp1) myotubes differentiated during 6 days were either left untreated or treated with 100 nM of bafilomycin A1 during 6 hours. After total protein extraction, LC3-I/LC3-II and Actin levels were analyzed by immunoblot. (J) Fold change quantification: LC3-II/Actin ratios from 3 western blots were calculated and reported to the untreated condition of each type of myotube (Scramble or Sh3KBP1). Wilcoxon-Mann-Whitney two-sided test, **p < 0.01, *p < 0.05. (K) Quantification of the percentage of highly LC3 positive myofibers in *Tibialis Anterior* muscles from WT or *KI-Dnm2*^{R465W/+} injected with either PBS or shRNA targeting *SH3KBP1* mRNA. Data from at least 4 mice in each condition were combined. Unpaired t-test, ***p < 0.001. Center lines show the medians; box limits indicate the 25th and 75th percentiles as determined by R software; whiskers extend 1.5 times the interquartile range from the 25th and 75th percentiles, outliers are represented by dots.

Materials and methods, lines 781 to 788: Inhibition of autophagy maturation was performed by treating C2C12 myotubes with Bafilomycin A1 (MedChemExpress, HY-100558) at 100 nM during 6 hours. To generate C2C12 stably expressing shRNAs (shControl or shCin85), cells were transfected with plasmids encoding control shRNA (Mission pLKO.1-Puro non mammalian shRNA control plasmid, Merck SHC002) or shRNA directed against SH3KBP1 (mission shRNA clone TRCN0000088508 targeting the 3'UTR sequence CCCACCACTCTAAGAGAAATT) and selected using 2 µg/mL of Puromycin (Gibco, A1113803) for two weeks. The clones were then amplified and analyzed for their content of Cin85 protein, proliferation and differentiation capacities.

Minor Comments

- *Figure 1B is very difficult to interpret at first glance. I guess it should be "% change relative to scramble" and not "% of scramble", otherwise I would think that scramble would be equal to 100%. I think the way that it is shown now, it is actually calculated as a fold increase not percent increase. For example, based on Table 1, by my calculation Sh3kbp1 has either a 1.48-fold increase or 47.23% increase in length compared to scramble control.*

We thank the reviewer for pointing out this discrepancy. The legend has been changed accordingly.

- *Similarly, the wording in line 110 and 111 is confusing. I think authors mean that every candidate appears to be a negative regulator of myotube length expansion, as knocking them down lead to an increase in myotube length relative to the scramble control. I suggest rewording for clarity.*

We thank the reviewer and we have now edited the sentence accordingly.

Results, lines 112 to 115: Interestingly, we found that nearly all selected candidates appear as negative regulator of myotube length expansion, as knocking them down lead to an increase in myotube length relative to the scramble control condition, with the exception of KIFAP3 protein.

- *Suppl Fig 1A. - at what time was knockdown assessed?*

We edited the figure legend with the time of the knock-down assessment.

- *In Fig 3H, it is not clear which conditions are being compared for the statistics.*

We now have made appropriate changes in the figure to visualize which conditions are compared.

- I suggest adding labels for the different antibodies in Fig 5 D and E, similar to how it is in 5F. It will make the figure easier to follow.

Done

- Perhaps it's because it's a maximum intensity projection, but I do not see SH3KBP1 as a "nuclear cage" in Fig. 5F and Suppl. Fig. 2. It appears to be clearly inside the nucleus *in vivo*. Do the authors have any explanation for this localization?

Yes, this is because we used the maximum intensity projection. We mentioned in figures legends, each time this technique was used.

- Double check line 207 - it seems some text may be missing.

We edited the sentence accordingly.

- How were triads quantified? Was it an automated analysis? This analysis would seem to be highly subjective.

We adapted a previously published technique to measure *in vitro* the concatenation of triads proteins and Actin structure (Roman *et al*, Nat Cell Biol, 2017). However, we made a shortcut by naming it "triads" whereas it corresponds to DHPR doublets. We have edited the figure accordingly. Regarding the quantification of DHPR doublet categories, we provided in Figure EV4A a panel of DHPR staining, illustrating the 3 categories.

- What does GFP represent in Fig 6E? It is not described in the methods whether the AAV plasmid also contains a GFP.

We thank the reviewer for pointing out this imprecision: two different approaches were used *in vivo*, to down-regulate SH3KBP1: electroporation of shRNA tagged with GFP or AAV viruses (without GFP). In figure 6, we only used AAV viruses. We now have made appropriate changes in the figure and in the text accordingly.

- Similarly, it is difficult to interpret the representative images in 6E without any sarcomeric marker.

We edited the figure 6 and the text accordingly.

- Fig. 7A. Normalizing to Nat10 seems a bit odd. Is there a reason for using this somewhat obscure gene?

No special reason except that NAT10 expression is relatively stable along muscle development.

Referee #5:

This revised manuscript by Guiraud and colleagues provides strong evidence that SH3KBP1 is an adapter protein that has important roles in muscle formation, nuclear positioning, and possibly homeostatic maintenance. This reviewer finds this to be an interesting work, and the authors have responded well to previous reviews.

I only have one comment on the work: It is interesting that the authors comment on the importance of microtubule organization in muscle, however did not investigate how microtubules are involved in the the SH3KBP1-mediated processes. Particularly in regards to nuclear positioning, the work of Wilson & Holtzbaaur (2012) suggest that microtubule association with the nuclear membrane may be disrupted by SH3KBP1- thus contributing to the observations of this report.

We agree with this comment and we now provide in our revised manuscript an additional figure (New Expanded View Figure 2) that aims at clarifying the role of SH3KBP1 on the organization of the microtubule network in developing muscle fibers. To this end, we used stable C2C12 cell lines expressing either a scramble shRNA or a shRNA against *sh3kbp1* to decipher the organization of the microtubule network using live-Sir-tubulin[®] staining (Figure EV2A). This labeling indicates that the overall organization of microtubules seems not to be affected. For a more detailed analysis of this organization, we extracted from these experiments the directionality of the microtubules bundles in-between non-aggregated myonuclei and showed that the global orientation of the microtubule network is not impaired in the absence of SH3KBP1, excepted for a small pool of perpendicular microtubules (Figure EV2B, 90°). As shown in Figure 8B, the perinuclear Endoplasmic Reticulum architecture is disrupted in myotubes downregulated for SH3KBP1; we thus asked whether the recruitment of proteins able to drive/initiate the polymerization of microtubules such as Pericentrin or PCM1 at the nuclear membrane was affected. To do so, we quantified the number of myonuclei with a correct recruitment of both proteins at the vicinity of the myonuclei membrane and showed that Pericentrin and PCM1 were correctly recruited (Figure EV2C-F), suggesting that SH3KBP1 under-expression is not affecting microtubules anchoring at the myonuclei membrane. Finally, as we observed a slight increase in the number of perpendicular microtubules in myotubes in the absence of SH3KBP1 and as *Havrylov et al* identified using mass spectrometry, that the MTs-binding-proteins MAP7 was a potential interacting partner of SH3KBP1, we conducted immunoprecipitation assay using various constructs of MAP7 showing that SH3KBP1 does not interact with MAP7, excluding the possibility of a direct role of SH3KBP1 on the microtubule network organization *via* MAP7 interaction. **With these new experiments, we can conclude that SH3KBP1 has no direct role on the Microtubule network organization in myotubes, reinforcing the “Microtubule-independent” role of SH3KBP1 on myonuclei spreading.**

The modifications of the paper regarding this point are listed below:

Expanded View Figure 2

Figure legend, lines 697 to 714: **Expanded View Figure 2: SH3KBP1 is not affecting microtubule nucleation and organization in developing myotubes.** (A) Representative immunofluorescence staining of 5 days C2C12 myotubes expressing either scramble or *Sh3kbp1* shRNA and stained with sirTubulin[®]. Scale bars, 10 μ m. (B) Quantification of the microtubule bundle directionality (orientation angle normalized according to myotubes longitudinal axis) in myotubes using the “directionality plugin” of ImageJ[®]. Unpaired t-test, * $p < 0.05$. (C-D) Representative images of immunofluorescent staining of Pericentrin (red), PCM1 (green) and nuclei (Blue) in 5 days differentiated C2C12 myotubes expressing either scramble or *Sh3kbp1* shRNA. Scale bars, 10 μ m. (E-F) Quantification of the myonuclei peripheral staining of Pericentrin (E) or PCM1 (F) in 5 days differentiated C2C12 myotubes expressing either scramble or *Sh3kbp1* shRNA. Unpaired t-test, *** $p < 0.001$. (G) MAP7 constructs used in the experiment. (H) Representative western blot of crude extracts of C2C12 cells expressing various GFP-MAP7 constructs (FL: Full length, NT: N-terminal part of MAP7; NTL: N-terminal long part of MAP7; M: Middle part of MAP7, CT: C-terminal part of MAP7 and CTL: C-terminal long part of MAP7) and GFP-DNM2 and stained for endogenous SH3KBP1 (top) or with anti-GFP (bottom) antibodies. (I) Representative western blot after GFP immunoprecipitation (MAP7 and DNM2 constructs) using GFP-Trap in C2C12 cell extracts (see H). The membrane was revealed with anti-GFP (bottom) and anti-SH3KBP1 (Top) antibodies $n > 3$.

Results, lines 190 to 213: As myonuclei movement and spreading in developing myotubes have been largely related to the microtubule network (Cadot *et al*, 2012; Metzger *et al*, 2012; Ghasemizadeh *et al*, 2021; Wilson & Holzbaaur, 2012), we investigated the impact of *sh3kbp1* depletion on the microtubule network organization. Microtubule network was revealed by Sir-tubulin[®] stainings in 5 days differentiated myotubes obtained from stable C2C12 cell lines expressing either a scramble or an shRNA against *sh3kbp1*. In these conditions, we did not observe a dramatic impairment in the global microtubule network architecture/orientation, with the exception of the myotubes zones containing aggregated myonuclei (Figure EV2A). The quantification of the microtubule bundles orientation in-between individual myonuclei showed that the “microtubules bundle orientation” was not impaired in the absence of SH3KBP1, with the exception of a small pool of microtubules bundles perpendicular to the length of myotubes (Figure EV2B). As microtubule network emanates from the membrane of the nucleus in myotubes (Bugnard *et al*, 2005; Winje *et al*, 2018), we asked whether the recruitment of specific proteins able

to drive/initiate the polymerization of microtubules at the nuclear membrane such as Pericentrin or PCM1 was affected (Figure EV2C-F). This approach shows that the re-localization of these proteins at the membrane of myonuclei was not affected by the absence of SH3KBP1. Finally, as we observed a small increase of the number of perpendicular microtubules (with respect to the longitudinal axis of myotubes) and that *Havrylov et al* previously identified the MTs-binding-proteins MAP7 as a potential interactor of SH3KBP1 protein, we performed immunoprecipitation assay using various constructs of MAP7 to evaluate the MAP7-SH3KBP1 interaction (Figure EV2G-I). Various GFP-tagged-MAP7 constructs were expressed in C2C12 cells and immunoprecipitated using GFP as a trap. However, we failed to identify an interaction with MAP7 in our conditions (Figure EV2I). **All together, these results suggested that the expression of SH3KBP1 during muscle fibers formation and maturation contribute to slow down myonuclei movements (time in motion and speed), which could contribute to a relative myonuclei motion stability during myotube formation, that in turn will facilitate myonuclei spreading, independently from any microtubule network organization impairment.**

Discussion, lines 449 to 454: Alternatively, our *in vitro* data suggest that the movements of organelles related to the MTs network are improved, reflected by an increase of the myonuclei motion in absence of SH3KBP1 (Figure 4). We show that SH3KBP1 does not impact the global organization/orientation of the microtubules network in developing myotubes and that the recruitment of microtubules nucleators such as pericentrin or PCM1 to the membrane of myonuclei is not altered, suggesting alternative processes in myonuclear displacement independently from the microtubule network.

Discussion, lines 470 to 475: *Havrylov et al* identified, using mass spectrometry, several MTs-binding-proteins such as MAP7 and MAP4 that can potentially interact with SH3 domains of SH3KBP1 and that have already been shown to control myonuclei positioning in myotubes (*Metzger et al, 2012; Mogessie et al, 2015; Havrylov et al, 2009*). We show in the present study that SH3KBP1 role in myonuclei spreading is not related to its interaction with MAP7 (Figure EV2 G-I) and thus suggest that additional partners of SH3KBP1 will have to be determined in the future.

Materials and methods lines 934 to 940: **Quantification methods for the orientation of microtubules bundles inside myotubes.** Myotubes were treated in live with sir-tubulin® according to the manufacturer protocol (spirochrome). Fluorescence microscopy was performed using Nikon AX confocal microscope with a 60X oil objective. Images obtained were cropped as shown in Figure EV2A (zooms) and image analysis was performed in ImageJ® software using the directionality plugin (<https://imagej.net/plugins/directionality>). This provides quantifications of parameters regarding the orientation of Microtubules bundles classified by angles categories, that were then plotted using BoxPlotR (<http://shiny.chemgrid.org/boxplotr/>).

Dear Dr. GACHE

Thank you for the submission of your revised manuscript to EMBO reports. We have now received the report from referee #4 whom we had asked to assess it (copied below).

As you can see, the referee finds that the revision has significantly strengthened your study and recommends publication. Before I can accept the manuscript, I need you to address some minor points below:

- Please provide up to 5 keywords on the title page.
- Please update the 'Conflict of interest' paragraph to our new 'Disclosure and competing interests statement'. For more information see <https://www.embopress.org/page/journal/14693178/authorguide#conflictsofinterest>
- We noted the following author name discrepancies between the manuscript file (ms) and the information provided in the online manuscript tracking system (eJP): Norma Beatriz Romero in the ms vs. Norma Romero in eJP; Mai Thao Bui in the ms file vs. Thao Mai Bui in eJP.
- Corresponding author(s) need to be clearly labeled on the title page of the manuscript and their email(s) need to be provided.
- The information on funding is not congruent as grant number R14074CS needs to be listed in the manuscript too (currently only in the online system).
- Funding should be part of Acknowledgments section.
- Regarding the Author Contributions, we now use CRedit to specify the contributions of each author in the journal submission system. Therefore, please remove the Author Contributions from the manuscript file and make sure that the author contributions in our online manuscript tracking system are correct and up-to-date. The information you specified in the system will be automatically retrieved and typeset into the article. You can enter additional information in the free text box provided, if you wish.
- Author checklist: please complete the information on Experimental animals/ model organisms (D57) and on Data availability/ primary datasets (D112).
- Experiments involving mice: Please provide the reference number for approval in addition to the specification of the ethics committee.
- Please provide a Reagents and Tools Table listing key reagents, experimental models, software and relevant equipment and including their sources and relevant identifiers. You can download our Reagents and Tools Table template (.docx), in our author guidelines: <https://www.embopress.org/page/journal/14693178/authorguide#structuredmethods>. When submitting your revised manuscript, please do not include the Reagents and Tools Table in the Methods section of the manuscript but upload it as a separate file choosing the file type "Reagent Table".

Please incorporate the tables that are currently in the methods section (antibodies etc) in this table.

- I did a spot check on the source data at Biostudies and could not locate the cells shown in Figure 2A. Maybe I just missed the relevant part that was chosen for the figure panel, but please make sure that the supplied source data is the correct one. (Btw: the images from the two channels have slightly different length and width. In order to merge the two channels in Fiji, I had to copy-paste the images into new ones with the same size.)
- Quantitative source data: Figure 3B: the numbers in column D, Efficiency, are all the same. Is this OK, or was there an error?
- Please provide the main and EV figures as individual production quality figure files.
- Table 1-3 are Datasets and should be titled Dataset EV1-EV3; their legends need to be removed from the manuscript and inserted in each Excel file (as a separate sheet/tab). Then please upload these as file type "Dataset" and also update the callouts in the manuscript.
- Movies: the correct nomenclature (source file names, titles in the online manuscript tracking system, manuscript callouts) should be Movie EV1 and Movie EV2. Each legend should be removed from the manuscript and provided as a readme.txt file,

then each movie should be zipped up with its legend and uploaded as zip folder Movie EV1, Movie EV2.

- The summary should be removed from the manuscript file as well as the heading "Additional information".
- The manuscript sections should be in the following order: Title page - Abstract & Keywords - Introduction - Results - Discussion - Methods - Data Availability - Acknowledgments - Disclosure Statement & Competing Interests - References - Figure Legends - (Main Tables with legends if applicable) - Expanded View Figure Legends.
- Materials and methods should be Methods.
- The nomenclature for the EV figures should be "Figure EV1" instead of "Expanded View Figure 1"; this needs to be corrected in the legends in the manuscript file and the individual Figure files.
- Our production/data editors have asked you to clarify several points in the figure legends (see below). Please incorporate these changes in the manuscript and return the revised file with tracked changes with your final manuscript submission.

A) Statistical test information. Only p-values that are actually shown in the figure panel(s) should (and must) be defined in the legends, all others should be removed from (or added to) the legend. Moreover, we ask for the specification of exact p-values:

- Please note that the exact p values are not provided in the legends of figures 2D-F; 3H, 4B, H, I; 6D, H; 7A, C, D, G, H, I; 8J, K; EV1 C, D, I; EV2 B; EV4 B, C; EV5 A, B, D, G, H;
- Please indicate what */ **/ ***/ **** represents; if this represents p value(s), please indicate the statistical test used and where appropriate, specify the exact p value in the legend(s) of figure(s) 8H"

B) Replicates and error bars:

- Please note that the box plots need to be defined in terms of minima, maxima in the legends of figures 2C-F; 3H; 7A, C, D, G, H, I; 8H, J, K; EV1 C, D, I; EV5 A, B, D, E, F-H.
- Please note that the box plots need to be defined in terms of minima, maxima, centre, bounds of box and whiskers, and percentile in the legends of figures 4B, H, I; 6D, H; EV2 B; EV4 B, E, F; G, H.
- Please note that information related to n is missing in the legends of figures 6H, 8H; EV2 B, E, F.
- Please note that the error bars are not defined in the legends of figures EV2 E, F; EV4 C, D; EV5 C

- As a standard procedure, we edit the title and abstract of manuscripts to make them more accessible to a general readership. Please find my suggestions below my signature and please check them for accuracy.

- Finally, EMBO Reports papers are accompanied online by

- A) a short (1-2 sentences) summary of the findings and their significance,
- B) 2-3 bullet points highlighting key results and
- C) a schematic summary figure that provides a sketch of the major findings (not a data image).

Please provide the summary figure as a separate file in PNG or JPG format at a size of 550x300-600 pixels (width x height). Please note that the size is rather small and that text needs to be readable at the final size. Please send us this information along with the revised manuscript.

With kind regards,

=====

Referee #4:

I commend the authors on their efforts in addressing the reviewer's comments. The new data greatly strengthens the conclusions and adds to the manuscript's overall quality.

=====

SH3KBP1 promotes skeletal myofiber formation and functionality through ER/SR architecture integrity

Dynamic changes in the arrangement of myonuclei and the organization of the sarcoplasmic reticulum are important determinants of myofiber formation and muscle function. To find factors associated to muscle integrity, we perform an siRNA screen and identify SH3KBP1 as a new factor controlling myoblast fusion, myonuclear positioning and myotube elongation. We find that the N-terminus of SH3KBP1 binds to dynamin-2 while the C-terminus associates with the endoplasmic reticulum through calnexin, which in turn control myonuclei dynamics and ER integrity, respectively. Additionally, in mature muscle fibers, SH3KBP1 contributes to the formation of triads and modulates the Excitation-Contraction Coupling process efficiency. In *Dnm2^{R465W/+}* mice, a model for centronuclear myopathy (CNM), depletion of *Sh3kbp1* expression aggravates CNM-related atrophic phenotypes and impaired autophagic flux in mutant skeletal muscle fiber. Altogether our results identify SH3KBP1 as a new regulator of myofiber integrity and function.

All editorial and formatting issues were resolved by the authors.

Dr. vincent GACHE
INSERM
U1217
Neuromyogene institute, team MNCA
8 avenue Rockefeller
Lyon 69008
France

Dear Dr. GACHE,

I am very pleased to accept your manuscript for publication in the next available issue of EMBO reports. Thank you for your contribution to our journal.

Yours sincerely,
